# Effects of visual inputs on neural dynamics for coding of location and running speed in medial entorhinal cortex

**Holger Dannenberg\*, Hallie Lazaro, Pranav Nambiar, Alec Hoyland, Michael E Hasselmo**

Center for Systems Neuroscience, Department of Psychological and Brain Sciences, Boston University, Boston, United States

**Abstract** Neuronal representations of spatial location and movement speed in the medial entorhinal cortex during the 'active' theta state of the brain are important for memory-guided navigation and rely on visual inputs. However, little is known about how visual inputs change neural dynamics as a function of running speed and time. By manipulating visual inputs in mice, we demonstrate that changes in spatial stability of grid cell firing correlate with changes in a proposed speed signal by local field potential theta frequency. In contrast, visual inputs do not alter the running speed-dependent gain in neuronal firing rates. Moreover, we provide evidence that sensory inputs other than visual inputs can support grid cell firing, though less accurately, in complete darkness. Finally, changes in spatial accuracy of grid cell firing on a 10 s time scale suggest that grid cell firing is a function of velocity signals integrated over past time.

**\*For correspondence:**
hdannenb@gmail.com

## Introduction

The spatially periodic firing of grid cells in the medial entorhinal cortex (MEC) (*Fyhn et al., 2008*; *Fyhn et al., 2004*; *Hafting et al., 2005*) is widely recognized as a neuronal correlate of memory-guided and/or self-motion-based spatial navigation (*McNaughton et al., 2006*). Based on that idea, computational models of grid cells including attractor network models (*Burak and Fiete, 2009*; *Fuhs and Touretzky, 2006*), oscillatory interference models (*Blair et al., 2008*; *Burgess, 2008*; *Burgess et al., 2007*; *Hasselmo, 2008*; *Zilli and Hasselmo, 2010*), hybrid models of attractor networks and oscillatory interference (*Bush and Burgess, 2014*), and probabilistic learning models (*Cheung, 2016*) suggest that grid cells integrate running direction and traveled distance to maintain a spatially periodic firing pattern during path integration (*Gil et al., 2018*; *Jacob et al., 2019*). Path integration depends on a velocity signal which depends on vestibular and proprioceptive signals (*Chen et al., 2019*; *Tennant et al., 2018*) and also moving sensory features such as optic flow (*Campbell et al., 2018*; *Raudies et al., 2012*; *Raudies and Hasselmo, 2015*). However, any path integration mechanism ultimately suffers from the accumulation of error over distance traveled or elapsed time (*Kraus et al., 2015*; *Sreenivasan and Fiete, 2011*) necessitating error correction by static environmental sensory inputs (*Evans et al., 2016*; *Giocomo, 2016*) such as visual landmark cues (*Campbell et al., 2018*; *Hafting et al., 2005*) or environmental boundaries (*Barry et al., 2007*; *Hardcastle et al., 2015*) to maintain an accurate representation of relative self-location within a spatial environment. Experimental data from studies on rats support the hypothesized role of grid cells in performing self-motion-based path integration and error correction by external sensory cues: Self-motion-based signals such as head direction are necessary for grid cell firing (*Campbell et al., 2018*; *Winter et al., 2015b*), and grid cell firing persists during darkness with reduced spatial selectivity (*Allen et al., 2014*; *Hafting et al., 2005*) reflecting the loss of error correction by visual cues. In contrast, two more recent studies on mice reported a rapid and nearly complete loss of spatially

periodic grid cell firing during complete darkness (*Chen et al., 2016*; *Pérez-Escobar et al., 2016*) suggesting that spatially periodic grid cell firing and path integration depend on error correction by visual cues. However, motion-related interoceptive representations, such as proprioceptive, vestibular, and motor efference copy signals, can combine with visual representations to form a single multimodal representation supporting path integration (*Tcheang et al., 2011*) suggesting that external sensory cues other than visual cues may be sufficient to stabilize grid cells and support path integration. In this study, we therefore investigated whether and how the selective removal of visual inputs affects grid cell firing in mice under conditions which leave other landmark features such as walls, auditory, and olfactory cues still available to the animal.

Errors in path integration originate primarily in the velocity signal (*Stangl et al., 2020*), which combines information about movement direction and movement speed. However, the source of both the directional signal and the speed signal are still under debate. The direction signal has been attributed to the head direction signal terminating in the MEC (*Sargolini et al., 2006*; *Taube et al., 1990a*; *Taube et al., 1990b*; *Taube, 2007*), but head direction does not always match movement direction (*Raudies et al., 2015*). Likewise, the dominant source of the speed signal is still under debate. One potential candidate for the speed signal is the linear modulation of firing rates in a subpopulation of 'speed cells' in the MEC (*Kropff et al., 2015*). However, such a speed signal by firing rate has been shown to be more heterogeneous than previously assumed (*Hardcastle et al., 2017*) and not accurate on sub-second-long time scales (*Dannenberg et al., 2019*) as would be required for updating grid cell firing in real time. An alternative or complementary oscillatory speed signal is provided by the positive correlation between the theta rhythmic firing frequency of theta modulated MEC neurons and the running speed of an animal and/or the positive correlation between the local field potential (LFP) theta frequency and the running speed of an animal (*Hinman et al., 2016*; *Jeewajee et al., 2008a*; *McFarland et al., 1975*; *Rivas et al., 1996*; *Shin and Talnov, 2001*; *Sławińska and Kasicki, 1998*; *Whishaw and Vanderwolf, 1973*). The slope of the LFP theta frequency vs. running speed relationship in particular has been linked to spatial cognition and locomotion in both hippocampus and MEC (*Korotkova et al., 2018*; *Monaghan et al., 2017*; *Wells et al., 2013*). Interestingly, environmental novelty selectively reduces the slope of the LFP theta vs. running speed relationship (*Jeewajee et al., 2008b*; *Wells et al., 2013*) and also results in an increase in grid scale (*Barry et al., 2012*) as predicted by oscillatory interference models of grid cell firing (*Burgess, 2008*). In contrast, the y-intercept of the LFP theta vs. running speed relationship has been linked to emotion-sensitivity, in particular anxiolysis (*Korotkova et al., 2018*; *Monaghan et al., 2017*; *Wells et al., 2013*). However, little is known about the time courses of changes in coding of location by grid cells and coding of running speed by theta oscillations or firing rate (for instance during changes from light to darkness) although changes in grid cell firing, theta rhythmic dynamics, and speed modulation of firing rates may underlie many cognitive functions on a behavioral time scale ranging from seconds to minutes. By examining the temporal correlations between grid cell spatial stability and the two proposed speed signals—by firing rate or theta oscillations—we test if observed changes in grid cell firing can be explained by changes in the speed signal by firing rate or changes in the speed signal by LFP theta frequency. We further test if changes in the representation of a spatial metric by grid cells in the MEC is accompanied by similar changes in the tuning of head direction cells, border cells, or both. In addition, data presented in this study characterize the visual input-evoked changes in neural dynamics in the MEC. These data are relevant for models of path integration and for our understanding of how visual inputs modulate the coding of location and running speed in the entorhinal cortex.

## Results

### The dynamic range of speed coding by local field potential theta frequency decreases in darkness

To study changes in neural dynamics after transitioning from foraging in an illuminated environment (light) to foraging in complete darkness (dark) and vice versa, we recorded single unit and LFP activity in mice freely exploring an open-field environment during alternating light and dark epochs (*Figure 1A*). We first studied if the absence of visual inputs during darkness changed the relationship between LFP theta frequency and running speed. We found a large change in the slope and a

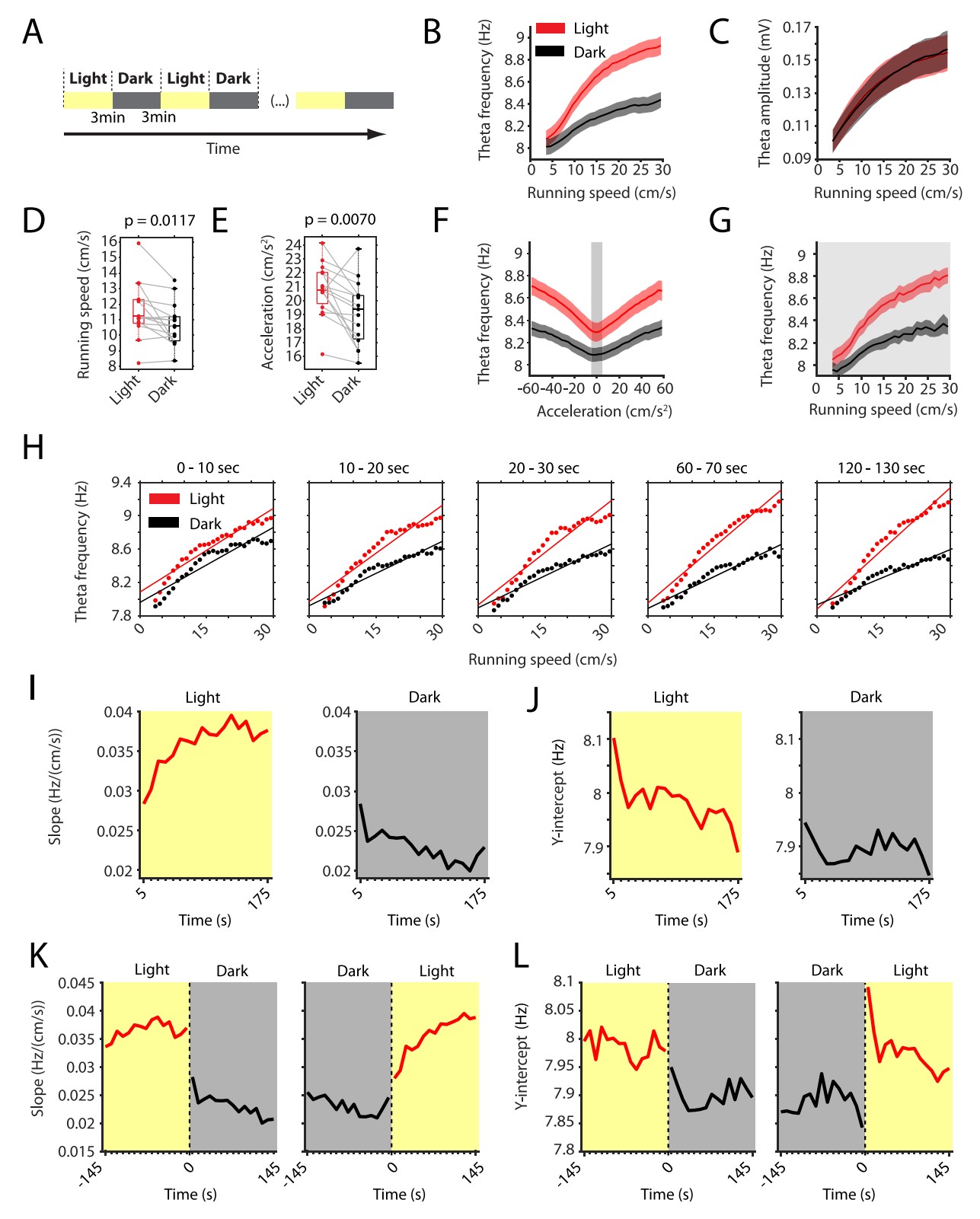

**Figure 1.** The slope of the local field potential theta frequency to running speed relationship decreases in darkness. (A) Experimental paradigm. Local field potential (LFP) theta oscillatory activity was recorded in mice freely foraging in a 1 m square open-field environment containing a visual cue card during alternating epochs of illumination by a ceiling light (light) and complete darkness (dark). A typical recording lasted 42 min with 3 min alternating light and dark epochs. (B) Plot of theta frequency vs. running speed shows that the slope of the relationship decreases during darkness. Solid lines

*Figure 1 continued on next page*

*Figure 1 continued*

show the mean, shaded area show the s.e.m. for n = 15 mice. Data on the light condition are shown in red, data on the dark condition are shown in black and gray. (C) Same as in B, but for theta amplitude. (D) Running speed decreases slightly during darkness. Each data point shows the mean value for one mouse, box plots show mean, 25$^{th}$ and 75$^{th}$ percentiles, gray lines connect data from the same mouse; t = 2,90, df = 14, p=0.0117, Student's paired t-test. (E) Running acceleration decreases slightly during darkness. Data visualization as in D; t = 3.16, df = 14, p=0.0070, Student's paired t-test. (F) LFP theta frequency vs. running acceleration relationship is reduced for all acceleration values in complete darkness. Gray background highlights the interval of acceleration values between −5 cm/s$^2$ and 5 cm/s$^2$. (G) LFP theta frequency vs. running speed relationship based on the subset of time points where the running acceleration is near zero in the interval shown in F between −5 cm/s$^2$ and 5 cm/s$^2$. (H–L) Each data point shows the mean across all epochs of the same condition (light or dark) from 78 recording sessions from 14 mice. (H) LFP theta-frequency speed tuning curves for sequential 10 s blocks after the start of light and dark conditions. Straight lines show the results of linear regression. (I–L) Yellow background indicates the light condition; gray background indicates the dark condition. 95% confidence intervals are within the line width. (I, J) Slopes (I) and y-intercepts (J) of the theta frequency vs. running speed relationships for sequential 10 s time bins for the first three minutes in light and dark. (K, L) Left and right panels show 10 s binned slopes (K) and y-intercepts of the theta frequency vs. running speed relationship around the transition points from light to dark and from dark to light, respectively.

The online version of this article includes the following figure supplement(s) for figure 1:

**Figure supplement 1.** Slow c anges in the local field potential theta frequency to running speed relationship as a function of visual inputs can be observed in a single animal and do not correlate with changes in running speed.

smaller change in the y-intercept of the LFP theta frequency vs. running speed relationship during the dark condition (t-statistic of interaction effect (slope): 18.045, df = 806, p=2×10$^{−61}$; t-statistic of change in y-intercept: −3.676, df = 806, p=0.0003; n = 15 mice) (*Figure 1B*). In contrast, the theta amplitude vs. running speed relationship did not change significantly (t-statistic of interaction effect (slope): 1.7343, df = 806, p=0.0832; t-statistic of change in y-intercept: −1.764, p=0.0781; n = 15 mice) (*Figure 1C*). The average running speed during light, 11.6 ± 0.5 cm/s, slightly decreased during darkness by 0.89 ± 0.31 cm/s (7.7%) to 10.72 ± 1.4 cm/s (mean ± s.e.m., p=0.0117, n = 15 mice, Wilcoxon's signed-rank test) (*Figure 1D* and *Figure 1—figure supplement 1A*). Similarly, running acceleration also slightly decreased from 20.8 ± 1.9 cm/s$^2$ in the light condition by 1.7 ± 0.7 cm/s$^2$ (8.0%) to 19.1 ± 2.3 cm/s$^2$ in the dark condition (p=0.0070, n = 15 mice, Wilcoxon's signed-rank test) (*Figure 1E*). We therefore examined if the observed decrease in the slope of the LFP theta frequency vs. running speed relationship is independent from the decrease in running *acceleration* during darkness. If truly independent, the y-intercept of the correlation between LFP theta frequency and running *acceleration* is expected to decrease during the dark condition. In fact, the y-intercept of the LFP theta frequency vs. running *acceleration* relationship was significantly lower during darkness (t-statistic: 8.115, p=1.16×10$^{−6}$, n = 15 mice, Student's paired t-test) (*Figure 1F*). Furthermore, restricting our analysis of the LFP theta frequency to running *speed* analysis to time points when running *acceleration* was near zero (0 ± 5 cm/s$^2$) did not significantly change speed tuning curves of LFP theta frequency (*Figure 1G*). These data demonstrate that the LFP theta frequency is directly modulated by running speed and that the observed change in the slope of the LFP theta frequency vs. running speed relationship in darkness reflects a change in visual inputs.

The LFP theta frequency vs. running speed relationship has been hypothesized to serve as a speed signal important for coding of location during self-motion-based navigation including path integration. Having established that the removal of visual inputs changes the speed signal by LFP theta frequency, we further examined the time course of that change by analyzing sequential 10 s blocks after the start of light and dark epochs. Averaging data from 78 sessions from 14 mice, we found that the LFP theta frequency vs. running speed relationships in light and dark became more different with elapsed time after start of light and dark epochs at a time scale of tens of seconds (*Figure 1H* and *Figure 1—figure supplement 1B*). This divergence appeared to be largely driven by a change in the slope of the theta frequency vs. running speed relationship as opposed to changes in the y-intercept. We therefore quantified the time courses of those changes in slope and y-intercept during light and dark conditions. This analysis revealed a slow increase in slope over time during the light condition (Pearson's R = 0.79, p=0.0001) and a slow decrease in slope over time during the dark condition (Pearson's R = −0.79, p=0.0001) (*Figure 1I*). The increase in slope during the light condition was accompanied by a small, yet significant, decrease in the y-intercept (Pearson's R = −0.77, p=0.0002) (*Figure 1J*). In contrast, the y-intercept did not change significantly over time during the dark condition (Pearson's R = −0.16, p=0.5251) (*Figure 1J*). In addition to those

slow components of change, analyzing the transition points between light and dark conditions revealed fast (<10 s) light-to-dark decreases and dark-to-light increases in the slope (*Figure 1K*) and y-intercept (*Figure 1L*) of the LFP theta frequency to running speed relationship.

## Changes in theta rhythmic firing as a function of running speed depend on visual inputs

Local field potential activity in the MEC primarily reflects the summed activity of synaptic inputs to the MEC, which is likely to be correlated with firing patterns of MEC neurons. Many neurons in the MEC show a theta rhythmic firing pattern and—similarly to the LFP theta frequency—the frequency of theta rhythmic firing increases with running speed (*Hinman et al., 2016*). We asked if and how theta rhythmic firing in theta modulated MEC neurons (n = 342 neurons from 14 mice) is affected by removing and reinstating all visual inputs. Towards that aim we used an MLE approach (*Climer et al., 2015*) to fit a model of theta spiking rhythmicity to the observed spike train autocorrelations and identified the frequency and magnitude of theta rhythmic firing in theta modulated MEC neurons in light and dark conditions (see *Figure 2A* for one example neuron). 339 out of 342 neurons could be analyzed using this model. We observed a decrease in theta rhythmic firing frequency during the dark condition (*Figure 2B* and *Figure 2—figure supplement 7*). Similar to the observed changes in the LFP theta frequency, the decrease in theta rhythmic firing frequency was largely attributed to a decrease in the slope of its relationship to running speed (*Figure 2C*) as previously shown in rats (*Hinman et al., 2016*).

However, averaging data across cells can hide differences between cells and important characteristics of the population response. We therefore analyzed changes in theta rhythmic firing on the single cell and population level as a function of binned running speed (immobility, defined as running speed <3 cm/s; 5–10 cm/s; 10–15 cm/s; 15–20 cm/s; 20–25 cm/s) (n = 339 neurons, *Figure 2D*). To our surprise, we found that neurons continued to fire rhythmically during immobility periods in both light and dark conditions. However, theta rhythmic frequencies varied largely across neurons within a wide range of theta frequencies (4–12 Hz) during immobility. Intriguingly, theta rhythmic firing synchronized across neurons around a narrower theta-frequency range of 6–10 Hz when the mouse started running. Importantly, running speed-dependent theta dynamics differed between light and dark conditions (*Figure 2D–G*). In order to quantify the changes in theta rhythmic firing dynamics as a function of running speed, we analyzed theta rhythmic firing separately for each running speed bin. In the light condition, the number of significantly theta rhythmic neurons increased sharply from 181 during immobility to 287 at the onset of running. Interestingly, this number then steadily decreased to 242, 215, and 186 for higher running speeds. In contrast, the number of significantly theta rhythmic neurons sharply increased at the onset of running and then saturated close to the maximal number during darkness (157 during immobility, 322, 326, and 328 out of 339 neurons at higher running speeds) (*Figure 2E*). A linear regression model for running speeds in the range from 5 cm/s to 25 cm/s revealed a significant difference in the number of theta rhythmic neurons as function of running speed between light and dark conditions (p=0.0003). These data demonstrate that the ensemble of theta rhythmic neurons decreased with running speed in the light condition but increased to ceiling during the dark condition (*Figure 2E*) contributing to the stronger synchrony of theta rhythmic firing observed on the population level (*Figure 2D*). These results were consistent across different significance levels between alpha = 0.001 and alpha = 0.1 for classification of theta rhythmic neurons (*Figure 2—figure supplement 1*). We next identified the cells which remained theta rhythmic across all running speed bins and quantified changes in the frequency and magnitude of theta rhythmic firing as a function of running speed during light and dark conditions. The frequency of theta rhythmic firing increased with running speed during the light condition (p=$4.2 \times 10^{-8}$, linear mixed effects model), but not during the dark condition (p=0.8085, linear mixed effects model; significant difference in slopes between light and dark, p=$6.8 \times 10^{-6}$) (*Figure 2F*). Similarly, the magnitude of theta rhythmic firing increased with running speed during the light condition (p=0.0002, linear mixed effects model), but not during the dark condition (p=0.0845, linear mixed effects model; difference in slopes between light and dark, p=$9.1 \times 10^{-6}$) (*Figure 2G*). Similar results were obtained when different significance levels were used to identify theta rhythmic neurons (*Figure 2—figure supplement 1*). Notably, the amplitudes of the average autocorrelogram models across all cells were very similar between light and dark conditions (*Figure 2D*, third row) despite the observed differences in the magnitudes of theta rhythmic firing on the single cell level

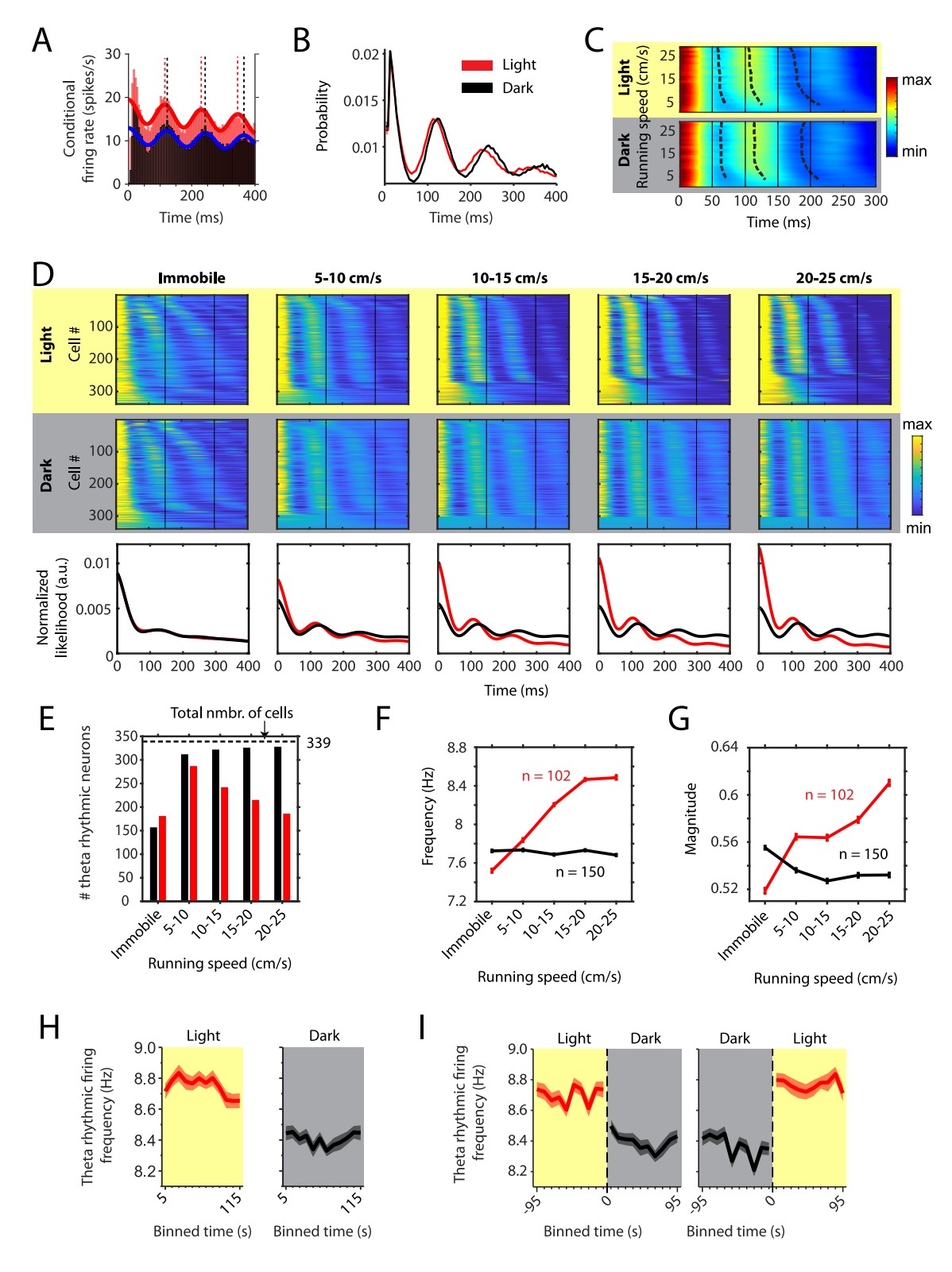

**Figure 2.** Changes in theta rhythmic firing as a function of running speed differ between light and dark conditions. Red color or yellow background indicates data on the light condition, black and gray color or gray background indicates data on the dark condition. Figures show data from theta rhythmic neurons from a total of 14 mice. (**A**) Autocorrelogram of one example neuron showing theta rhythmic firing in light and dark. Solid red and blue lines show the results of the maximum likelihood estimate (MLE) model fit. Vertical dashed lines indicate the positions of the peaks in the

*Figure 2 continued on next page*

*Figure 2 continued*

autocorrelogram. (B) Mean probability-normalized autocorrelograms for light and dark conditions from n = 342 neurons. s.e.m. is within line width. (C) Figure shows color-coded mean autocorrelogram fits across n = 340 neurons per running speed bin. Speed bins range from 1 ± 4 cm/s to 29 ± 4 cm/s. Dashed lines connect the first peaks in the mean autocorrelogram models. (D) Upper panels show the color coded MLE autocorrelogram models for each of the n = 339 neurons at different running speed intervals. Immobility was defined as running speed <3 cm/s. Neurons are sorted per condition in order of their theta rhythmic firing frequency. Lower row shows the average across autocorrelogram models per condition. Note that higher synchrony across neurons results in higher rhythmicity magnitude in the average across neurons. (E) Number of neurons shown in D with statistically significant spiking rhythmicity at alpha = 0.1 determined by the MLE model for different running speeds in light and dark. (F, G) Longitudinal analysis of frequency (F) and magnitude (G) of theta rhythmic firing of neurons with significant theta rhythmic firing across all running speed bins (light: n = 102 neurons; dark: n = 150 neurons). Data show mean ± s.e.m. (H) Theta rhythmic firing frequency plotted for consecutive 10 s time bins after the start of light and dark conditions. (I) Theta rhythmic firing frequency plotted for 10 s time bins around the transition points from light to dark and from dark to light. Spike time autocorrelograms of theta rhythmic neurons are provided in *Figure 2—source data 1*.

The online version of this article includes the following source data and figure supplement(s) for figure 2:

**Source data 1.** Spike time autocorrelograms of the 342 theta rhythmic neurons analyzed for data shown in *Figure 2* during light and dark conditions.
**Figure supplement 1.** Observed changes in theta rhythmic firing as a function of visual inputs are consistent across a wide range of significance levels chosen to identify theta rhythmic neurons.
**Figure supplement 2.** Theta rhythmic firing as a function of running speed and visual inputs in putative principal neurons.
**Figure supplement 3.** Theta rhythmic firing as a function of running speed and visual inputs in putative interneurons.
**Figure supplement 4.** Changes in theta rhythmic firing as a function of running speed and visual inputs in theta rhythmic grid cells.
**Figure supplement 5.** Changes in theta rhythmic firing as a function of running speed and visual inputs in theta rhythmic head direction cells.
**Figure supplement 6.** Changes in theta rhythmic firing as a function of running speed and visual inputs in theta rhythmic border cells.
**Figure supplement 7.** Spike time autocorrelograms of randomly picked 24 theta rhythmic neurons showing theta rhythmic spiking during the light and dark condition.

(*Figure 2G*). These data suggest that—during darkness— higher synchrony across MEC neurons on the population level counterbalances the lower magnitude of theta rhythmicity on the single cell level which would explain that we did not observe a change in LFP theta amplitude in the dark condition (*Figure 1C*).

Population characteristics of running speed-dependent theta rhythmic firing and their changes with removal of visual inputs may differ between principal neurons and interneurons. We therefore classified all theta rhythmic neurons in our data set (n = 342) into putative principal neurons (PNs, n = 273) and putative interneurons (INs, n = 69) based on the spike width and firing rate of these neurons (*Figure 2—figure supplement 2E*). In general, both the frequency and magnitude of theta rhythmic firing was higher in PNs compared to INs (*Figure 2—figure supplements 2C* and *3C*). In the light condition, the frequency of theta rhythmic firing showed a similar increase as a function of running speed for both PNs and INs but was consistently higher in PNs compared to INs. In contrast, the magnitude of theta rhythmic firing increased as a function of running speed in INs but not in PNs, which showed high theta rhythmic magnitudes at all running speeds. In the dark condition, frequencies of theta rhythmic firing differed less between PNs and INs compared to the light condition but showed an opposite change as a function of running speed with values slightly increasing in INs but slightly decreasing in PNs. In contrast, the magnitude of theta rhythmic firing was significantly higher in PNs compared to INs at all running speeds. Statistical results on changes in the frequency and magnitude of theta rhythmic firing within and between light and dark conditions for PNs and INs are reported in *Tables 1* and *2*. Statistical results on differences between PNs and INs with respect to changes in theta rhythmic firing as a function of running speed during light and dark conditions are reported in *Tables 3* and *4*. We next addressed the question whether the observed changes in theta rhythmic spiking activity in PNs was different for different functional cell types such as grid cells, or alternatively was similar across PNs. We found that changes in theta rhythmic spiking activity as a function of running speed during and between light and dark conditions were similar across grid cells, head direction cells, and border cells (*Figure 2—figure supplements 4*, *5* and *6*) and indistinguishable from all PNs, indicating that changes in theta rhythmic activity is ubiquitous across different functional cell types in the medial entorhinal cortex.

To study the time course of changes in theta rhythmic firing, we next analyzed theta rhythmic firing in sequential 10 s blocks after the start of light and dark epochs (*Figure 2H*). The frequency of theta rhythmic firing was significantly reduced in the dark condition (effect size: –0.381 Hz, t-statistic: –8.598, df: 4316, p=1×$10^{-17}$, CI: [–0.468 –0.294]). Similar to the fast initial changes observed in the

**Table 1.** related to *Figure 2*.
Statistics on the changes in the frequency of theta rhythmic firing as a function of running speed for principal cells and interneurons during light and dark conditions. Results from a linear mixed effects model. PN = principal neurons, IN = interneurons, SE = standard error, df = degrees of freedom, CI = confidence intervals.

| | | Estimate | SE | t-statistic | df | p-value | CI | |
|---|---|---|---|---|---|---|---|---|
| PN | Light | 0.0463 | 0.0126 | 3.69 | 303 | 0.0003 | 0.0216 | 0.0710 |
| | Dark | −0.0133 | 0.0091 | −1.47 | 503 | 0.1424 | −0.0312 | 0.0045 |
| | Light−Dark | −0.0597 | 0.0152 | −3.92 | 806 | 0.0001 | −0.0895 | −0.0298 |
| IN | Light | 0.0583 | 0.0132 | 4.43 | 203 | 1.55E-05 | 0.0324 | 0.0843 |
| | Dark | 0.0220 | 0.0124 | 1.77 | 243 | 0.0774 | −0.0024 | 0.0465 |
| | Light−Dark | −0.0363 | 0.0181 | −2.00 | 446 | 0.0461 | −0.0720 | −0.0006 |

slope of the LFP theta frequency vs. running speed relationship, we observed a fast decrease in the frequency of theta rhythmic firing when transitioning from light to dark and a fast increase when transitioning from dark to light (*Figure 2I*). In contrast to the observed changes in the slope of the LFP theta frequency vs. running speed relationship, the frequency of theta rhythmic firing did not further change on a time scale of tens of seconds during light or dark epochs. However, differentiating between PNs and INs revealed a decrease in frequency of theta rhythmic firing as a function of time in both light and dark conditions in INs but not PNs (*Figure 2—figure supplements 2F* and *3E*).

## Theta-frequency phase locking of neuronal spiking increases during darkness

Theta-frequency phase locking of single neuron spiking is a prominent feature of the hippocampal and entorhinal network (*Klausberger et al., 2003*; *Mizuseki et al., 2009*). Increased phase locking has been associated with dynamical switching between encoding and retrieval dynamics (*Hasselmo et al., 2002*; *Siegle and Wilson, 2014*), memory performance (*Liebe et al., 2012*; *Rutishauser et al., 2010*; *Schomburg et al., 2014*), and spatial cognitive demand (*Barry et al., 2016*). We therefore asked if the larger synchronization across neurons observed in the dark condition was associated with an increase of theta-frequency phase locking of theta rhythmic spiking activity. To compare phase locking of firing between light and dark conditions, we computed the mean resultant length (MRL) of theta-phase firing rate maps for the light and dark condition (*Figure 3A*). Interestingly, phase locking of theta rhythmic neurons increased during the dark condition (median MRL (Light) = 0.29, median MRL (Dark) = 0.37, p=$1\times10^{-32}$, Wilcoxon signed-rank test; *Figure 3B*). Such an increase is consistent with the increased numbers of theta rhythmic neurons during the dark condition as shown in *Figure 2E*. No significant differences were observed between PNs and INs (*Figure 3—figure supplement 1*). Notably, each neuron's preferred phase of firing remained

**Table 2.** related to *Figure 2*.
Statistics on the changes in the magnitude of theta rhythmic firing as a function of running speed for principal cells and interneurons during light and dark conditions. Results from a linear mixed effects model. PN = principal neurons, IN = interneurons, SE = standard error, df = degrees of freedom, CI = confidence intervals.

| | | Estimate | SE | t-statistic | df | p-value | CI | |
|---|---|---|---|---|---|---|---|---|
| PN | Light | 0.0013 | 0.0014 | 0.9084 | 303 | 0.3644 | −0.0015 | 0.0041 |
| | Dark | −0.0012 | 0.0008 | −1.46 | 503 | 0.1441 | −0.0027 | 0.0004 |
| | Light−Dark | −0.0025 | 0.0015 | −1.63 | 806 | 0.1036 | −0.0054 | 0.0005 |
| IN | Light | 0.0079 | 0.0014 | 5.70 | 203 | 4.17E-08 | 0.0051 | 0.0106 |
| | Dark | −0.001 | 0.0008 | −0.95 | 243 | 0.3424 | −0.0022 | 0.0008 |
| | Light−Dark | −0.0086 | 0.0015 | −5.69 | 446 | 2.33E-08 | −0.0115 | −0.0056 |

**Table 3.** related to *Figure 2*.
Statistics on the differences in frequency of theta rhythmic firing between putative principal neurons and interneurons. Results from a linear mixed effects model. Values are referenced to putative principal neurons. SE = standard error, df = degrees of freedom, CI = confidence intervals.

| | | Estimate | SE | t-statistic | df | p-value | CI | |
|---|---|---|---|---|---|---|---|---|
| Light | y-intercept | −1.34 | 0.36 | −3.73 | 506 | 0.0002 | −2.05 | −0.64 |
| | slope | 0.0117 | 0.0169 | 0.69 | 506 | 0.4921 | −0.0216 | 0.0450 |
| Dark | y-intercept | −0.48 | 0.33 | −1.46 | 746 | 0.1454 | −1.12 | 0.17 |
| | slope | 0.0327 | 0.0142 | 2.31 | 746 | 0.0212 | 0.0049 | 0.0605 |

unchanged between light and dark conditions (*Figure 3C & D*). We next investigated the time course of those changes in phase locking (*Figure 3E*). The time course of changes in theta-frequency phase locking was similar to the changes in the frequency of theta rhythmic spiking, showing a fast initial change but no further slow component of change. No differences were observed between PNs and INs.

## Visual inputs alter mean firing rates of neurons in the medial entorhinal cortex

Motivated by the differences in LFP activity during light and dark conditions, we asked if these changes were accompanied by changes in firing rates of MEC neurons. We therefore first asked whether and how the removal and reinstating of visual inputs affected firing rates in MEC neurons. Consistent with a previous study (*Pérez-Escobar et al., 2016*), we found striking differences in mean firing rates between light and dark epochs among a wide variety of MEC neurons including speed cells, head direction cells, grid cells, border cells and interneurons (*Figure 4A & B*; *Figure 4—figure supplement 1*). In general, changes in the mean firing rates of MEC neurons could be positively or negatively correlated to visual inputs. Positive modulation by visual inputs was observed ~4 times more frequently than negative modulation (139 out of 842 or 16.5% positively modulated neurons; 32 out of 842 or 3.8% negatively modulated neurons at significance level alpha = 0.05; data from 15 mice). In both cases, changes in firing rate after transitions between light and dark conditions occurred in less than 5 s (*Figure 4C & D*). Since we observed slight but significant changes in the animals' mean running speeds during darkness and many neurons' firing rates are modulated by running speed, we first tested whether the observed light vs. dark changes in mean firing rates of neurons are merely a consequence of the interaction between speed modulation of firing rates and changes in the animal's running speed. Towards that goal, we computed the dark-to-light ratios of changes in mean firing rates for negatively (0.62 ± 0.16) and positively (1.53 ± 0.28) modulated neurons, respectively, and compared these observed ratios with running speed-adjusted dark-to-light ratios computed from the areas under the speed tuning curves in light and dark conditions (*Figure 4—figure supplement 2*). After running speed adjustment, dark-to-light ratios were 0.61 ± 0.17 (mean adjustment = −0.014, CI = [−0.002 −0.026], t = −2.31, df = 138, p=0.023, t-test) and 1.36 ± 0.23 (mean adjustment = −0.174, CI = [−0.112 −0.235], t = −5.78, df = 31, p=$2.3 \times 10^{-6}$) for

**Table 4.** related to *Figure 2*.
Statistics on differences in magnitude of theta rhythmic firing between putative principal neurons and interneurons. Results from a linear mixed effects model. Values are referenced to putative principal neurons. SE = standard error, df = degrees of freedom, CI = confidence intervals.

| | | Estimate | SE | t-statistic | df | p-value | CI | |
|---|---|---|---|---|---|---|---|---|
| Light | y-intercept | −0.20 | 0.05 | −4.23 | 506 | 2.8E-05 | −0.30 | −0.11 |
| | slope | 0.0060 | 0.0019 | 3.17 | 506 | 0.0016 | 0.0023 | 0.0097 |
| Dark | y-intercept | −0.1903 | 0.0472 | −4.03 | 746 | 6.10E-05 | −0.28 | −0.10 |
| | slope | 0.0004 | 0.0011 | 0.36 | 746 | 0.7163 | −0.0018 | 0.0026 |

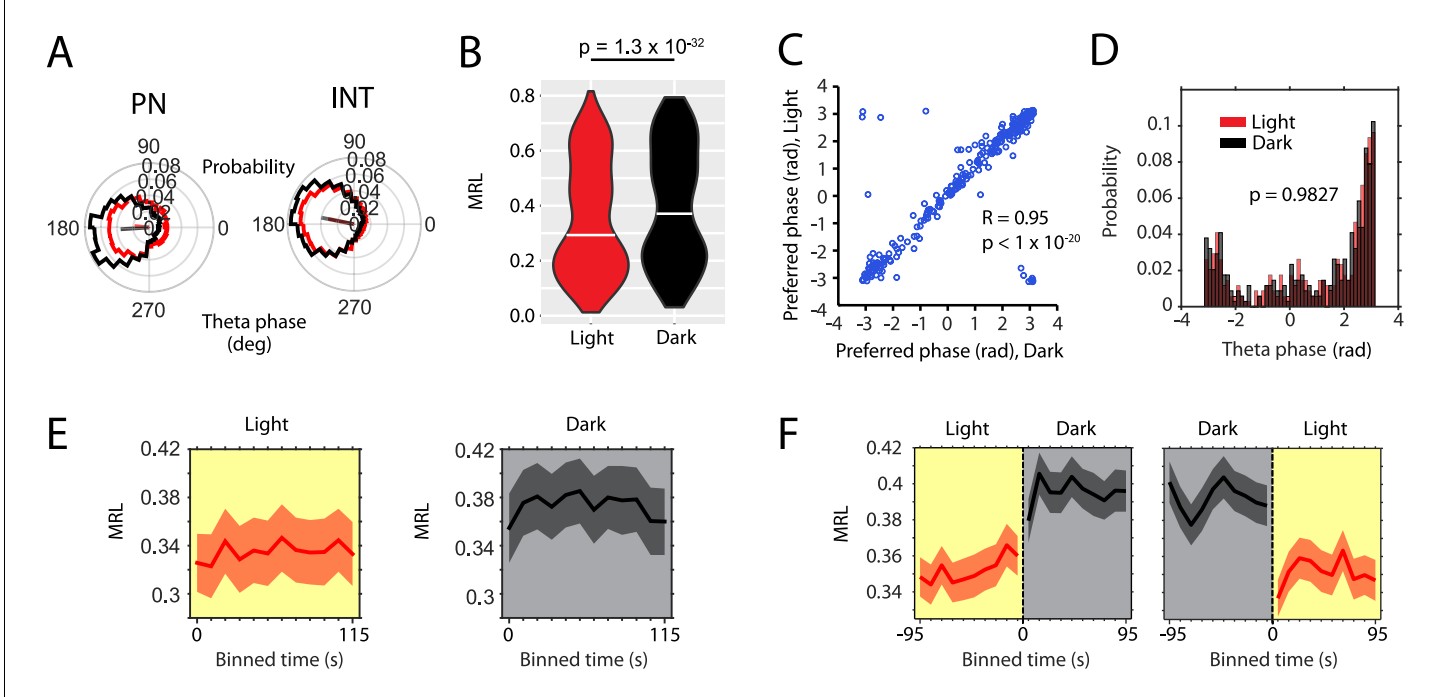

**Figure 3.** Absence of visual inputs increases LFP theta-frequency phase locking of single neuron spiking. (**A**) Probability-normalized polar histograms of LFP theta-phase-binned spiking activity for two example neurons (PN = putative principal neuron, IN = putative interneuron). Straight lines show the mean resultant vector scaled between 0 (center of polar histogram) and 1 (outer edge of polar histogram). (**B**) Violin plots of theta-phase locking of n = 342 theta rhythmic neurons during light and dark, p=1.3×10$^{-32}$, Wilcoxon signed-rank test. (**C**) Scatter plot of preferred LFP theta phases during light and dark. Note the high circular-linear correlation. (**D**) Probability-normalized histograms of preferred LFP theta phases during light and dark. Note that the majority of neurons preferred firing around the trough of pyramidal layer LFP theta oscillations and a minority preferred firing around the peak. No difference was observed between light and dark conditions, p=0.9827, Kolmogorov-Smirnov test. (**E**) Longitudinal analysis of the time course of the mean resultant length (MRL) of LFP theta-frequency phase-locking plotted in 10 s time bins after the start of light or dark conditions, n = 55 neurons. (**F**) Longitudinal analysis of MRL plotted for 10 s time bins around the transition points from light to dark and from dark to light, n = 58 cells. Probability-normalized theta-phase tuning curves of principal neurons and interneurons are provided in *Figure 3—source datas 1* and *2*.

The online version of this article includes the following source data and figure supplement(s) for figure 3:

**Source data 1.** Probability-normalized polar histograms showing spiking activity as a function of layer II/III theta phase for each of the 273 theta rhythmic principal neurons during the light (red) and dark (black) conditions including the mean resultant vector scaled between 0 and 1.

**Source data 2.** Probability-normalized polar histograms showing spiking activity as a function of layer II/III theta phase for each of the 69 theta rhythmic interneurons during the light (red) and dark (black) conditions including the mean resultant vector scaled between 0 and 1.

**Figure supplement 1.** Data for n = 273 putative principal neurons (PN) (**A**) and n = 69 putative interneurons (IN) (**B**).

negatively and positively modulated neurons, respectively. Notably, even after adjustment for running speed, the dark-to-light effect sizes were very similar for negatively and positively modulated neurons, namely a 39% decrease and a 36% increase in firing rates, respectively. These data demonstrate that the observed dark versus light changes in mean firing rates are not an artifact of the interaction between speed modulation of firing rates and changes in the animals' running speed.

One possible explanation for the changes in mean firing rates during darkness can be a binary change in the available spatial information gained from visual cues. Alternatively, changes in firing rates could be driven by changes in net excitatory drive from areas in the visual cortex due to changes in luminance and retinal stimulation independent of any gain or loss of spatial information. To distinguish between these two alternative hypotheses, we analyzed data from optogenetic control experiments in a subset of two mice, in which mice carried a light fiber attached to their head-stage with the fiber tip approximately 1 cm above the animal's head. Green laser light (561 nm, 1 mW) was emitted from the tip of that light fiber and illuminated the head and immediate surroundings of the animal's head in alternating epochs. Importantly, this experiment was performed with the ceiling lights turned on during the whole recording session with the head-mounted green laser light being turned on and off in alternating epochs during that session. While being visible to the animal

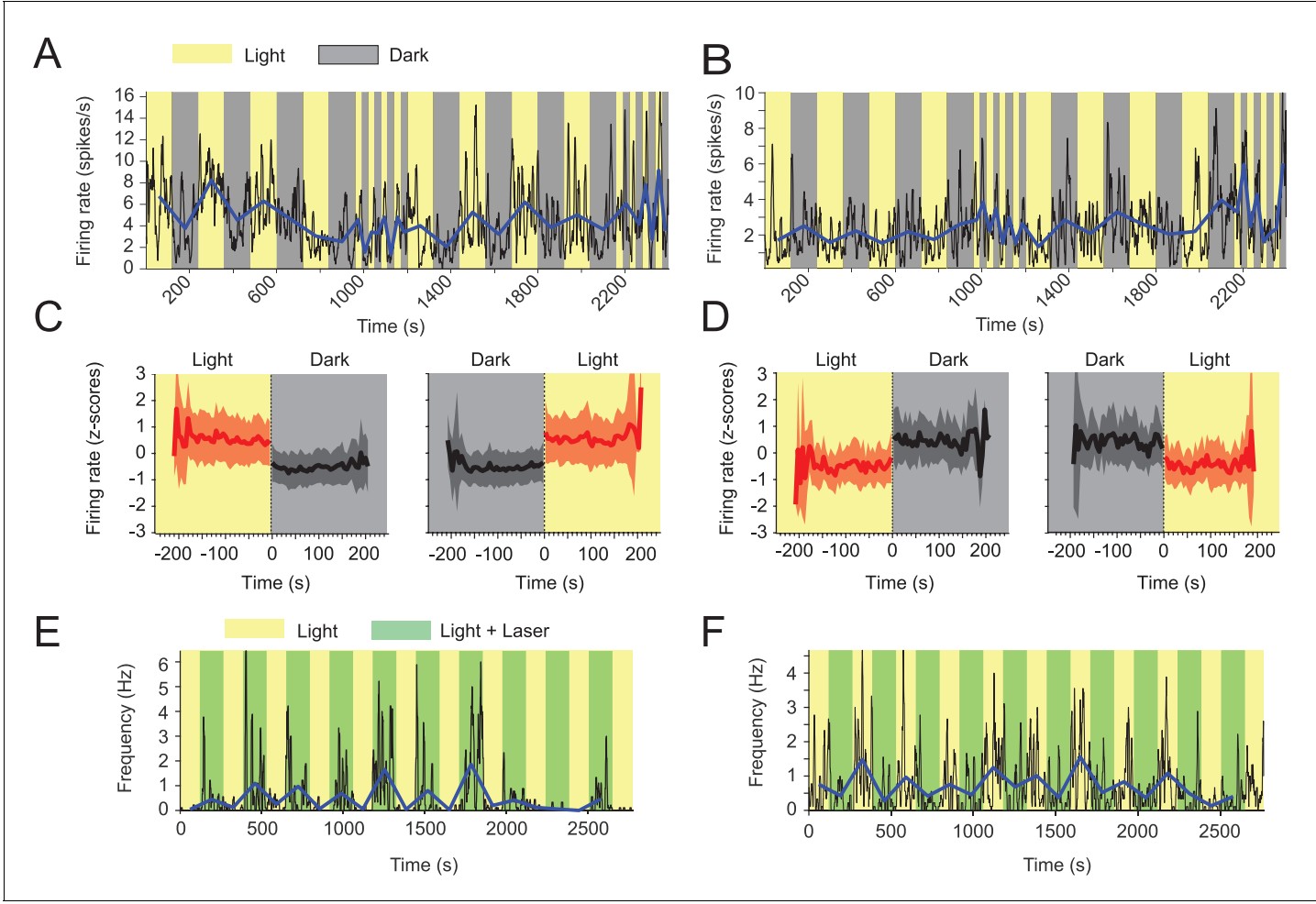

**Figure 4.** Visual inputs alter mean firing rates of neurons in the medial entorhinal cortex. (A-D) Yellow and gray backgrounds indicate duration of light and dark epochs, respectively. (A) Data on firing rate from one example neuron in the medial entorhinal cortex (MEC) with its baseline firing rate negatively modulated by darkness. Black lines show the firing rate at 1 s resolution. Blue line connects the mean firing rate in each light or dark epoch. (B) Data on firing rate from one example neuron in the MEC whose baseline firing rate is positively modulated by darkness; same visualization scheme as in A. (C) Left and right panels show the mean z-scored firing rate across n = 139 neurons with significantly negative modulation of firing rates by darkness (see Materials and methods) in 5 s bins around the transition point from light to dark and dark to light, respectively. Red color indicates data from the light condition, black and gray color indicate data from the dark condition. Solid lines and shaded areas show mean and SD. (D) Same as C, but for neurons with positive modulation of baseline firing rates by darkness; n = 32. (E, F) Yellow background indicates light condition, green background indicates the presence of an egocentrically stable spot of green laser light above the animal's head during the standard light condition. (E) Data on one example neuron in the MEC showing positive modulation of baseline firing rate in presence of the green laser light spot. Same visualization scheme as in A and C. (F) As in E, but for one example neuron with negative modulation of baseline firing rate. Mean firing rates and running speed-adjusted mean firing rates of neurons during light and dark conditions are provided in *Figure 4—source data 1*.

The online version of this article includes the following source data and figure supplement(s) for figure 4:

**Source data 1.** Data table of mean firing rates and running speed-adjusted mean firing rates during light and dark conditions of the 171 neurons with significant modulation of firing rates by visual inputs.

**Figure supplement 1.** Examples of functional cell types in the medial entorhinal cortex whose baseline firing rates are modulated by visual inputs.

**Figure supplement 2.** Observed changes in mean firing rates during darkness are not an artifact of changes in running speed.

as an egocentrically stationary spot of light, the laser light was too weak to illuminate the open-field environment and thus did not change the visibility of distal visual cues useful for navigational purposes (e.g. walls or the cue card of the arena). Nevertheless, we found strong effects of that kind of visual point stimulation on firing rates of five MEC neurons from two mice during foraging under the light condition (*Figure 4E & F*). Four out of those five neurons showed positive modulation by the egocentrically stationary spot of laser light, and one showed negative modulation. The effects of

presenting an egocentrically stationary visual stimulus during the light condition on baseline firing rates of MEC neurons resembled the effects of transitioning from the dark to the light condition with respect to effect sizes in individual neurons and with respect to the ~4:1 ratio in the number of neurons that are positively and negatively modulated by retinal stimulation. In summary, these data demonstrate that the changes in retinal stimulation due to changes in the amount of spatially non-informative light hitting the retina after transitioning from dark to light and vice versa are sufficient to explain the observed changes in mean firing rates of MEC neurons.

## Changes in mean firing rates of speed cells result in changes in the slopes of speed tuning curves without changing the running speed-dependent gain

We next turned to the analysis of the hypothesized speed signal by firing rate, in particular speed cells in the MEC (*Kropff et al., 2015*), and asked if the speed modulation of firing rates in MEC neurons is changed in complete darkness. The slopes of speed tuning curves have previously been reported to change after the removal of visual inputs and this change has been interpreted as a change in the hypothesized speed signal by speed cells, which has further been indicated to underlie changes in grid cell firing and path integration during darkness (*Chen et al., 2016*; *Pérez-Escobar et al., 2016*). An alternative approach, however, analyzes the running speed-dependent gain in firing rates. When speed modulation is analyzed after normalization *across* conditions, any change in external or internal parameters affecting a neuron's mean firing rate will inevitably also affect the y-intercept and the slope of the neuron's speed tuning curve. In contrast, analyzing speed modulation after normalization *within* conditions allows analyzing the running speed-dependent gain in firing rates independently from changes in a neuron's mean firing rate. In this study, we do not examine which of the two alternative measures of speed modulation–the slope of the speed tuning curve or the running speed-dependent gain in firing rate—is the more useful measure for speed modulation of firing rates. However, we argue in favor of the second approach, analyzing the running speed-dependent gain in firing rate after normalization of speed tuning curves within conditions, on the grounds that it is standard in the field to compare tuning of other parameters such as head directional tuning by head direction cells or spatial tuning by place cells, grid cells, or border cells after normalization within conditions. As shown above, the mean firing rates of many neurons are responsive to changes in visual inputs. In addition, many speed cells show conjunctive coding properties which may be changed in complete darkness (*Hardcastle et al., 2017*; *Hinman et al., 2016*). To our knowledge, previous analyses of changes in firing rate modulation by running speed between two experimental conditions, such as light and dark environments, have not been corrected for possible confounding factors, such as changes in mean firing rate or conjunctive coding properties. In particular, data presented in this study show that changes in visual inputs cause changes in the firing rates of neurons and those changes may be independent from possible additional changes in the firing rate modulation by running speed. The same logic applies for changes in conjunctive coding properties, which may be misinterpreted as changes in the running speed modulation of firing rate if not properly controlled for. To account for possible changes in conjunctive coding properties, such as conjunctive coding for position, head direction, and LFP theta phase, we applied a linear-nonlinear (LN) model (*Hardcastle et al., 2017*) which provides a method for identification of significantly speed-modulated neurons without restricting the analysis to linear modulation of firing rates and computes speed response curves instead of speed tuning curves. Speed response curves are speed tuning curves corrected for the influence of conjunctive coding properties (*Hardcastle et al., 2017*). The firing rates of n = 164 neurons out of 495 cells from 15 mice showed significant modulation by running speed in either the light or the dark condition. Consistent with previous reports, speed modulation of firing rates was heterogeneous (*Figure 5A & B*; *Dannenberg et al., 2019*; *Hardcastle et al., 2017*; *Hinman et al., 2016*). Furthermore, we noticed a distribution of peaks in the speed response curves covering the full range of running speeds (*Figure 5A & B*). Hierarchical clustering of speed cells resulted in two main clusters of 138 positively and 26 negatively modulated neurons (*Figure 5—figure supplement 1*). Consistent with previous results (*Chen et al., 2016*; *Pérez-Escobar et al., 2016*), we observed a significant decrease in the slope of speed tuning curves of positively speed-modulated neurons during the dark condition if speed response curves were normalized to the mean firing rate across conditions (*Figure 5A & C* and *Table 5*). 153 out of the 164 speed-modulated neurons could also be examined for potential co-

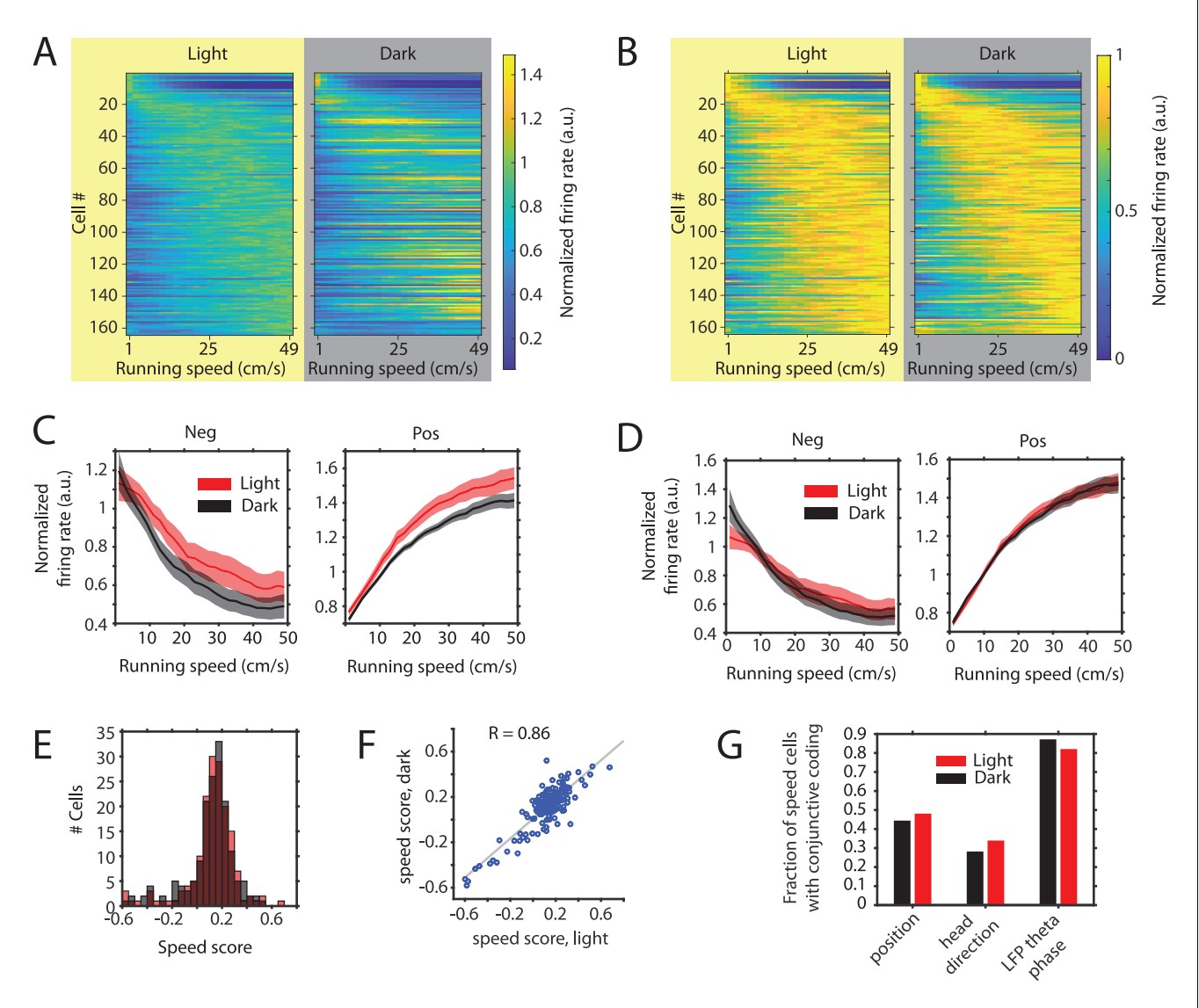

**Figure 5.** Changes in the slope of speed tuning curves during darkness are largely explained by changes in mean firing rate as opposed to changes in the running speed-dependent gain in firing rates. (**A**) Left and right panels show speed response curves for the n = 164 neurons passing statistical significance of speed modulation in the LN-model (see Materials and methods) for either the light or dark condition, respectively. For comparison of speed response curves across neurons and between light and dark conditions, firing rates were normalized *across* conditions to the maximum in the light condition; Cells were sorted by the location of peak firing on the running speed axis averaged across light and dark conditions and each row in the left and right panels corresponds to the same cell. (**B**) Same data as in A, but speed response curves were normalized to their maxima *within* conditions, which allows a comparison of the running speed-dependent gain in firing rates. (**C**) Left and right panels show speed response curves of negatively (n = 26) and positively (n = 138) modulated neurons after normalization to the mean firing rate *across* conditions; solid lines and shaded areas show mean ± s.e.m. across neurons, see *Table 5* on statistics (**D**) Left and right panels show speed response curves of negatively (n = 26) and positively (n = 138) modulated neurons after normalization to the mean firing rate *within* conditions; solid lines and shaded areas show mean ± s.e.m. across neurons; see *Table 6* on statistics (**E**) Histogram of speed scores for all neurons showing significant speed modulation in either the light or dark condition. Red and gray color indicate data on light and dark, respectively. (**F**) Scatter plot comparing speed scores during the light with speed scores during the dark condition. (**G**) Percentage of cells showing conjunctive coding of running speed and either position, head direction, or LFP theta phase during the light and dark conditions. Speed response curves of neurons during light and dark conditions are provided in *Figure 5—source data 1*. The online version of this article includes the following source data and figure supplement(s) for figure 5:

**Source data 1.** Speed response curves during light and dark conditions of the 164 neurons showing significant modulation of firing rates by running speed in either light or dark condition.

**Figure supplement 1.** Clustergram visualizing the results of the hierarchical clustering of neurons by similarity of speed modulation of firing rates.

*Figure 5 continued on next page*

*Figure 5 continued*

**Figure supplement 2.** Figure shows the effect of normalizing LN-model derived speed response curves (*Hardcastle et al., 2017*) and speed tuning curves across or within sessions for ten example cells.

modulation of firing rates by visual inputs. 34 neurons out of those 153 (22.2%) speed cells showed significant modulation of mean firing rate by visual inputs. This proportion is very similar to the proportion of 171 out of 842 (20.3%) neurons with significant modulation of mean firing rates by visual inputs observed in the whole data set. Notably, this is a conservative estimate because the firing rate of many more neurons may be modulated by visual inputs without that modulation being detected as significant with the given statistical power and an alpha level of 0.05. To address the question whether the observed changes in the slope of speed response curves are caused by changes in the running speed-dependent gain in firing rates or are instead artifacts of changes in the mean firing rate of neurons, we compared speed response curves between light and dark conditions after normalization *within* conditions as opposed to normalization *across* conditions. Note that normalization within conditions corrects for changes in mean firing rate between conditions. Interestingly, we found that speed response curves appeared very similar between light and dark conditions after correcting for changes in mean firing rate and conjunctive coding properties (*Figure 5B &D*). In particular, no significant difference was observed between speed responses of positively speed-modulated neurons in the dark condition compared to the light condition (*Table 6*). With respect to negatively speed-modulated neurons, we even observed a small increase in the running speed-dependent gain in firing rate during the dark condition (*Figure 5D* and *Table 6*). Likewise, speed scores (Pearson's correlation coefficient between firing rate and running speed) of all speed-modulated neurons were not significantly changed between light and dark conditions on the population level (p=0.912, n = 164, Kolmogorov-Smirnov test) (*Figure 5E*) and were highly correlated between light and dark conditions on the single cell level (R = 0.86) (*Figure 5F*). Moreover, the numbers of speed-modulated cells with conjunctive coding of position, head direction and theta phase were unchanged in the dark condition (*Figure 5G*). In summary, these data demonstrate that the running speed-dependent gains in firing rate of MEC speed cells remain unaltered during the absence of visual inputs, even if mean firing rates and the slopes of speed tuning curves change substantially.

## Grid cell spatial firing is maintained but less stable during darkness

While in earlier studies grid cells have been reported to show stable grid cell firing during complete darkness in rats and mice (*Allen et al., 2014*; *Hafting et al., 2005*), two more recent studies in mice showed a strong reduction in the hexagonal symmetry of grid cell firing during complete darkness (*Chen et al., 2016*; *Pérez-Escobar et al., 2016*). To test if this discrepancy is due to differences in the experimental settings, the focus in this study was on selectively eliminating visual inputs only, while leaving other environmental and/or sensory cues such as walls or odors and auditory cues potentially present in the testing arena intact. We first tested if grid cell firing can emerge in complete darkness. We therefore introduced mice to the familiar open-field arena during complete darkness (the experimenter was wearing night-vision goggles) and recorded grid cells during the initial dark session and a following light and dark session (10–20 min duration each). We found that spatial

**Table 5.** related to *Figure 5*.

Statistics on the differences in speed response curves of positively and negatively speed-modulated neurons between light and dark conditions after normalization across conditions. Results from a linear mixed effects model with the light condition as reference. Pos = positively modulated, Neg = negatively modulated, SE = standard error, df = degrees of freedom, CI = confidence intervals.

|  |  | Estimate | SE | t-statistic | df | p-value | CI | |
|---|---|---|---|---|---|---|---|---|
| Pos | y-intercept | −0.0527 | 0.017571 | −2.99941 | 6896 | 0.0027 | −0.08715 | −0.01826 |
|  | slope | −0.00176 | 0.000609 | −2.89551 | 6896 | 0.0038 | −0.00296 | −0.00057 |
| Neg | y-intercept | −0.06784 | 0.025989 | −2.61042 | 1296 | 0.0091 | −0.11883 | −0.01686 |
|  | slope | −0.00149 | 0.0009 | −1.65711 | 1296 | 0.0977 | −0.00326 | 0.000274 |

**Table 6.** related to *Figure 5*.

Statistics on the differences in speed response curves of positively and negatively speed-modulated neurons between light and dark conditions after correcting for changes in mean firing rates by normalization within conditions. Results from a linear mixed effects model with the light condition as reference. Pos = positively modulated, Neg = negatively modulated, SE = standard error, df = degrees of freedom, CI = confidence intervals.

|  |  | Estimate | SE | t-statistic | df | p-value | CI | |
|---|---|---|---|---|---|---|---|---|
| Pos | y-intercept | 4.10E-05 | 0.012502 | 0.003278 | 6896 | 0.9974 | −0.02447 | 0.024548 |
|  | slope | −0.00023 | 0.000433 | −0.52947 | 6896 | 0.5965 | −0.00108 | 0.00062 |
| Neg | y-intercept | 0.067964 | 0.025506 | 2.66459 | 1296 | 0.0078 | 0.017926 | 0.118002 |
|  | slope | −0.00349 | 0.000884 | −3.95082 | 1296 | 8.21E-05 | −0.00523 | −0.00176 |

periodicity of grid cell firing was low—but not completely eliminated—in the initial dark session and then strongly increased during the first light session (*Figure 6A & B*). Interestingly, after this initial orientation, spatial periodicity of grid cell firing decreased again after removal of visual inputs, but remained significantly higher than in the initial dark session (Chi-square = 10.89, p=0.0043, n = 9 cells from two mice, Friedman test; post-hoc tests revealed p<0.05 comparing initial dark and light, and initial dark and second dark sessions). We next aimed to study the changes in grid cell spatial firing patterns after transitioning from light to dark in more detail. Towards that aim, we recorded grid cells during alternating light/dark epochs, always beginning with a light epoch followed by a dark epoch. 28 cells from six mice were identified as grid cells based on their grid scores (see Materials and methods). To compare spatial firing properties of grid cells between light and dark conditions, we concatenated all light and dark sessions to compute firing rate maps for light and dark conditions. We found that grid cell field centers remained stable, but grid fields appeared wider during complete darkness (*Figure 6C*). Quantification of spatial periodicity and spatial precision of grid cell firing by computing grid scores and spatial information (see Materials and methods) revealed a significant decrease in both spatial periodicity and spatial precision of spike locations (*Figure 6D & E*). However, firing rate maps are the results of integrating firing rates of grid cells over long time periods, thereby measuring only spatial variability, but not spatio*temporal* variability of grid cell firing. This distinction becomes important when considering two alternative hypotheses with regard to the underlying cause for the observed broadening of grid fields in complete darkness. Grid fields could appear wider because grid cells fire over a longer distance throughout a single grid field traversal. Alternatively, grid fields could appear wider because grid cell firing becomes less stable at the level of single grid field traversals. To distinguish between these two alternative hypotheses, we used two complementary approaches, First, we computed the spatiotemporal correlation between the observed firing rate over time and the firing rate predicted by the animal's location at any given point in time and the associated value in the firing rate map at that location. The results show a significant reduction in spatiotemporal correlation during complete darkness (*Figure 6F*). Second, we computed spike-triggered firing rate maps of grid cells for the light and dark condition. Spike-triggered firing rate maps are not affected by slow translational shifts of the grid pattern. Nevertheless, we observed a significant decrease in the grid scores computed from spike-triggered firing rate maps (p=0.0354, n = 15 cells, Wilcoxon signed-rank test) (*Figure 6—figure supplements 1* and *2A*) suggesting that the phase of each grid map remained stable in darkness. Furthermore, we did not observe a ring-like pattern in the spike-triggered firing rate maps suggesting that the orientation of each grid map remained stable in darkness too. Lastly, we measured the grid spacing in each of the spike-triggered firing rate maps. Grid spacing remained unchanged in darkness (median grid spacing = 42 cm in light and dark, p=1, Wilcoxon signed-rank test, Bayes factor = 0.2005) (*Figure 6—figure supplement 2B*). Taken together, these data support the hypothesis that grid fields in the firing rate map appear broader because the spatiotemporal accuracy of grid cell firing is reduced without changes in the phase, orientation, or spacing of grid maps.

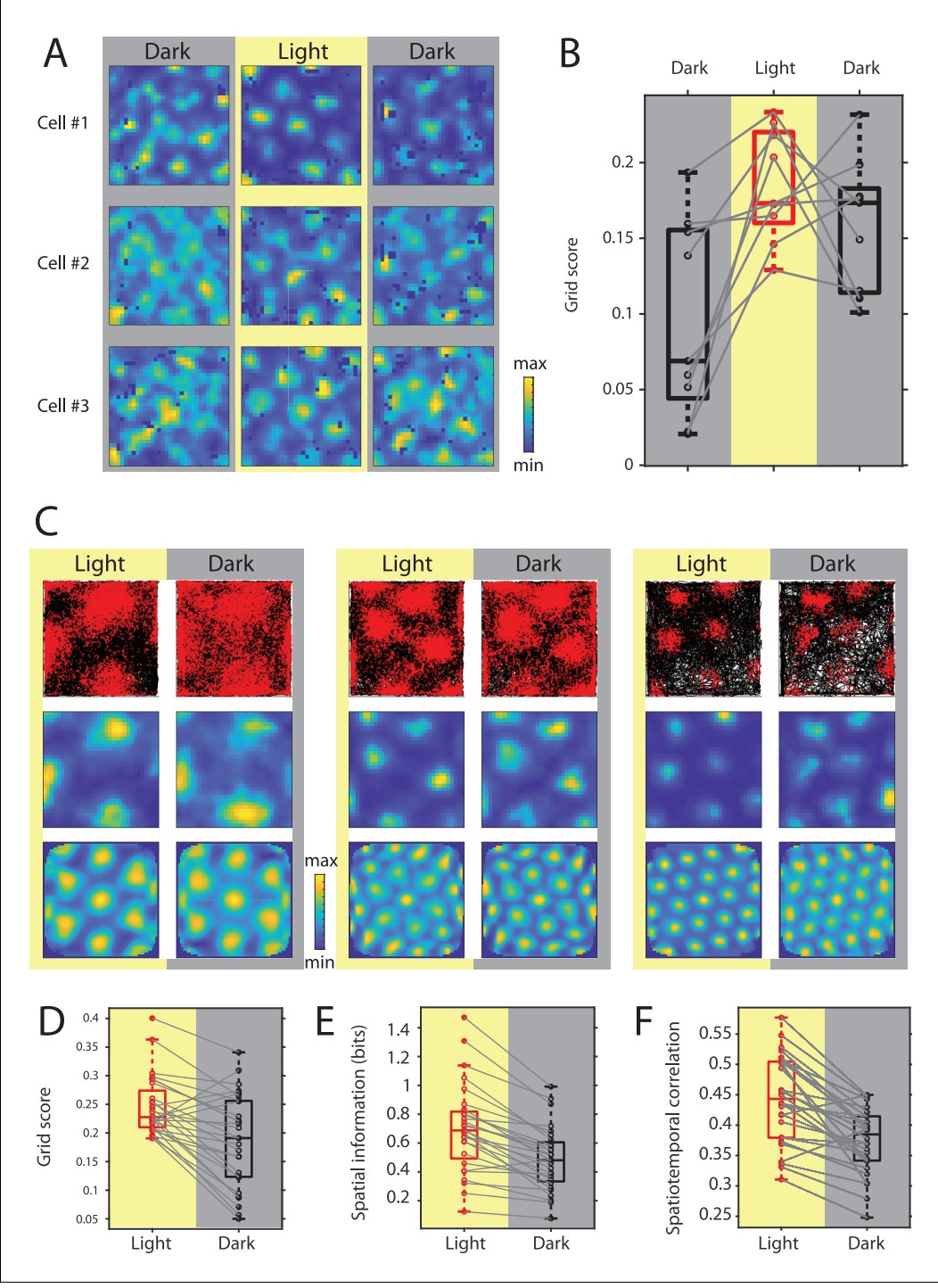

**Figure 6.** Grid cell spatial firing is maintained but less stable in the absence of visual inputs. (**A**) Firing rate maps of three example neurons during sequential dark, light, and dark sessions of 10–20 min duration. In the initial dark session, the animal was introduced to the open-field environment in complete darkness. (**B**) Box plots of grid scores during sequential dark, light, and dark session, n = 9 neurons from two mice. Black color and gray background indicate data on dark conditions, red color and yellow background indicate data on the light condition. Solid gray lines connect data points from the same grid cell. Chi-square = 10.89, p=0.0043, Friedman's test; post-hoc tests revealed significant differences between the initial dark and light sessions as well as the initial dark and second dark sessions at alpha = 0.05. (**C**) Top, middle, and bottom rows show trajectory plots, firing rate maps, and spatial autocorrelations of firing rate maps for three example grid cells recorded over multiple days during alternating light and dark epochs. Solid black lines in the trajectory plot show the trajectory of the animal,

*Figure 6 continued*

red dots mark the location of spikes during that trajectory. (D–F) Box plots of grid scores, spatial information, and spatiotemporal correlation during light and dark; Spatiotemporal correlation is the correlation between the observed firing rate and the expected firing rate computed from the animal's position in the grid firing rate map during light; gray solid lines connect data points from the same grid cells; p=2.25×10$^{-4}$, p=7.26×10$^{-6}$, and p=2.79×10$^{-5}$; n = 28, Wilcoxon signed-rank test. Firing rate maps of grid cells are provided in *Figure 6—source data 1*.

The online version of this article includes the following source data and figure supplement(s) for figure 6:

**Source data 1.** Trajectory plots, firing rate maps, and rate map autocorrelograms of the 28 grid cells analyzed in the current study.

**Figure supplement 1.** Spike-triggered firing rate maps of grid cells with a grid score >0.06 in either light or dark condition; grid scores computed from the spike-triggered firing rate maps.

**Figure supplement 2.** Grid scores are reduced without changes in the spacing of grid maps.

## Visual inputs sharpen the tuning curves of head direction cells and border cells in the medial entorhinal cortex

Grid cell firing may rely on an intact head direction signal, in particular on the firing of head direction cells (*Winter et al., 2015a*). We therefore investigated whether the reduction in spatial accuracy of grid cell firing was accompanied by changes in the head direction tuning of head direction cells in the MEC. We identified 67 cells with significant unidirectional head direction tuning which were classified as head direction cells (*Sargolini et al., 2006*; *Taube et al., 1990b*; *Taube et al., 1990a*). Interestingly, head direction cells maintained their head direction preference in the dark condition (*Figure 7A & B* and *Figure 7—figure supplement 1*) suggesting that animals remained oriented within the environment during complete darkness. However, the mean resultant lengths of head directional firing rate maps were decreased in the dark condition (Cohen's d = 0.238, p=0.0337, Wilcoxon signed-rank test) suggesting that the accuracy of head directional tuning was reduced in darkness (*Figure 7C*). In addition to grid cells, border or boundary vector cells (*Lever et al., 2010*; *Solstad et al., 2008*) are hypothesized to contribute to a representation of space in the MEC. We identified 27 border cells (*Figure 7D* and *Figure 7—figure supplement 2*) and compared their border scores (*Solstad et al., 2008*) between light and dark conditions. Border scores were significantly decreased during darkness (Cohen's d = –0.4975, p=0.0325, Wilcoxon signed-rank test) (*Figure 7E*) suggesting that representations of boundaries in the open-field environment were less accurate during darkness.

## Changes in grid cell spatial stability as a function of time correlate with changes in a proposed speed signal by local field potential theta frequency

Our data set included data on n = 7 grid cells from three mice recorded over an exceptionally long time period across multiple days. These data provided sufficient statistical power and sufficient sampling of the environment to analyze the time courses of the observed changes in spatial periodicity, spatial selectivity and spatiotemporal stability of grid cell firing during light and dark conditions. Towards this goal, we first computed firing rate maps for each of the 10 s bins of activity to analyze the time course of spatial periodicity quantified by the grid score (*Figure 8A & B*). Second, we computed the spatial correlations between the firing rate maps of the 10 s binned data and the overall firing rate map for the light condition (*Figure 8C*). Third, we computed the spatiotemporal correlation for each of the 10 s bins (*Figure 8D*). We found a saturating exponential increase in spatial periodicity of grid cell firing (*Figure 8B*), spatial correlation (*Figure 8C*), and spatiotemporal correlation (*Figure 8D & E*) during the light condition. During darkness, we observed a similar, though weaker, trend in the opposite direction. Intriguingly, grid cell stability measured as spatiotemporal correlation of grid cell firing was strongly correlated with the slope of the LFP theta frequency vs. running speed relationship (R = 0.966, p=1.83×10$^{-21}$, n = 36 (18 time points x two conditions); *Figure 8F* and *Figure 8—figure supplement 1*). The time courses of changes in grid cell stability and the slope of the LFP theta frequency vs. running speed relationship were also correlated within light and dark conditions (Light: R = 0.825, p=2.55×10$^{-5}$, n = 18 time points; Dark: R = 0.490, p=0.039, n = 18 time points) further confirming the link between grid cell stability and the LFP theta frequency to

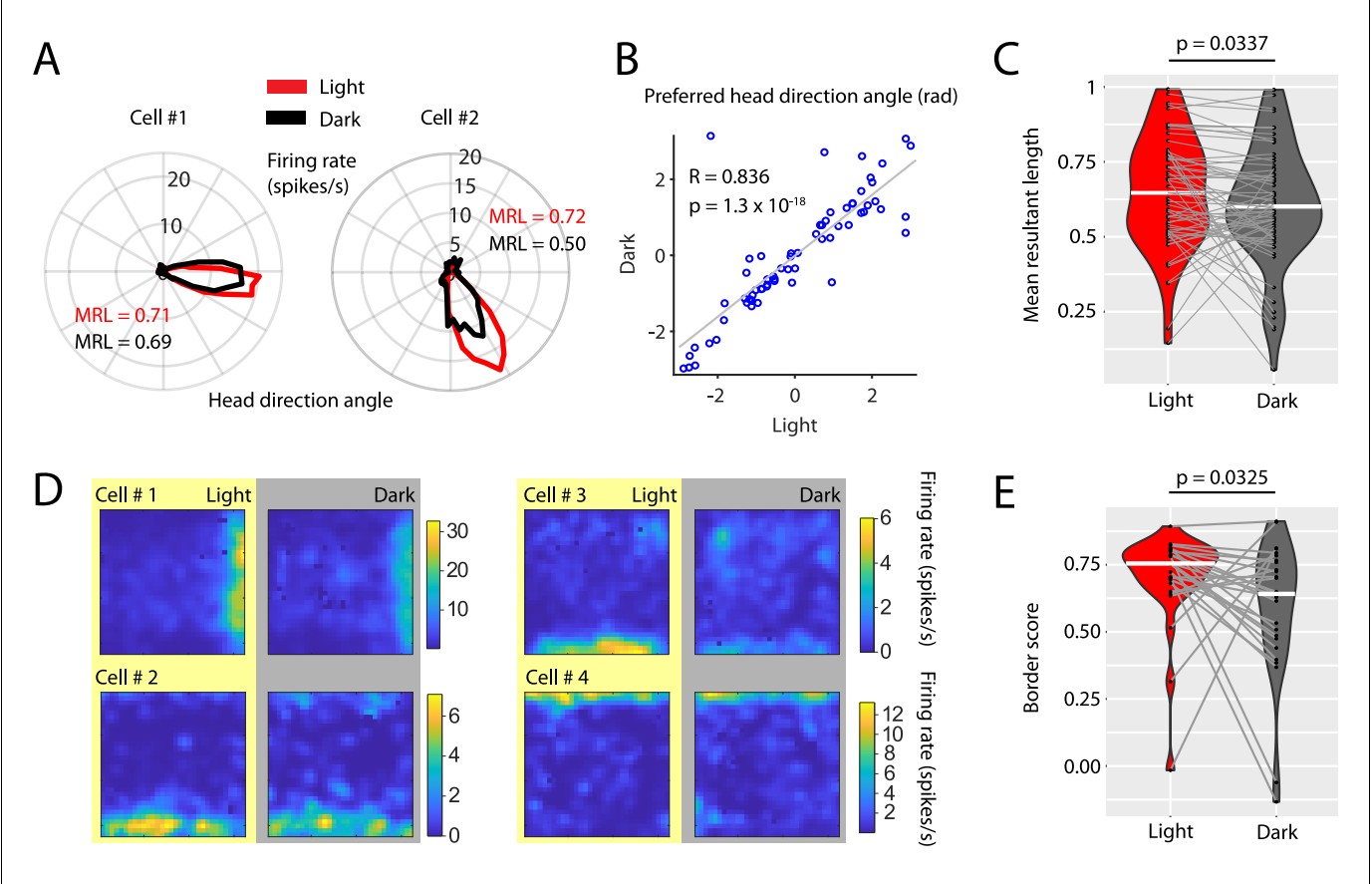

**Figure 7.** Visual inputs sharpen the tuning curves of head direction cells and border cells. (**A**) Polar histograms showing the head directional tuning of two head direction cells during light (red) and dark (black) conditions. (**B**) Scatter plot comparing the preferred head direction angles of n = 67 head direction cells between light and dark conditions. Note the strong circular-linear correlation. (**C**) Violin plots showing the distributions of mean resultant lengths of head directional tuning curves for light (red) and dark (gray) conditions. White horizontal lines mark the medians. Gray horizontal lines connect data points from the same neuron. p=0.0377, Wilcoxon signed-rank test. (**D**) Firing rate maps of four border cells for light and dark conditions. (**E**) Violin plots comparing the distribution of border scorers of n = 27 border cells between light and dark conditions. p=0.0325, Wilcoxon signed-rank test. Underlying data on mean resultant lengths of head direction tuning during light and dark conditions are provided in *Figure 7—source data 1*. The online version of this article includes the following source data and figure supplement(s) for figure 7:

**Source data 1.** Data on mean resultant lengths of head direction tuning during light and dark conditions for the 67 neurons with significant head direction tuning in either light or dark condition.

**Figure supplement 1.** Polar histograms showing the head directional tuning in light and dark conditions of the 69 head direction cells analyzed in this study.

**Figure supplement 2.** Firing rate maps for the 27 border cells identified in this study.

running speed relationship. In contrast, changes in the y-intercept of the LFP theta frequency vs. running speed relationship as a function of time were negatively correlated with changes in grid cell stability in the light condition (R = –0.633, p=0.0048, n = 18 time points) and uncorrelated to changes in grid cell stability in the dark condition (R = 0.056, p=0.83, n = 18 time points) (*Figure 8—figure supplement 2*). Taken together, these data demonstrate that the representation of space by grid cell firing is correlated to the speed signal by LFP theta frequency and that both of these signals undergo plastic changes at a behavioral time scale of tens of seconds.

## Discussion

Data presented in this study show how changes in visual inputs affect firing rates and theta rhythmic firing dynamics in the MEC. Furthermore, these changes correlate with changes in the representation

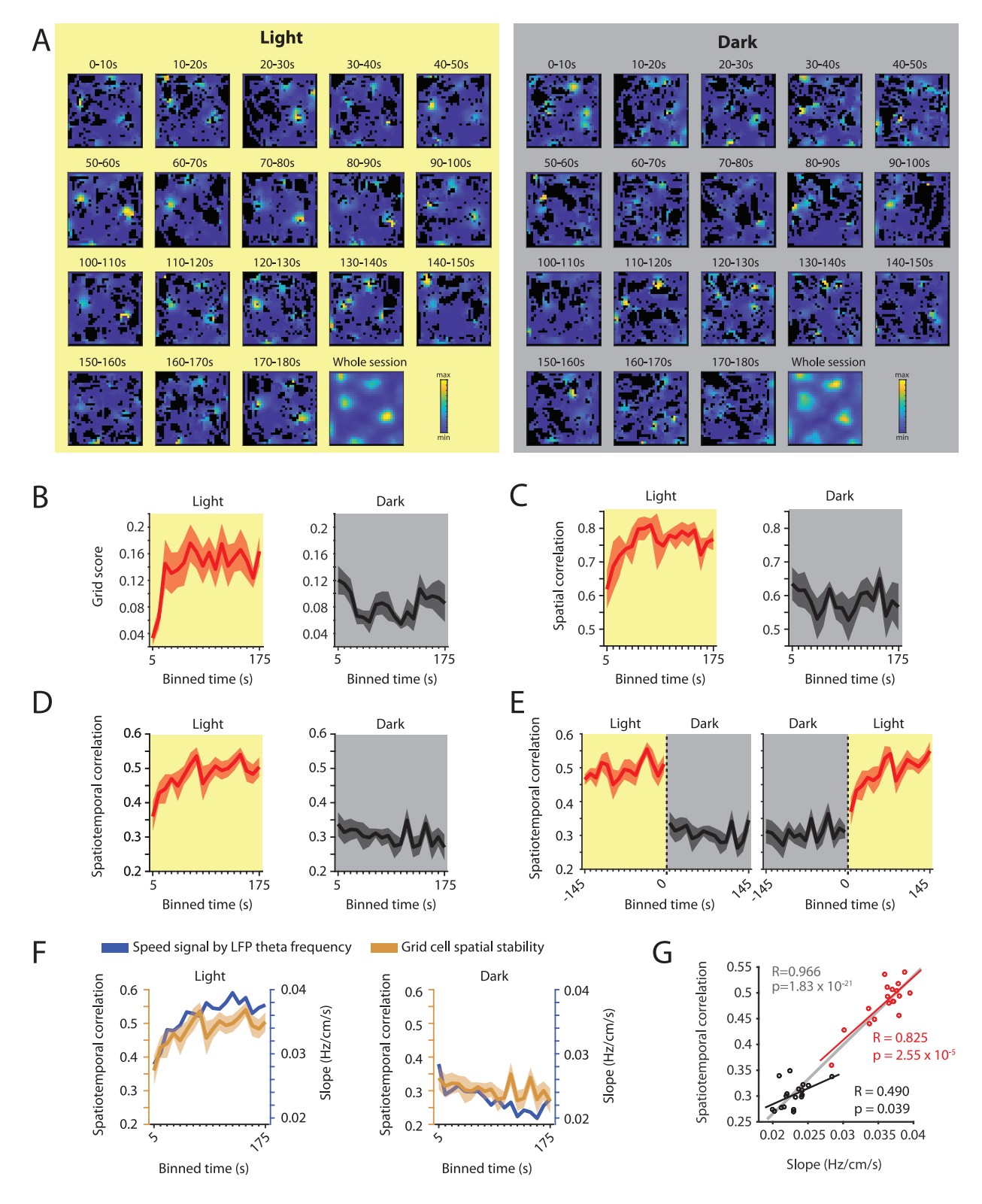

**Figure 8.** Changes in grid cell spatial stability as a function of time correlate with changes in the slope of the local field potential theta frequency vs. running speed relationship. (**A**) Firing rate maps for consecutive 10 s bins of data after transitioning from dark to light (left side, yellow background) and light to dark (right side, gray background) for one example neuron. (**B–D**) Time course measured in 10 s bins of grid scores (**B**) spatial correlation (**C**) and spatiotemporal correlation (**D**) during the first three minutes of light and dark sessions. Solid lines show mean firing rates, shaded area indicates

*Figure 8 continued on next page*

*Figure 8 continued*

s.e.m.; n = 7 cells from three mice. (E) Left and right panels show 10 s binned data on spatiotemporal correlation around the transition points from light to dark and dark to light, respectively. (F) Overlay of data shown in D and 1I showing the temporal alignment of changes in the slope of the theta frequency vs. running speed relationship (blue colors) and the spatial stability of grid cell firing measured by the spatiotemporal correlation (ocher colors). Each data point shows the mean values in 10 s time bins of the first 180 s after the start of the light (left panel) or dark (right panel) condition. (G) Scatter plot of time-binned data shown in F; each data point shows the mean values of the slope and grid cell stability at each of the time points shown in F. Red and black lines show the linear regressions of data in the light and dark conditions, respectively (Light: R = 0.825, p=2.55×10$^{-5}$, n = 18 time points; Dark: R = 0.490, p=0.039, n = 18 time points). Gray line shows the linear regression of the combined data set, n = 36 (18 time points x two conditions), R = 0.966; p=1.83×10$^{-21}$.

The online version of this article includes the following figure supplement(s) for figure 8:

**Figure supplement 1.** Left side shows the slope of the local field potential (LFP) theta frequency vs. running speed relationship as a function of 10 s time bins around the transition points from light to dark and from dark to light for the three individual mice in which grid cells could be recorded over multiple days and sessions.

**Figure supplement 2.** Changes in grid cell spatial stability as a function of time do not align with changes in the y-intercept of the local field potential theta frequency vs. running speed relationship.

of location by grid cells, representation of environmental boundaries by border cells, and representation of head direction by head direction cells in the MEC. Intriguingly, a proposed speed signal by firing rate—the modulation of firing rate by running speed—remains unaltered if the analysis is adjusted for changes in mean firing rate. However, an alternative speed signal by LFP theta frequency—the linear correlation between LFP theta frequency and running speed—is a function of visual inputs and reduced in darkness. Intriguingly, changes in the slope of the LFP theta frequency vs. running speed relationship are strongly correlated to the changes in spatial stability of grid cell firing. While changes in firing rates of MEC neurons and changes in theta rhythmic firing properties of MEC neurons occur almost instantaneously during the transitions between light and dark, changes in the proposed speed signal by LFP theta frequency and changes in grid cell stability show a bimodal change with a fast (<10 s) component and a further slow (10–180 s) saturating exponential component. Such changes in response to changes in sensory input are longer than previously addressed by theoretical and computational models of grid cell firing. Lastly, we report changes in the running speed-dependent population dynamics of theta rhythmic MEC neurons upon changes in visual inputs. Synchronization of theta rhythmic firing across principal and interneurons increased with the onset of running, and even more so during darkness. The stronger synchronization across neurons during darkness was accompanied by an increase in theta-frequency phase locking of spiking activity (*Figure 3*). While the frequency of theta rhythmic firing increased with running speed during the light condition in both principal and interneurons, no such changes in theta rhythmic firing frequency as a function of running speed were observed during darkness. Likewise, the magnitude of theta rhythmic firing of interneurons increased as a function of running speed during the light, but not during the dark condition.

It has previously been proposed that disruption of speed signals may cause disruption of spatially periodic grid cell firing in the MEC. Current computational models of grid cell firing and path integration require a linear speed signal (*Burak and Fiete, 2009*; *Burgess, 2008*; *Burgess et al., 2007*; *Fuhs and Touretzky, 2006*; *McNaughton et al., 2006*). Two candidates for such a speed signal terminating in the MEC have been proposed: a speed signal by firing rate, most clearly present in a subpopulation of 'speed cells' showing positive and linear speed tuning curves (*Kropff et al., 2015*); and an oscillatory speed signal by changes in LFP theta frequency or by changes in the theta rhythmic firing frequency of individual MEC neurons (*Burgess, 2008*; *Hinman et al., 2016*). Both speed signals can change in response to changes of the size and shape of the environment (*Munn et al., 2020*), as do grid cells (*Barry et al., 2007*; *Krupic et al., 2015*). In this study, we examined how changes in spatial stability of grid cells relate to changes in the potential speed signals by firing rate and LFP theta frequency as a function of time.

In order to analyze the time course of grid cell stability over time after transitions between light and dark, we took advantage of long-term recordings of the same grid cells over multiple days providing us with sufficient statistical power. Intriguingly, changes in spatial stability of grid cell firing showed an instantaneous component and a slow saturating exponential component at a behavioral time scale of tens of seconds after transitions between light and dark conditions. Notably, the

observed saturating exponential component in changes of grid cell spatially periodic firing supports an earlier report by *Pérez-Escobar et al., 2016*. We next tested if the observed change in spatial stability of grid cell firing could be explained by a change in the potential speed signal by firing rate. We distinguished the effect of visual inputs on the mean firing rate of neurons from the effect of visual inputs on the running speed-dependent gain in firing rates, which is equivalent to the slope in speed response curves after normalization within conditions. We further corrected for multiplexed or conjunctive coding properties by computing speed response curves using a linear-nonlinear model (*Hardcastle et al., 2017*). After adjusting for changes in mean firing rates and conjunctive coding properties, we found that visual inputs had no impact on the running speed-dependent gain in firing rates of positive speed cells in the MEC, while the running speed-dependent gain was even slightly increased in negative speed cells. We therefore conclude that changes in the potential speed signal by firing rate cannot explain the change in spatially periodic grid cell firing. A previous study (*Pérez-Escobar et al., 2016*) has found that speed scores were changed between light and dark conditions. However, that study used a speed score threshold to identify speed cells which is biased for identification of linearly modulated neurons because the speed score is a measure of linear correlation. In contrast, the current study used the LN-model to identify speed cells. As a consequence, our data set includes many speed cells with non-linear speed response curves, where the speed score is a sub-optimal measure. Like the slope of the speed tuning curve, the speed score is also a function of firing rate and previously reported changes in the speed score of speed cells during darkness may have been driven by changes in mean firing rates as opposed to changes in the running speed-dependent gain in firing rate.

We next turned to the analysis of the time courses of changes in the two oscillatory speed signals, namely the speed signal by LFP theta frequency and the speed signal by theta rhythmic firing frequency. Notably, the neuronal distribution of theta rhythmic firing frequencies covers a broad range of theta (4–12 Hz) indicating that rhythmic firing of MEC neurons is only loosely coupled to the LFP theta frequency. Moreover—in contrast to LFP theta rhythmicity—theta rhythmic firing persisted during immobility, though significantly less synchronous across neurons. The observed changes in theta rhythmic firing frequency at the single cell and population levels provide a possible explanation for why the LFP theta amplitude vs. running speed relationship is not changed during complete darkness. During darkness, movement-related theta-rhythmicity magnitude is decreased on the single cell level, while theta synchronicity across neurons is increased on the population level. These two effects would influence the LFP theta amplitude in opposite directions and thus may cancel each other out. Intriguingly, the time course of changes in the slope of the LFP theta frequency vs. running speed signal matched the time course of changes in grid cell spatial stability (*Figure 8F*). In contrast, changes in theta rhythmic firing frequency showed only the fast (< 10 s) component of change (*Figure 2H & I*). These data on the parallel time course of changes in the oscillatory speed signal by LFP theta frequency and grid cell stability suggest that grid cell firing may be more strongly associated with changes in the mean attractor state of rhythmic firing in the MEC reflected by changes in the LFP theta signal than with changes in the rhythmic firing of individual neurons. Alternately, the exponential change in LFP theta may reflect a slow change in an oscillatory speed signal arising from other cortical structures such as the medial septum.

Other data support the role of LFP oscillatory dynamics in the generation of the grid cell spatial code. In contrast to the speed signal by firing rate (*Dannenberg et al., 2019*; *Hinman et al., 2016*), the theta oscillatory speed signal depends on the MSDB (*Chrobak et al., 1989*; *Dannenberg et al., 2015*; *Givens and Olton, 1994*; *Hinman et al., 2016*; *Lawson and Bland, 1993*), and pharmacological inactivation of MSDB neurons strongly reduces both theta oscillations and spatial periodicity of grid cell firing in the MEC (*Brandon et al., 2011*; *Koenig et al., 2011*). Pharmacological inactivation of the medial septum also impairs self-motion-based linear distance estimation (*Jacob et al., 2017*). Furthermore, passive transportation of rats reduces theta oscillatory power and abolishes both spatially periodic grid cell firing and the speed signal by LFP theta frequency (*Winter et al., 2015b*). Likewise, pharmacological inactivation of the vestibular system or preventing linear/translational head movements in a virtual reality setup abolish the modulation of LFP theta frequency by running speed and are associated with a decrease in the spatial information score and an increase in grid scale (*Chen et al., 2018*; *Jacob et al., 2014*). Interestingly, the increase in grid scale observed during path integration on a circular track is associated with a change in the frequency of LFP theta oscillations, but not with a change in the firing patterns of speed and head direction cells

(*Jacob et al., 2019*). Moreover, a recent study demonstrated that artificial low-theta stimulation can affect an animal's estimation of linear distance during approaches to a goal zone (*Mouchati et al., 2020*) further strengthening a potential link between LFP theta frequency and path integration. The slope of the LFP theta frequency vs. running speed relationship has thus been linked to spatial cognition and grid cell firing (*Korotkova et al., 2018*). In addition, speed modulation of hippocampal theta frequency has been shown to correlate with spatial memory performance (*Richard et al., 2013*). We demonstrate in this study that the slope of the LFP theta frequency to running speed relationship and grid cell stability change as a function of visual inputs and as a function of time in a familiar environment. One possible explanation for the correlation between changes in spatial accuracy of grid cell firing and the slope of the LFP theta frequency is that the latter serves as a speed signal in the path integration process generating the spatially periodic grid cell firing pattern as suggested by oscillatory interference models of grid cell firing (*Burgess, 2008*; *Burgess et al., 2007*; *Hasselmo, 2008*). The reduced slope may then result in a lower resolution for controlling a potential path integration mechanism due to the reduced dynamic range of the oscillatory speed signal resulting in less accurate readout and increased variability. Importantly, this would require a rescaling of the oscillatory speed signal which may be facilitated by knowledge of the distances between the borders of a familiar environment. Previous studies have shown that environmental novelty reduces the slope of the LFP theta frequency to running speed relationship to a similar degree as darkness (*Wells et al., 2013*) but resulted in an increase in grid scale (*Barry et al., 2012*) as opposed to an increase in spatial stability. Taken together, these data may indicate that in a novel environment, a reduced slope in the LFP theta frequency to running speed relationship may be the consequence of underestimating the distances between borders resulting in a larger grid scale but that in a familiar environment, the lack of error correction in a path integration mechanism may result in a reduction of the dynamic range in an oscillatory speed signal by LFP theta frequency resulting in less accurate speed coding.

Data presented in this study also show that the reduction in spatial accuracy of grid cell firing is accompanied by a reduction of accuracy in head directional tuning and a reduction of spatial accuracy of border cell firing (*Figure 7*). An alternative explanation for the observed reduction of spatial accuracy in grid cell firing during darkness is therefore a reduction of accuracy in head directional signals (*Winter et al., 2015a*), boundary signals, or both due to the loss of visual information. Another possibility is that the frequency of theta rhythmic firing of MEC neurons serves as a speed signal that is integrated over longer time scales than previously accounted for by computational models of grid cells (*Dannenberg et al., 2019*). In that scenario, both the slope of the LFP theta frequency and spatial periodicity of grid cell firing would depend on the history of a velocity signal, which would be consistent with the observed slow saturating exponential component of change. Such an integration over seconds has previously been shown to optimize the proposed speed code by firing rate (*Dannenberg et al., 2019*). Notably, the slow component of change in spatial accuracy of grid cell firing cannot be explained by error accumulation alone. While error accumulation could account for the observed slow *decrease* in spatial stability after removal of visual inputs, error accumulation cannot be the reason for the observed slow *increase* in spatial accuracy after reinstating visual inputs. If signals required for grid cell firing are integrated over time, however, such a slow increase is expected. Notably, a similar increase in spatial accuracy of grid cell firing was also reported recently by *Weber and Sprekeler, 2019*. Future experimental and computational modeling studies are needed to address the questions whether and how longer integration time windows could support grid cell firing.

The observed changes in neural correlates of running speed and location during complete darkness may be attributed to either a loss of visual cues for navigational purposes or alternatively to a net decrease in excitatory drive from areas in the visual cortex due to the lack of retinal stimulation (*Campbell and Giocomo, 2018*). In order to distinguish between these two possibilities, we took advantage of control data from an optogenetic study (*Dannenberg et al., 2019*), where a visible bright spot of light was moving geostationary to the animal without further illuminating the environment. During this manipulation, the amount of light hitting the retina increased without changing the visibility of static visual landmark cues. This perturbation resulted in very similar changes of firing rates in MEC neurons as observed during the light vs. dark paradigm (*Figure 4*). Importantly, changes of firing rates, theta rhythmic firing frequency, and theta-phase coupling of rhythmic firing occurred almost instantaneously upon changes in visual inputs (*Figure 4C & D* and *Figure 2I*). Taken

together, these data suggest that changes in firing rates and theta rhythmic firing are primarily driven by changes in net excitatory input from visual cortex and do not reflect changes in the perception of the spatial aspects of the environment. Instead, our data suggest that changes in firing rates and theta rhythmic firing are driven by changes in retinal stimulation. In contrast, the modulation of firing rates by running speed is unaltered by changes in illumination or presence or absence of visual cues.

Two recent studies on freely moving mice reported a major disruption of spatially periodic grid cell firing in the absence of visual cues (*Chen et al., 2016*; *Pérez-Escobar et al., 2016*). However, grid cell firing is intrinsically linked to path integration-based navigation (*Gil et al., 2018*; *Jacob et al., 2019*) and results from studies using virtual reality demonstrate that physical motion has a greater influence on grid cell firing in mice than visual inputs (*Campbell et al., 2018*; *Chen et al., 2019*; *Tennant et al., 2018*). While the recording environment in the Chen et al. study did not differ significantly from the recording environment used in this study, the majority of light and dark trials performed in the Chen et al. study was discontinuous and, in most trials, mice were introduced to the recording environment in complete darkness. Likewise, we also found a significantly stronger disruption in spatially periodic grid cell firing when mice were introduced to the recording environment in complete darkness (*Figure 6A and B*). Conversely, Chen et al. found a less severe disruption in grid cell firing during darkness in the few trials where mice were introduced to a familiar environment with the room lights turned on. Interestingly, Chen et al. reported changes in the preferred head direction angles of head direction cells suggesting that mice were frequently disoriented. The study by Pérez-Escobar et al. used a circular maze environment surrounded by water with no walls to eliminate tactile and olfactory cues. In addition, a white noise generator was used to mask auditory cues. Finally, mice underwent an explicit disorientation procedure before the start of recordings. These recording settings resulted in an almost complete loss of head directional tuning and very likely left animals disoriented. In contrast to those two previous studies, this study investigates the specific contribution of visual inputs on spatially periodic grid cell firing and theta oscillatory network dynamics in the MEC by selectively manipulating visual inputs on alternating epochs but leaving other static environmental and sensory cues such as walls, auditory cues, and potential olfactory cues unaltered. During this more selective manipulation, grid cells in the MEC persisted during complete darkness as shown by unaltered orientation, phase, and scale of grid cell firing fields. However, accuracy of grid cell firing was reduced during darkness. This reduction of accuracy in grid cell firing was accompanied by a reduction in the accuracy of the representation of head direction and environmental boundaries in the MEC. While the preferred angle of head directional firing in head direction cells was not altered during darkness, the variability in head directional firing increased during darkness (*Figure 7A–C*). Likewise, border cells continued to fire in their respective place fields yet with less spatial accuracy (*Figure 7D & E*). Taken together, the data presented in this study suggest that animals need an initial orientation in the environment supported by static visual cues but can then use sensory cues other than visual cues, including wall encounters, to roughly maintain that orientation over long time periods. For maintaining a spatial representation in a completely dark environment, vestibular signals, motor efference copies, proprioceptive feedback, and external geometrical cues may be sufficient to maintain the orientation, scale, and phase of grid cell firing fields. During the light condition, static visual cues may offer additional cues to self-location since the distance and direction to these cues systematically change with the animal's movement, and optic flow may be used for refining an estimate of self-motion.

Theta signals within the hippocampal formation have been suggested to not only reflect alterations in sensorimotor integration and the flow of sensory input, but also distinct cognitive operations (*Hinman et al., 2011*). Furthermore, much of theta physiology and function seems to be shared across rodents and primates including the correlation of theta frequency with movement speed (*Goyal et al., 2020*). However, there is currently little data on dissociations between the slope and y-intercept of the theta frequency to running speed relationship. Data presented in this study show that spatial accuracy of grid cell firing is intrinsically linked to the slope but not the y-intercept of the LFP theta frequency vs. running speed relationship. Slow changes in both the slope of the LFP theta frequency vs. running speed relationship and the spatial periodicity of grid cells after transitions between light and dark conditions suggest that the representation of space in a cognitive map may integrate velocity signals over past time and may therefore be history-dependent and updated at a

slow behavioral time scale. The result of such an integration would be a representation of space which is malleable by internal memory processes.

## Materials and methods

### Subjects

Data were collected from a total of 17 adult 3–6 month-old male mice (*Mus musculus*) of wild type and transgenic backgrounds (Wild type, C57Bl/6J; ChAT-IRES-Cre, B6;129S6-*Chat*$^{tm2(cre)Lowl}$/J; PV-IRES-Cre, B6;129P2-*Pvalb*$^{tm1(cre)Arbr}$/J). Experimental PV-IRES-Cre mice were homozygous and experimental ChAT-IRES-Cre mice were heterozygous. All experimental procedures were approved by the Institutional Animal Care and Use Committee for the Charles River Campus at Boston University. Mice underwent surgery for implantation of 2 movable tetrodes targeting the superficial layers of the MEC in the left hemisphere. After surgery, mice were separated for individual housing and maintained on a reversed 12 hr light/12 hr dark cycle.

Mice were habituated to the experimenter and testing room before the start of all experimental procedures. For the purpose of additional experiments not reported in this study, mice also underwent rAAV injections of Archaerhodopsin T or GCaMP6f into the medial septum and light fiber implantation targeting the medial septum (*Dannenberg et al., 2019*). Light delivery of ~1–15 mW into the medial septum during one or two test recording sessions did not show any effects and we did not continue light delivery for the rest of the study. We did not notice any differences between data recorded before those test stimulations and after those test stimulations and also did not observe any differences between those mice which had and had not been stimulated. No differences were observed between wild type or transgenic mice. Prior to surgery, mice were housed in Plexiglas cages together with their siblings.

### Surgery for microdrive implantation in mice

Mice were injected with buprenorphine (0.1 mg/kg, s.c.) and atropine (0.1 mg/kg, i.p.), and chronically implanted under isoflurane anesthesia with 8-channel Axona microdrives (Axona, St. Albans, UK) carrying two movable tetrodes targeting the superficial layers of the dorsal MEC in the left hemisphere. One ground screw was implanted above the cerebellum, five anchoring screws were positioned across the skull, and a craniotomy was performed for implantation of the microdrive, 0.5 mm anterior to the edge of the transverse sinus and 3.4 mm lateral to the midline. Tetrodes were then implanted with a 6° polar angle and lowered 1 mm into the brain. The microdrive was secured to the skull with dental cement. Animals were given buprenorphine (0.1 mg/kg, i.p.), enrofloxacin (7.5 mg/kg, i.p.), and ketoprofen (3 mg/kg, i.p.) during a 5 day postsurgical care period and allowed one week in total to fully recover after surgery before the start of recordings.

### In vivo electrophysiological recordings

Before implantation, impedance of tetrodes (17 µm platinum wires, California Fine Wires) was adjusted by electroplating to <150 kOhm. Tetrodes were lowered daily by 50 µm until the abundance of spikes and the presence of theta rhythmic cells in test recordings indicated that tetrode tips were located near or within the superficial cell layers of the MEC. Neural signals were pre-amplified by unity-gain operational amplifiers located on each animal's head (head stage) and AC-wire-coupled to an Axona recording system (Axona Limited, St. Albans, UK). Spikes were threshold-detected and recorded at 48 kHz. The LFP signal was recorded single-ended with the ground screw as reference at 4800 Hz. Position and head direction of mice were tracked with headstage-mounted invisible infrared LEDs at 50 Hz. Spikes were clustered into single units using spike clustering software (DacqUSB/Tint, Axona, St. Albans, UK). Further offline analyses were performed using Matlab (MathWorks, Natick, MA, USA), the circular statistics toolbox (*Berens, 2009*) for Matlab, and R (R Foundation for Statistical Computing, Vienna, Austria).

### Behavior during light and dark conditions

All recordings were performed while animals foraged for randomly scattered small pieces of Froot Loops (Kellogg Company, Battle Creek, MI, USA) in a 1 m square open-field arena with 60 cm high black walls and a white letter size cue card at one wall. Recordings were performed during

alternating epochs of open-field illumination (light) and darkness (dark). The dark condition was completely dark from the perspective of the human experimenter after adaptation to the dark condition for at least five minutes. A ceiling light provided full illumination of the open field during the light condition. Turning that ceiling light off resulted in complete darkness, which was achieved by the following measures. The open-field environment was located at the end of the recording room surrounded by three walls with no doors or windows and a thick laser-proof black curtain covering the front side. All LED lights of all electrical equipment were completely shielded with black duck-tape. Tracking of position and head direction of the mouse was achieved by using invisible infrared LEDs. These infrared LEDs were additionally covered with tape to minimize their brightness and preventing a potential spectral bleed into the visible spectrum. The slits around the door to the recording room were covered with black shower curtain material to minimize light leaking into the space within the recording room outside the laser-proof curtain. The computer monitors outside of the curtain were also covered with black shower curtains. Importantly, the experimenter frequently tested for any potential light sources within or light leaks into the space around the open-field environment. Recordings only began when it was completely dark from the perspective of the human eye after having adopted to the dark condition for at least 5 min. Alternating epochs of recordings in light and darkness could vary in duration between 30 s and 30 min; most experiments used 3 min alternating epochs allowing high statistical power due to multiple repetitions from light to darkness in each session. A typical recording session lasted 40–60 min.

Experiments with alternating epochs of light and dark began with the light condition. Experiments with a dark-light-dark paradigm started in complete darkness. For those experiments, mice were brought into the room with their cage covered by a black curtain, and mice were plugged in and introduced into the arena during complete darkness by an experimenter wearing night-vision goggles. The experimenter then left the arena side of the curtain before recordings began.

## Histology

After the end of electrophysiological data collection, animals were deeply anesthetized by isoflurane or i.p. injection of Euthasol (390 mg/kg), followed by electro-lesions (20-μA, 16 s) through one channel of each tetrode and subsequent transcardial perfusion with saline followed by 10% buffered formalin (SF100-4, ThermoFisher Scientifc). Brains were extracted and stored in fixative for one day. Cresyl violet staining was performed on 50 μm sagittal slices of MEC to visualize tetrode tracks and confirm positions of recording sites in the MEC.

## Data analysis

### Computation of running speed and firing rate signals

The animal's running speed was estimated by applying a Kalman filter to the positional data obtained from the video tracking sampled at 50 Hz (*Fyhn et al., 2004*; *Hinman et al., 2016*; *Kropff et al., 2015*). To obtain an instantaneous firing rate signal matching the sampling rate of the behavioral tracking, we calculated the firing rate as the number of spikes per video frame period (20 ms) and smoothed this signal with a 125 ms Gaussian filter.

### Modulation of mean firing rates by visual inputs

In order to identify neurons, whose firing rates are modulated by the illumination of the environment (consistently either higher or lower firing rates during light or darkness), we computed the average firing rate for each light and dark epoch and tested if the differences between light epochs and the following dark epochs and the differences between dark epochs and the following light epochs reached statistical significance at alpha = 0.05 using Student's paired t-test. To analyze the time course of changes in mean firing rates after transition from light to dark or from dark to light, we first computed the peristimulus time histogram with 5 s binning around the transition point and then computed the average of z-scored firing rates across neurons. Only cells which showed spiking activity across three light-to-dark and three dark-to-light epoch pairs were included into this analysis.

## LFP theta rhythm

LFP data were down-sampled to 600 Hz and power spectral density analyses were performed using custom written MATLAB scripts and the FieldTrip toolbox (*Oostenveld et al., 2011*). A theta-

frequency range was determined for each session based on the range of the theta peak in the power spectral density plot (*Dannenberg et al., 2019*). The LFP signal was then band-pass-filtered at this theta-frequency range with a third order Butterworth filter, and the instantaneous theta amplitude and phase were determined via a Hilbert transform. The instantaneous theta frequency could then be inferred from the instantaneous phase. To analyze changes in the LFP theta frequency vs. running speed relationship over time, we first down-sampled the LFP theta-frequency signal to 50 Hz in order to match the temporal binning of the behavioral tracking. We next sectioned data into 10 s blocks after transitions between light and dark and pooled data from all sessions and all mice (*Figure 1H–L*). For each of those 10 s blocks of data, the slope and y-intercept of the correlation between LFP theta frequency and running speed were computed using a linear model. The same analysis as described above was applied to 10 s blocks around the transition points from light to dark or from dark to light (*Figure 1K & L*).

## Theta rhythmic firing

We used maximum likelihood estimation (MLE) of a parametric model of the interspike intervals underlying the spike time autocorrelogram to analyze theta rhythmic firing, which has been proven a more sensitive and statistically rigorous method than traditional measures of spike time autocorrelogram features (*Climer et al., 2015*). In particular, this model allowed us to statistically test significance of theta modulation in MEC neurons and compare theta rhythmic firing frequency and theta-rhythmicity magnitude on a single cell level between light and dark conditions. The model tested for rhythmic firing in the frequency range between 1–13 Hz. For the analysis of the relationship between theta rhythmic firing and running speed (*Figure 2*), we first identified continuous temporal epochs larger than 400 ms, in which the running speed of the animal varied less than ±4 cm/s. We next sorted these epochs according to the average running speed during the epochs into 1 cm/s bins from 1 cm/s to 29 cm/s and used the MLE approach to fit an autocorrelogram model to each of those epochs. The minimum number of interspike intervals which need to be present in the autocorrelogram to compute an accurate model is 100 (*Climer et al., 2015*), and no autocorrelogram model was computed if the number of interspike intervals was smaller than 100 for the epoch of interest. For each neuron and running speed bin, an average autocorrelogram was computed across the autocorrelograms for each epoch. The maximal run time for the theta rhythmic firing vs. running speed analysis on the research computing cluster was set to two weeks resulting in the successful analysis of 340 out of 342 neurons. For the analysis of the time course of changes in theta rhythmic firing after transition from light to dark or from dark to light, the first two minutes after the transition were sectioned into 10 s blocks (n = 342 neurons, *Figure 2H*). Alternately, 10 s blocks around the transition points from light to dark or from dark to light were analyzed (n = 342 neurons, *Figure 2I*). For each neuron, an autocorrelogram model was fitted to each of those 10 s blocks of data using the MLE approach. For each of those 10 s blocks of data, the average theta rhythmic firing frequency across neurons could then be computed from the peaks of the average autocorrelogram model across neurons by taking the inverse of the first peak in the autocorrelogram model. To analyze running speed-dependent theta rhythmic firing on the population level, we first identified continuous epochs of running speed within the ranges of 0–3 cm/s (immobility), 5–10 cm/s, 10–15 cm/s, 15–20 cm/s, and 20–25 cm/s. We next used the MLE approach to model an autocorrelogram for each running speed bin of each neuron. Neurons with sufficient number of spikes resulting in at least 100 interspike intervals to compute an MLE-modeled autocorrelogram in at least one of the running speed bins were included into the analysis (n = 339 neurons).

## Analysis of theta-frequency phase locking

The analysis of theta-frequency phase locking is only valid when LFP theta rhythm is present and a sufficiently large number of spikes is present to estimate the phase distribution. We therefore restricted our analysis to time points where the running speed of the mouse was larger than 5 cm/s and where the sample size was larger than 30 spikes. LFP theta phase and the mean resultant length were computed as described in detail in *Dannenberg et al., 2015*. In short, the instantaneous LFP theta phase was computed from the Hilbert transform of the LFP signal filtered within the theta-range (6–10 Hz). For higher accuracy, the boundaries of this theta range were adjusted separately for each animal as described in detail in *Dannenberg et al., 2019*. We next interpolated the LFP

theta-phase signal and assigned each spike of a theta rhythmic neuron to the instantaneous LFP theta phase at the time of spiking.

## Classification of putative principal neurons and putative interneurons

For classification of single units into putative principal neurons or interneurons, we computed the spike width and firing rate for each unit. The spike width was defined as the time between the peak and the trough of the spike waveform. The distribution of spike widths was bimodal with a minimum between 0.29 ms and 0.34 ms. Units with a spike width <0.29 ms or a firing rate >10 Hz were classified as putative interneurons, the remaining units were classified as putative principal neurons.

## Computation of spatial firing rate maps

Occupancy-normalized spatial firing rate maps were generated by dividing the open-field environment into 3 cm spatial bins. For each spatial bin, an occupancy-normalized firing rate was computed by dividing the total number of spikes falling into that bin by the total time the animal spent in that bin. Spatial rate maps were then smoothed by a 3 cm wide two-dimensional Gaussian kernel.

## Grid score and grid cells

The grid score was used as a measure of hexagonal symmetry of grid fields. The grid score was calculated as follows. We first computed the two-dimensional autocorrelogram of the spatial rate map, extracted a ring that encased the six peaks closest to the center peak excluding the central peak and computed correlations between this modified spatial autocorrelogram and rotated versions of itself after correction of elliptical eccentricity as described in detail in the supplementary methods for computation of gridness measure three in *Brandon et al., 2011*. If hexagonal symmetry is present, those correlations will peak at multiples of 60 degrees and repeat themselves infinitely at a frequency of 6 peaks per rotational cycle. We therefore can compute the power spectrum of those correlations as a function of rotational angle. The power at 60 degree was then defined as the grid score. Compared to previously used grid scores, this grid score largely reduced the number of false-positive grid cells while reliably assigning high grid scores to clear grid cell examples. This grid score is defined as a measure between 0 and 1, with 0 meaning no hexagonal symmetry of firing fields in the spatial rate map or simply 'gridness' and one meaning perfect gridness. Grid cells were defined as cells with a grid score >= 0.19 and at least 300 spikes in either light or dark condition, or across the whole recording session for experiments that started in complete darkness.

## Spatial information

We defined the spatial information as the delta entropy $\Delta H = H_{max} - H$. $H_{max}$ is the maximal amount of randomness given by a completely random distribution of spike locations in the open-field environment defined by $H_{max} = -\sum_{i=1}^{n} p * log_2 p$, with $p = \frac{1}{n}$, where $n$ is the number of spatial bins. $H$ is the observed amount of randomness in the distribution of spike locations given by $H(X|spike) = -\sum_{i=1}^{n} p(x_i|spike) * log_2 p(x_i|spike)$, where $X$ is the animal's location, $x_i$ is the $ith$ spatial bin, $n$ is the number of spatial bins and $p(x_i|spike)$ is the probability that $X = x_i$ when a spike is observed. $\Delta H$ can then be understood as an increase in the amount of order in the distribution of spike locations relative to a completely random (uniform) distribution of spike locations.

## Spatial correlation

Firing rate map similarities were quantified by the spatial correlation between two firing rate maps. To compute the spatial correlation, the spatial bins of the firing rate maps were transformed into a single vector. The spatial correlation was then computed as the Pearson's correlation coefficient between two of those spatially binned firing rate vectors.

## Spatiotemporal correlation

To analyze spatial stability of grid cell firing in the time domain, we used the firing rate map of the light condition to compute the expected firing rate for each of the 20 ms time bins of the animal's

trajectory. We next computed the Pearson's correlation coefficient between the observed 20 ms binned firing rate and the expected 20 ms binned firing rate.

## Spike-triggered firing rate maps

Spike-triggered firing rate maps (*Bonnevie et al., 2013*; *Chen et al., 2016*; *Pérez-Escobar et al., 2016*) were constructed as follows. For each spike, occupancy-normalized firing rates were computed from the next 10 s of data and centered to the spatial bin at the time of the spike. All those firing rate maps were then averaged and smoothed with a Gaussian kernel (SD = 3 cm) to create the spike-triggered firing rate map. To compute the occupancy-normalized firing rate as a function of radial distance, the circumferences of circles with increasing radii ranging from zero to 95 cm were overlaid with the spike-triggered firing rate map. For each of those circles, the spatial firing rates of all bins falling on the circle's circumference were then averaged to compute the spatial firing as a function of radial distance. The location of the first peak in the plot of the firing rate as function of radial distance defined the spacing of the grid cell.

## Border score and border cells

A border score was computed as described in detail by *Solstad et al., 2008*. Border scores ranged from –1 for cells with central firing fields to +1 for cells with firing fields that perfectly aligned with at least one entire wall of the open-field environment. Cells with a border score larger than 0.68 in either the light or dark condition were defined as border cells. The threshold of 0.68 was the 90th percentile of the distribution of border scores from all neurons computed from whole recording sessions.

## Computation of head direction firing rate maps

Head directional firing rate maps were generated by dividing the head directional parameter space into six-degree bins and computing the occupancy-normalized firing rate as the total number of spikes falling into that bin divided by the total time the animal spent in that bin.

## Head directional tuning and head direction cells

Head direction cells were classified based on the criteria reported in *Peyrache et al., 2015* and *Viejo and Peyrache, 2020*. Only cells with at least 10 spikes in each of the light and dark conditions were included in the analysis. Cells were classified as head direction cells if each of the following was true. First, the inverse of the circular standard deviation was greater than one in either the light or the dark condition; second, the p-value of the Rayleigh test was smaller than 0.001 in either the light or the dark condition; third, the peak firing rate in the head directional field was larger than 1 Hz in either the light or the dark condition.

## Identification and classification of speed-modulated neurons

To identify speed-modulated neurons, we used a linear non-linear Poisson (LN) model (*Hardcastle et al., 2017*) which tested for significant modulation of firing rates by position, head direction, LFP theta phase, and/or running speed (*Figure 5*). For a robust analysis, we first screened for cells which showed stable firing across the whole recording session. Cells, whose mean firing rates were below 1 Hz or whose mean firing rates differed more than by a factor of two between the first and second halves of the recording session, were excluded from the analysis. To compare speed modulation of firing rates on the population level, we fitted the data with a quadratic polynomial curve (poly2 fit, MATLAB) and clustered neurons into positively and negatively speed-modulated neurons using hierarchical clustering (clustergram, bioinformatics toolbox, MATLAB).

## Speed score

The speed score was calculated as the Pearson's product moment correlation between instantaneous firing rate and running speed (*Kropff et al., 2015*).

## Experimental design and statistical analysis

Details on statistical analyses on each experiment are reported in the Results section. Statistical tests used in this study include Student's paired t-test and nonparametric statistical tests for repeated

measures, namely Wilcoxon's signed-rank test and Friedman's test. post-hoc testing was performed if the Friedman test yielded a significant result at alpha = 0.05. Data used for analysis of differences in neural dynamics between light and dark conditions were obtained from recordings in a total of 15 mice. Data included LFP signals from 14 out of those 15 mice and 1267 single units from all 15 mice. Out of 1267 recorded single units, 495 single units with stable firing across the recording session and mean firing rates larger than 1 Hz were included in the LN-model analysis of firing rate modulation by running speed; 842 single units which showed spiking activity over at least three light-to-dark and three dark-to-light epoch pairs were used for the analysis of firing rate modulation by visual inputs. Data with green laser light stimulation (related to *Figure 4*) were recorded in two additional mice.

## Code accessibility

Custom MATLAB code used for data analysis can be found at https://github.com/hasselmonians or can be made available upon request. In particular, MATLAB code for analyzing single unit activity can be found at (https://github.com/hasselmonians/CMBHOME; *Chapman and Bogaard, 2020*; copy archived at swh:1:rev:fd5829a812414144df7ff76a177eb6b7263dbc5c); MATLAB code for analyzing firing rate modulation by light and dark conditions can be found at (https://github.com/hasselmonians/light-modulation; *Dannenberg et al., 2020*; copy archived at swh:1:rev:b949888144712b048e20812f4c25c7c3394ee4a6); and MATLAB code for analyzing single unit theta-rhythmicity can be found at (https://github.com/hasselmonians/mle_rhythmicity; *Climer, 2020*; copy archived at swh:1:rev:7c50ba73fa34dbe2586fb0f9e0204bd1e28f3209). Violin plots were created using R software (*R Development Core Team, 2019*).

## Acknowledgements

This work was supported by the National Institute of Mental Health, grant numbers R01 MH60013, R01 MH120073 and R01 MH052090, by the National Institute of Neurological Disorders and Stroke, grant number K99NS116129, and by the Office of Naval Research MURI N00014-16-1-2832 and MURI N00014-19-1-2571, and Office of Naval Research DURIP N00014-17-1-2304.

## Additional information

### Funding

| Funder | Grant reference number | Author |
|---|---|---|
| National Institutes of Health | R01MH60013 | Holger Dannenberg<br>Hallie Lazaro<br>Pranav Nambiar<br>Alec Hoyland<br>Michael E Hasselmo |
| National Institutes of Health | R01MH120073 | Holger Dannenberg<br>Hallie Lazaro<br>Pranav Nambiar<br>Alec Hoyland<br>Michael E Hasselmo |
| National Institutes of Health | R01MH052090 | Holger Dannenberg<br>Hallie Lazaro<br>Pranav Nambiar<br>Alec Hoyland<br>Michael E Hasselmo |
| National Institutes of Health | K99NS116129 | Holger Dannenberg |
| Office of Naval Research | MURI N00014-16-1-2832 | Holger Dannenberg<br>Hallie Lazaro<br>Pranav Nambiar<br>Alec Hoyland<br>Michael E Hasselmo |
| Office of Naval Research | MURI N00014-19-1-2571 | Holger Dannenberg<br>Hallie Lazaro<br>Pranav Nambiar |

| | | Alec Hoyland<br>Michael E Hasselmo |
|---|---|---|
| Office of Naval Research | DURIP N00014-17-1-2304 | Holger Dannenberg<br>Hallie Lazaro<br>Pranav Nambiar<br>Alec Hoyland<br>Michael E Hasselmo |

The funders had no role in study design, data collection and interpretation, or the decision to submit the work for publication.

## Author contributions
Holger Dannenberg, Conceptualization, Data curation, Software, Formal analysis, Funding acquisition, Investigation, Visualization, Methodology, Writing - original draft, Writing - review and editing; Hallie Lazaro, Pranav Nambiar, Investigation, Writing - review and editing; Alec Hoyland, Data curation, Software, Formal analysis, Visualization, Writing - review and editing; Michael E Hasselmo, Conceptualization, Supervision, Funding acquisition, Project administration, Writing - review and editing

## Author ORCIDs
Holger Dannenberg (iD) https://orcid.org/0000-0002-0340-0128
Alec Hoyland (iD) http://orcid.org/0000-0002-4732-5932
Michael E Hasselmo (iD) https://orcid.org/0000-0002-9925-6377

## Ethics
Animal experimentation: This study was performed in strict accordance with the recommendations in the Guide for the Care and Use of Laboratory Animals of the National Institutes of Health. All handling of animals and experimental procedures were approved by the Institutional Animal Care and Use Committee (IACUC) for the Charles River Campus at Boston University under protocol #16-008. All surgery was performed under isoflurane anesthesia and buprenorphine analgesia, and every effort was made to minimize suffering.

## Decision letter and Author response
Decision letter https://doi.org/10.7554/eLife.62500.sa1
Author response https://doi.org/10.7554/eLife.62500.sa2

## Additional files

### Supplementary files
• Transparent reporting form

### Data availability
All data generated or analyzed during this study are included in the manuscript and supporting files. Matlab code for Figure 4 is provided on the laboratory's GitHub page (https://github.com/hasselmonians/light-modulation; copy archived at https://archive.softwareheritage.org/swh:1:rev:b949888144712b048e20812f4c25c7c3394ee4a6/ and https://github.com/hasselmonians/mle_rhythmicity copy; archived at https://archive.softwareheritage.org/swh:1:rev:7c50ba73fa34dbe2586fb0-f9e0204bd1e28f3209/). Source data files have been provided for Figures 2, 3, 4, 5, and 6.

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
