## [Decision Letter]

**Acceptance summary:**

This paper shows how vision contributes to neuronal dynamics in the medial entorhinal cortex, a brain area essential for spatial navigation. Neurons fire rhythmically, in the theta frequency range (6-9Hz), and this intrinsic rhythm is itself modulated by animal's speed, a process believed to enable the integration of animal's displacement (speed and direction of movement) over time to update the internal representation of spatial position. Recording neurons in freely moving mice in light and dark conditions, the authors report that the relationship between speed and rhythmic properties is altered in darkness, resulting in decreased spatial accuracy in the medial entorhinal cortex.

**Decision letter after peer review:**

[Editors’ note: the authors submitted for reconsideration following the decision after peer review. What follows is the decision letter after the first round of review.]

Thank you for submitting your work entitled "Effects of visual inputs on neural dynamics for coding of location and running speed in medial entorhinal cortex" for consideration by *eLife*. Your article has been reviewed by a Senior Editor and three reviewers.

Our decision has been reached after consultation between the reviewers. Based on these discussions, and the individual reviews below, we regret to inform you that your work will not be considered further for publication in *eLife*.

As you can see from the individual reviews below, there was a mix of positive and negative comments about the paper in its current form. Reviewer 1 was generally more positive about the work, while reviewers 2 and 3 were less enthusiastic. As you can see in the individual reviews (and as is explained below), concerns were raised that the data, in their current form, cannot fully support the central claims of the paper.

The conclusion that "mice use an oscillatory speed signal to perform path integration" is based on a correlation between the changes in LFP speed signal and grid cell periodicity when the lights are turned on and off. A concern was raised that there might be other variables with a similar time course (e.g. head-direction tuning, border scores, etc.) and thus that it is unclear which variables are mechanistically related to the grids. Reviewers felt that strengthening the link between oscillatory speed coding and grid cell periodicity would require additional analyses that, depending on the outcome, may change the central conclusions of the work. Reviewers offered several suggestions as to how this major concern could potentially be addressed:

1) A tighter link could be demonstrated between the variables, not just at the aggregate level. Observations speaking to these links could perhaps be shown at the single animal level of the slope and grid cell ensembles.

2) Even if the variables are convincingly shown to be linked, a better elimination of alternative factors is needed. If there are factors that have a similar time course as speed, then a problem arises. If other possible variables in their experiment are unchanged or somehow changed in an uncorrelated manner, it would more strongly support the authors' conclusions. The authors should test whether head-direction cells, and possibly border cells, have a similar profile. This may or may not be feasible depending on how many head-direction cells and border cells were recorded. Some variables that could be considered are:

a) Drift in the preferred direction of head-direction cells

b) Width of head-direction tuning curves

c) Border scores of border cells

d) Width of the firing fields of border cells

One potentially interesting analysis could be to look at distance coding in grid cells when the lights are turned on and off (Figure 4D in Chen et al. and Figure 3 in Perez-Escobar et al.,). The link between speed coding and distance coding was viewed as more direct than between speed coding and grid scores.

Another issue raised relates to the authors' analysis of speed modulation of firing rate. In previous papers (Chen et al., and Perez-Escobar et al.,), speed cells were classified based on the correlation of firing rate and speed, and their tuning curves were modulated in light vs dark. This paper uses the LN model from Hardcastle et al., and the overall tuning curve showed no significant difference in light vs dark. A question was raised about how the authors explain the difference between their results and the results in the previous two papers and how this difference contributes to mechanisms of grid activity.

There are other findings in the manuscript that are interesting to scientists working in this field. For example, the speed code based on the firing rate of neurons was not affected in darkness. However, not all reviewers were convinced that these findings on their own provide sufficiently major mechanistic insights to warrant publication in *eLife*.

In summary, reviewers felt that the work in its present form has promise, but also a number of major concerns were raised. If the authors can address these concerns and choose to do so, we would recommend resubmission of a majorly revised version. While we cannot guarantee a positive outcome for a future submission, the current reviewers indicated that they would be happy to consider a majorly revised version of the work that addresses these major concerns in the future.

The individual reviews are provided below for your information.

Reviewer #1:

Summary:

This exciting paper offers an important advance in an area of intense international interest. As soon as entorhinal grid cells were discovered, the consensus was that grid cells used self-motion signals to support path integration. What has remained elusive and controversial in the 15 years since their discovery is what signals are used and what computations are performed to achieve this. This study considers three candidate signals supporting computations of linear displacement: (1) the slope (not the y-intercept) of the theta frequency to running speed relationship in the LFP; (2) the cellular theta-band interburst frequency to running speed relationship in entorhinal cells; (3) firing rate signals, including in some results specifically the firing rate to running speed relationship in entorhinal cells. Briefly, theta frequency changes are crucial in the oscillatory interference models of grid cells, while rate changes are important in the attractor models. The main findings are: (A) darkness greatly reduces the average slope of the LFP theta frequency to running speed relationship, which is not attributable to changes in acceleration; (B) (somewhat underplayed by the authors) there are interesting dissociations of slope and y-intercept changes elicited by the light/dark transitions, which argue for the ability of the slope to shape grid cell firing independently of the y-intercept; (C) the temporal patterns in the LFP theta-frequency slope changes elicited by light/dark transitions mirror spatial stability changes in grid cells; other candidate signals do not. Finding A was shown by Chen et al., (which these authors frequently refer to). Findings B and C are completely novel, and merit publication in this journal. The authors also wish to emphasise the importance of their findings that their mice grid cells fire relatively robustly in “absolute darkness” and "grid cell firing can be maintained by path integration even in the absence of visual information" (Discussion, see also Abstract). These latter claims may need caveats/rephrasing, but are at any rate to me not actually central to the importance of the paper, and its grounds for acceptance.

I do not have any substantive concerns. I would suggest Points A-E should form part of rebuttal, points F-G need not do.

A) Modify claims regarding grid cell firing being maintained by path integration alone

There are two issues here, likely addressable with rewriting.

1) While the Chen et al., 2016 and Perez-Escobar, 2016 studies have a key result in common, ie grid cell disruption in darkness, Dannenberg and colleagues' description should be more accurate, and give more attention to the methodological differences in these studies. It is simply factually incorrect to say of Chen et al., "their studies carefully removed olfactory, auditory, and tactile cues in addition to the manipulation of visual cues to leave the animals completely disoriented". The walls of the environment remain. The walls present clearly consistent tactile cues, and likely have some smell. It might be better for the authors to describe what these two studies did in more detail. This summary is not convincing.

2) More importantly, the authors seem to claim that path integration alone is sufficient for good-but-not-perfect grid cell signals. Thus: "grid cell firing can be maintained by path integration even in the absence of visual information" (Discussion).

This language is, presumably unintentionally, misleading. In both these excerpts of discussion, the authors are ignoring the role of the walls, which can obviously be sensed haptically as well as visually. The walls and corners provide valuable spatial information regarding direction and distance. E.g.: In a square, depending upon paths taken, the corners should minimise head direction drift to maximum 90 degrees; wall- perpendicular travel from one wall to the other is always a metre. Imagine the same 1x1m area now unbounded in the central region of a 100mx100m walled environment. Would the grid cell signal be similar under darkness to that seen here? Surely not.

B) Slope vs intercept issues

On a first pass, candidate neural signals supporting speed coding can be divided into LFP frequency, interburst-frequency, and firing rate. Of course, however, there is an important further division, between the y-intercept and the slope of these activity-to-speed relationships. Surprisingly, this distinction is under-emphasised both in discursive material, and in the Results.

1) The Introduction should emphasise this more, noting theoretical and empirical dissociations between slope and intercept variation, the former linked to path integrative and spatial coding, the latter not so.

2) The analysis centred on Figure 3 talks of investigating changes in “baseline rates in MEC neurons” (Results), which sounds like changes in y-intercept, but doesn't show that. Analysis of this point would be very helpful. A key point is that there wouldn't necessarily be any theoretical expectation of y-intercept rate control over spatial firing. Figure 4 implies that speed cells showed only intercept not slope changes, a point that should be emphasised more, brought back to the ideas of the augmented Introduction.

3) I would expect more figures showing rate-to-speed plots of single cells in dark and light. Figure 3 and Figure 4 are very general, and should contain specific examples. Figure 4—figure supplement 1 usefully shows examples broken down into cell types (paper re each cell type should be cited if not already done so), but there is no dark vs light comparison.

C) Slope vs grid stability (Results).

Results and Figure 6F. Intriguingly, the time course of changes in spatiotemporal correlation showed a remarkable similarity to the time course of changes in the slope of the LFP theta frequency vs. running speed relationship (Figure 6F). Taken together, these data demonstrate that the representation of space by grid cell firing is correlated to the speed signal by LFP theta frequency and that both of these signals undergo plastic changes at a behavioural time scale of tens of seconds.

This finding, emphasised in the Abstract, is indeed interesting. Is there some way of providing an objective index of this “remarkable similarity” beyond visual inspection?

D) “Complete darkness” and light related issues

1) Are the authors really sure they want to claim the darkness was complete? What does “complete” darkness mean in practice? Note that it is not crucial for the main conclusions of the paper that this darkness is perfect. If they do, they need to offer some physics-type evidence regarding the LEDs, and set out their procedures in much more detail. For instance, "The open field was fully shielded from any remaining light (e.g. door slits) by a black curtain” (Materials and methods) sounds partial. As regards the LEDs, the authors used Axona systems: standard issue LEDs in Axona headstages bleed into the visible spectrum. Indeed, if you look at the LEDs without adaptation in normal lab lighting, it is easy to tell when they are off vs on, as they have a dim red glow when on. What were the part nos and suppliers for their LEDs? A less likely concern regards the night goggles. Did the experimenters ever use the beam function typically provided in night goggles? Again, these may bleed into the visible spectrum.

2) Results: While being visible to the animal as an egocentrically stationary spot of light, it was too weak to illuminate the open field environment and thus did not contribute to the visibility of distal visual cues useful for navigational purposes (e.g. walls or the cue card of the open field arena).

A) What is the evidential basis for this claim “too weak”? B) If all those cues mentioned, i.e. the cue card, the walls and thus, most helpfully, corners, can be visually sensed proximally, that would be helpful for orientation and spatial stability, even if they were not perceivable at a distance. See points above re can path integration alone sustain stable grid cell signals.

E) Datasets

The authors should better specify and offer more rationale re the methodological similarities and differences in the Results, and neuron overlap, of the different datasets in the paper. The following are just two examples. (1) What is the overlap between datasets in Figure 3 and Figure 4? (2) More descriptive details including resampling should be given of the dataset in the Results and Figure 6. The main text talks of n = 7 grids over many days, the Figure legend n = 27 cells.

Reviewer #2:

The current work studied different speed signals in the MEC when mice were navigating in the environment with alternating epochs of light and complete darkness. The authors compared the changes of two main types of speed signals: theta frequency and firing rate. The data showed that visual inputs only affected LFP theta frequency, but not firing rate. They also observed that the alteration in speed coding by LFP theta frequency was correlated with the changes in spatial stability of grid cell firing. The authors proposed that mice use an oscillatory speed signal to perform path integration.

The uniqueness of this study is that it specifically showed the effect of visual input to speed signal and grid cell activity while leaving other sensory information intact. In contrast, in the two previous publications (Chen et al., 2016 and Perez-Escobar et al., 2016) that investigated the role of visual inputs/landmarks in navigation, visual information was nearly the only sensory information available to the animal. However, while the data collection and analyses in the current study were performed very carefully and thoroughly, the results did not well support the rule of oscillatory speed signal in performing path integration.

1) In general, removing visual inputs by placing animal in a completely darkness can have very complicated effects. As the author mentioned, to form stable grid cell activity, signals like head-direction and speed are both potentially important. Visual landmarks and boundaries may also guide the navigation and correct errors of grid activity (Hardcastle, 2015, Hoydal, 2019, Pollock, 2018, Kinkhabwala, 2020…). While all these components could be altered in complete darkness and contribute to the change of grid cell activity, none of them was systematically studied in the current work, except speed encoding. The correlated changes of speed coding by LFP theta frequency and spatial stability of grid cell firing cannot support the hypothesis that mice use an oscillatory speed signal to path integration, because there was no evidence showing any causal effect of the oscillatory speed signal. The authors may have to consider designing different experiments to show the potential causal link.

2) In Figure 1, the authors claimed that the change in the slope of the LFP theta frequency vs running speed in darkness is not an artifact of the change of running behavior. They excluded the potential effect of running acceleration, but how about the overall running speed as they did observe a significant reduction of running speed in darkness (Sup Figure 1A)? To show that the running speed and speed acceleration doesn't change the slop of LFP theta frequency, the author may consider using the light-only data (or dark-only, basically the data without visual differences), bin them according to different ranges of speed acceleration and speed, and compute the slop.

3) In Figure 2, the authors showed a different development of the magnitude and frequency of theta rhythmic firing in light and dark conditions. In addition to showing the data, could the author explain how these different changes related to the change of grid cell activity in darkness at different time points? Among the cells with theta rhythmic firing, is the same development pattern true for different functional cell types?

4) In Figure 3, the authors claimed that the visual inputs altered baseline firing rates. Is it possible that the difference in running speeds in light vs dark also contribute to this effect? When the green light was on, did it change the animal's running speed or running pattern? If so, how much did that contribute to the change of firing rate?

5) In the Results, the authors "corrected for effects on baseline firing rates by random spike deletion to match the number of spikes across light and dark conditions". By doing this, will the cells very likely have similar firing rates in general? One important aspect for the speed cell activity is its degree of firing-rate increasing with speed (the slope of the firing rate vs speed curve) and the baseline firing rate should not contribute to this parameter. What happens to the slope if the authors do not correct the baseline firing rate?

6) In the Discussion: "However, if changes in the speed signal in question are uncorrelated to changes in grid cell firing, we can exclude a causal contribution of that potential speed signal to grid cell firing." This sentence is inappropriate. As mentioned in comment 1, when a speed signal did not correlate to changes to grid cell firing, it only means that this particular speed signal is not causal for the CHANGE of grid cell firing in the current experiment. It does not mean that it is not a potential speed signal to grid cells. Conversely, even though the change of a speed signal did correlate to the changes to grid cell firing, it still does not mean anything about its causal contribution.

7) In the Discussion: "these data suggest that vestibular signals and proprioceptive feedback as well as external geometrical cues are dominant factors maintaining the orientation, scale, and phase of grid cell firing fields, while visual cues are used for refining a self-motion-based cognitive representation of self-location." This is statement is inappropriate. In the darkness when visual information is unavailable, the animal may use a navigation strategy that is different from the visual-rich environment. The presence of grid cell activity in darkness only indicates that the animal can use other sensory information in darkness, but this does not necessarily mean that these types of information still remain dominant when visual information is rich.

Reviewer #3:

The question asked in the manuscript is a very important one: what is the nature of the speed signal integrated by grid cells? Two credible alternatives are put forward.

The evidence for suggesting that one speed signal is more relevant to grid cells is based entirely on the parallel time course of changes in LFP theta and grid cell stability. The assumption here is that the change in the speed signal in the LFP when the lights are turned on or off directly affects grid cell periodicity. However, I am concerned that the reported correlation could be explained by other hidden variables. When visual landmarks are reintroduced, grid scores might be lower for a few seconds for reasons that are not related to speed coding. For example, in darkness, the firing fields of grid cells are likely to drift over time. When the visual landmarks become available, errors that accumulated in darkness might be corrected over a few seconds. This resetting or re-anchoring process is likely to take a few seconds and to affect grid periodicity. This could explain the progressive increase in grid cell periodicity but is not mechanistically related to the change in theta frequency. In my view, additional lines of evidence are needed to strengthen the main conclusion. This could come from investigating distance coding in grid cells and ongoing theta frequency.

1) The comparison with the study of Chen et al., (2016) and Perez-Escobar et al., is interesting, but the current manuscript does not provide the first recordings of grid cells from mice exploring a standard open field in darkness. For example, Allen et al., (2014) showed that the firing fields of grid cells can remain relatively stable in a square open-field with walls. This observation could be mentioned somewhere in the manuscript.

2) When analyzing the rhythmic firing of neurons separately in light and dark and for 5 speed ranges, one concern is that the number of inter-spike intervals to analyze is relatively low for neurons with a mean firing rate below 2-3 Hz. This could potentially affect the conclusions since the difference in theta frequency between the light and dark conditions is relatively small (0.5 Hz). It could be beneficial to show the spike time autocorrelograms separately for the different conditions in a supplementary figure in order to give the reader a sense of the data being processed. A related question is whether the rhythmic firing of high firing rate interneurons has a change more similar to that of the LFP?

3) I would welcome some clarifications regarding the interpretation of the results of two papers cited in the manuscript (Chen et al., 2016; Pérez-Escobar et al., 2016). In the introduction, the authors correctly state that any path integration mechanism ultimately suffers from the accumulation of error over distance traveled or elapsed time necessitating error correction by static environmental sensory inputs such as visual landmark cues or environmental boundaries. They then state that the work of Chen et al., and Perez-Escobar et al., questioned the role of grid cells for path integration-based navigation Results). Since Chen et al., and Perez-Escobar et al., removed most olfactory, auditory, tactile and visual cues from their experimental setup, is it not expected that the grid fields are not stable? Why would this question the role of grid cells in path integration. I would argue that the results of Chen et al., and Perez-Escobar et al., just suggest that errors in the path integration process accumulate faster than previously thought, at least in mice. Also, in both papers, there is evidence that distance coding is partially preserved in darkness.

4) Results. The magnitude of the effects shown in Figure 2A-F is relatively small, and no statistical tests are presented in the text. Ideally, the authors would find a way to show that these effects are statistically significant. For example, are the increase and reduction shown in Figure 2G statistically significant?

[Editors’ note: further revisions were suggested prior to acceptance, as described below.]

Thank you for submitting your article "Effects of visual inputs on neural dynamics for coding of location and running speed in medial entorhinal cortex" for consideration by *eLife*. Your article has been reviewed by Laura Colgin as the Senior Editor, a Reviewing Editor, and three reviewers. The following individuals involved in review of your submission have agreed to reveal their identity: Colin Lever (Reviewer #1), Kevin Allen (Reviewer #2), and Yi Gu (Reviewer #3).

The reviewers have discussed the reviews with one another and the Reviewing Editor has drafted this decision to help you prepare a revised submission.

In their resubmission, the authors now present a characterization of the visual input-evoked changes in neural dynamics in the MEC. New results were added to characterize the impact of visual landmarks on the theta phase modulation of neuron spikes, as well as on the activity of head-direction cells and border cells. All three reviewers are enthusiastic about this resubmission. The paper has been greatly improved and the reviewers agree that most of the concerns raised for the initial submission have now been addressed. In their reports, the reviewers suggest to clarify several points.

1) The authors should better justify why firing rates are less affected by running speed. As pointed out by reviewer #3, the relationship between the two should be more clearly examined, in terms of effect size and distribution over the population as some neurons can show opposite effects. Reviewer #2 suggests that the authors discuss the discrepancies regarding the speed scores between their study and that of Perez-Escobar et al.

2) The authors should clarify Figure 8 as many different conditions are presented. Correlation analysis should be done for intercept and the authors should specify which points are from light and dark conditions.

3) The authors should clarify whether changes in theta rhythmic firing is different for different functional cell types, especially, whether there is anything specific to grid cells.

4) Some statements should be toned down, for example the slope in darkness is reduced but not lost (Discussion).

5) Some discussion and interpretation of the relevance of increased slope to spatial stability of grids, yet not to gridscale, would be helpful for the reader.

6) The 0.68 threshold for border score (Materials and methods) should be justified.

7) The presentation of the result can still be improved, paragraphs should be shorter and have clear headings.

8) The reviewers have suggested several changes in the wording, for example to use “Principal” neurons should be used instead of “pyramidal” as some of the neurons reported in the manuscript may be stellate cells.

Please note that reviewer #1 has pointed out during the discussion that examining the first-half vs second-half of trial grid cell spatial stability is not absolutely necessary, although it would be an interesting addition to the study.

Finally, the authors are encouraged to make their electrophysiological and behavioral data available online.

Below are the reviews in full.

Reviewer #1:

With many additional analyses, the authors have clearly improved an already-interesting manuscript on how the brain might use velocity integration for spatial representation. For instance: (1) The spike-triggered rate maps, and the within-condition as opposed to across-condition normalisation of the speed-by-rate measures, have added insight. (2) Importantly, the authors have quantified the apparently strong relationship between the slope of the theta frequency to running speed relationship and grid cell spatiotemporal stability. The data and analysis presented in Figure 8, in particular 8F and 8G is provocative, with some midpoints in the 8G correlation adding to plausibility. As commented below, this analysis should be extended to the intercept variable too, presumably showing that the intercept is less/not predictive. (3) The strong dissociation between the slope and intercept of the theta frequency to running speed relationship is striking and important. See Figure 1I in light vs part J in light, where the two variables have opposite timecourses, and the breakdown in Dark-to-light transition (right panels of 1K and 1L); slope increases, intercept decreases. In contrast, in darkness, the two variables follow broadly similar or unrelated timecourses. As per point 2, it is the slope that predicts the grid cell spatiotemporal stability. It is perhaps worth stating explicitly why these theta-related observations seem important. Previously perhaps, when inter-species differences seemed more prominent, theta phenomenology, especially relating to frequency rather than amplitude, could be seen as less central to the spatial cognition field. However, it is increasingly clear that much of theta physiology and function seems to be shared across rodents and primates. Most obviously relevant here, the Jacobs lab have now shown in humans that the posterior hippocampal 8Hz theta frequency correlates with speed of movement (Goyal et al., 2020), i.e. even in far-from-realistic locomotor conditions, and this same lab is now reporting theta phase precession in humans. Accordingly, there is a clear need to better understand issues re the slope of the theta frequency to running speed relationship, and its similarities and differences from other theta signals, and from other candidate speed/velocity signals. There is currently very little data on dissociations between the slope and intercept of the theta frequency to running speed relationship. This paper speaks to these needs.

The authors have satisfactorily addressed my comments upon the original submission in the Rebuttal. In answer to the *eLife* question for reviewers, yes, I think the paper is publishable while further revisions are awaited.

Reviewer #2:

In their revised manuscript, Dannenberg and colleagues have changed the focus of the manuscript, and the study is now presented as a characterization of the visual input-evoked changes in neural dynamics in the MEC. New results were added to characterize the impact of visual landmarks on the theta phase modulation of neuron spikes, as well as on the activity of head-direction cells and border cells. Consequently, the possible causal link between the speed signal by local field potential theta frequency and grid cell periodicity is now less central to the manuscript.

This change in focus addresses one of the main issues that I had: the correlative nature of the link between speed signal by LFP theta and grid cell firing.

A second significant improvement in the manuscript is how the authors relate their findings to previously published papers. Both the Introduction and Discussion section were much improved. The potential contribution of external geometrical cues to minimize error accumulation in grid cells' firing activity is acknowledged. The new findings reported on head-direction cells and border cells now provide a more thorough picture of the changes associated with manipulations of visual cues.

Reviewer #3:

The current focus of the revised manuscript is how visual input modulate the encoding of location and running speed. The authors removed the claim about the causality of theta rhythm to grid cell activity, which was the main criticism of previous reviews. The revised manuscript provided a large amount of information about how the MEC network changed after removing visual inputs while leaving other sensory information intact. The data have been thoroughly analyzed and interpreted, and the author nicely addressed most of my concerns. I just have two more questions to their responses:

1) Author's response to previous comment 3:

I think there is some miscommunication about the second half or the previous comment. My point was whether the change in theta rhythmic firing was different for different functional cell types, especially, whether there is anything specific to grid cells. If so, the link between the theta rhythm and grid formation maybe stronger. Alternatively, similar changes may occur in all cells as a network and the change correlated to the reduced accuracy in all cells.

2) Author's response to previous comment 4:

Although the authors gave an explanation about why the firing rates were less likely affected by running speed, I am still very confused by the logic here. It's true that the running speed in different light-level conditions didn't change dramatically whereas cell firing rates changed significantly, however, without determining the effect size of speed change on firing rate, it's hard to conclude that the speed doesn't contribute to the firing rate change. What if a small change in speed is enough to cause a big change in firing rate in some cells? In terms of the opposite changes in the firing rates of simultaneously recorded cells, it's certainly possible that these cells are regulated by speed in opposite manners, as shown by Iwase et al., 2020.

---

## [Author Response]

[Editors’ note: the authors resubmitted a revised version of the paper for consideration. What follows is the authors’ response to the first round of review.]

As you can see from the individual reviews below, there was a mix of positive and negative comments about the paper in its current form. Reviewer 1 was generally more positive about the work, while reviewers 2 and 3 were less enthusiastic. As you can see in the individual reviews (and as is explained below), concerns were raised that the data, in their current form, cannot fully support the central claims of the paper.The conclusion that "mice use an oscillatory speed signal to perform path integration" is based on a correlation between the changes in LFP speed signal and grid cell periodicity when the lights are turned on and off. A concern was raised that there might be other variables with a similar time course (e.g. head-direction tuning, border scores, etc.) and thus that it is unclear which variables are mechanistically related to the grids. Reviewers felt that strengthening the link between oscillatory speed coding and grid cell periodicity would require additional analyses that, depending on the outcome, may change the central conclusions of the work. Reviewers offered several suggestions as to how this major concern could potentially be addressed:

First of all, we would like to thank all reviewers for their excellent comments and suggestions as well as their constructive criticism which helped us to make a major revision and, in our opinion, significantly improve this manuscript. We agree with the reviewers that our previous suggestion that mice use an oscillatory speed signal to perform path integration is not the only possible interpretation of the reported data. The extensively revised manuscript now examines alternative explanations for the observed strong correlation between the slope of the LFP theta frequency vs. running speed relationship and the spatial periodicity of grid cell firing. Overall, the revised manuscript now focuses more clearly on the topic outlined in the title of the manuscript, namely how visual inputs affect neural dynamics underlying the coding of location and running speed in the medial entorhinal cortex. As outlined in the extensively revised abstract, examining neural dynamics in the medial entorhinal cortex as a function of running speed and time offers many insights which are useful to either verify or falsify predictions from computational models or previously proposed mechanisms related to grid cell firing and path integration. One major conclusion from our study is that the speed modulation of firing rates, the proposed speed code by firing rate, does not change between light and dark conditions. We provide evidence that visual inputs largely affect the mean firing rates of many neurons in the medial entorhinal cortex and that such changes in the mean firing rate will artificially affect the slope and y-intercepts of speed tuning curves without altering speed modulation per se. These findings argue against biological models that explained the decrease of spatial periodicity of grid cell firing in complete darkness by a disruption of the speed code by firing rate. We further demonstrate that many parameters of neural activity such as firing rate and the frequency and phase of theta rhythmic firing change more rapidly than the slope of LFP theta frequency and spatial periodicity of grid cell firing. The existence of a slow component of change on a 10-s time scale in both the time courses of the slope of the LFP theta frequency vs. running speed relationship and spatial accuracy of grid cell firing suggest that grid cells may integrate velocity signals over time and that this computation is intrinsically linked to an oscillatory speed signal. This finding is expected to stimulate more research on revising current computational models of grid cell firing and path integration.

We added additional data and analyses with respect to changes in neural correlates of head direction and environmental borders in response to changes in visual inputs. We also added additional analyses with respect to grid cell firing at short time scales, namely spike-triggered firing rate maps and an analysis of distance coding. These data confirm conclusions mentioned in the previous manuscript that the reason for less precise grid cell firing during darkness is an increase in spatiotemporal accuracy of firing as opposed to changes in the scale, orientation, or phase of grid cell firing. Last, we provide evidence that visual inputs are not necessary for the maintenance of grid cell representations if mice can use other cues to remain oriented in the environment, a finding which we think is important to understand previous apparent discrepancies in the literature (e.g. Hafting et al., 2005, and Allen et al., 2014 vs. Pérez-Escobar et al., 2016, and Chen et al., 2016).

1) A tighter link could be demonstrated between the variables, not just at the aggregate level. Observations speaking to these links could perhaps be shown at the single animal level of the slope and grid cell ensembles.

We agree that observations at the single animal level would strengthen the link between the LFP theta oscillatory speed signal and spatial periodicity of grid cell firing. In response to this comment, we analyzed the time course of the slope of the LFP theta frequency vs. running speed relationship, the proposed LFP oscillatory speed signal, and spatial periodicity of grid cells in each of three mice, in which we could perform long-term recordings of grid cells. These data demonstrate that the correlation shown on the aggregate level in the previous manuscript also exists on the single animal level. These data are shown in Figure 8—figure supplement 1.

2) Even if the variables are convincingly shown to be linked, a better elimination of alternative factors is needed. If there are factors that have a similar time course as speed, then a problem arises. If other possible variables in their experiment are unchanged or somehow changed in an uncorrelated manner, it would more strongly support the authors' conclusions. The authors should test whether head-direction cells, and possibly border cells, have a similar profile. This may or may not be feasible depending on how many head-direction cells and border cells were recorded. Some variables that could be considered are:a) Drift in the preferred direction of head-direction cellsb) Width of head-direction tuning curvesc) Border scores of border cellsd) Width of the firing fields of border cells

We thank the reviewers for these suggestions. In response to this comment, we investigated changes in border cells and head direction cells. We now report data showing that the preferred head direction angle and firing fields of border cells remain unaltered but that the accuracy of head directional tuning as well as accuracy of border cell firing decreases in darkness. While we investigated differences between light and dark conditions, we could not investigate the time course of these changes because the necessary long-term recordings of head direction cells and border cells over multiple days were not available. Such long-term recordings with tetrodes in the medial entorhinal cortex are very hard to obtain. Since the main focus of this study was the investigation of grid cells, theta dynamics, and potential speed signals, we aimed at long-term recordings of grid cells but not at long-term recordings of head direction cells or border cells.

One potentially interesting analysis could be to look at distance coding in grid cells when the lights are turned on and off (Figure 4D in Chen et al. and Figure 3 in Perez-Escobar et al.,). The link between speed coding and distance coding was viewed as more direct than between speed coding and grid scores.

We agree and have now included an analysis of spike triggered firing rate maps and distance coding of grid cells. These data show that distance coding is preserved in darkness and that the decrease in spatial accuracy is not due to changes or drifts in the scale, phase, or orientation of grid cell firing.

Another issue raised relates to the authors' analysis of speed modulation of firing rate. In previous papers (Chen et al., and Perez-Escobar et al.,), speed cells were classified based on the correlation of firing rate and speed, and their tuning curves were modulated in light vs dark. This paper uses the LN model from Hardcastle et al., and the overall tuning curve showed no significant difference in light vs dark. A question was raised about how the authors explain the difference between their results and the results in the previous two papers and how this difference contributes to mechanisms of grid activity.

We thank the reviewers for giving us the opportunity to comment more on this important difference. We have now added a paragraph in the Results section explaining that the slope of speed tuning curves depends on the mean firing rate of the cell. If the mean firing rate of a neuron changes, the slope in the speed tuning curve changes even if the speed modulation of firing rate by running speed remains unaltered. We now show the different outcomes in revised main Figure 5 and in Figure 5—figure supplement 1. Importantly, data shown in Figure 5—figure supplement are largely consistent with previous findings. In addition, Figure 5—figure supplement 2 shows examples of individual cells further demonstrating that changes in the mean firing rates of neurons are a major confounding factor when analyzing speed modulation of firing rates. Moreover, this Figure directly compares speed tuning curves (as used in many previous publications) with speed response curves derived from the LN-model. It becomes evident that the speed response curve derived from the LN-model does not differ much from the traditional speed tuning curve. The LN-model does correct for conjunctive coding, though, and also identifies non-linear speed tuning.

The new paragraph dedicated to this comment reads as follows. “Importantly, previous studies have compared speed tuning between conditions after normalization across conditions. […] In this study, we do not present evidence for the validity of either approach but argue in favor of the second approach of analyzing changes in speed modulation after normalization within conditions on the grounds that it is standard in the field to compare tuning of other parameters such as head directional tuning by head direction cells or spatial tuning by place cells, grid cells, or border cells after normalization within conditions.”

Reviewer #1:[…]I do not have any substantive concerns. I would suggest Points A-E should form part of rebuttal, points F-G need not do.A) Modify claims regarding grid cell firing being maintained by path integration aloneThere are two issues here, likely addressable with rewriting.(A1) While the Chen et al., 2016 and Perez-Escobar, 2016 studies have a key result in common, ie grid cell disruption in darkness, Dannenberg and colleagues' description should be more accurate, and give more attention to the methodological differences in these studies. It is simply factually incorrect to say of Chen et al., "their studies carefully removed olfactory, auditory, and tactile cues in addition to the manipulation of visual cues to leave the animals completely disoriented". The walls of the environment remain. The walls present clearly consistent tactile cues, and likely have some smell. It might be better for the authors to describe what these two studies did in more detail. This summary is not convincing.

We thank this reviewer for pointing us to the lack of detail in the summary of methodological differences between the studies by Chen et al., (2016) and Pérez-Escobar et al., (2016), and between these studies and our study. We have revised this paragraph to describe those differences in more detail.

The study by Chen et al., used a 1m x 1m square environment with walls and potential odors attached to those walls. In contrast, the study by Pérez-Escobar et al., used a circular environment surrounded by water and had the walls of the environment removed. In addition, the study by Pérez-Escobar et al., used a white noise generator to block auditory cues, and also changed the positions of visual cues throughout the recording session. Last, mice also underwent a “disorientation procedure” (Pérez-Escobar et al., 2016) before the start of recordings, and the mice had no prior experiences of the recording room. There were also important differences with respect to the temporal arrangement of light and dark epochs. In the majority of trials reported in the Chen et al., (2016) study, light and dark trials were separated from each other and mice were introduced to the recording environment in complete darkness. In contrast, the study by Pérez-Escobar et al., used alternating epochs of light and complete darkness.

Taken together, our study differed from the Chen et al., study in the temporal arrangement of light/dark epochs (Chen et al., used single dark and light session, most often starting in complete darkness; we used alternating epochs usually starting in light); and our study differed from the Pérez-Escobar et al., study in the presence of geometrical, tactile, auditory and likely olfactory cues (not present or masked in the Pérez-Escobar study but present in our study).

Importantly, the preferred angles of head directional firing were largely disrupted in both the studies by Chen et al., (2016) (Figure 2A and Figure 2—figure supplement 2C) and Peréz-Escobar et al., (2016) (Figure 6A). In contrast, we did not find any changes in the preferred head direction angles of head direction cells in our study. Taken together, these data suggest that changes in the head directional system were more pronounced in the studies by Chen et al. and Pérez-Escobar et al., which suggests that mice were more disoriented in those studies compared to our study.

(A2) More importantly, the authors seem to claim that path integration alone is sufficient for good-but-not-perfect grid cell signals. Thus: "grid cell firing can be maintained by path integration even in the absence of visual information" (Discussion).This language is, presumably unintentionally, misleading. In both these excerpts of discussion, the authors are ignoring the role of the walls, which can obviously be sensed haptically as well as visually. The walls and corners provide valuable spatial information regarding direction and distance. E.g.: In a square, depending upon paths taken, the corners should minimise head direction drift to maximum 90 degrees; wall- perpendicular travel from one wall to the other is always a metre. Imagine the same 1x1m area now unbounded in the central region of a 100mx100m walled environment. Would the grid cell signal be similar under darkness to that seen here? Surely not.

We thank this reviewer for bringing this to our attention. We have now revised our language to describe our interpretation of results with respect to the role of visual inputs for grid cell firing and coding of spatial location more precisely. The revised paragraph reads as follows. **“**Taken together, these data suggest that animals need an initial orientation in the environment supported by static visual cues but can then use sensory cues other than visual cues, including wall encounters, to roughly maintain that orientation over long time periods. For maintaining a spatial representation in a completely dark environment, vestibular signals and proprioceptive feedback as well as external geometrical cues may be sufficient to maintain the orientation, scale, and phase of grid cell firing fields, while visual cues may be used for refining the self-motion-based cognitive representation of self-location.”

B) Slope vs intercept issuesOn a first pass, candidate neural signals supporting speed coding can be divided into LFP frequency, interburst-frequency, and firing rate. Of course, however, there is an important further division, between the y-intercept and the slope of these activity-to-speed relationships. Surprisingly, this distinction is under-emphasised both in discursive material, and in the Results.

In response to this comment, we have tried to more strongly emphasize the difference between slope and y-intercept. We are thankful for the opportunity to include more data on changes in the y-intercept in addition to data on changes in the slope. With respect to the analysis of changes in the LFP theta frequency vs. running speed relationship, we now include data on the changes in slope and y-intercept as a function of time in main Figure 1 of the revised manuscript. We also added a paragraph explaining the link of slope to cognition/locomotion/path integration to the Introduction of the revised manuscript (please see our response to next comment below).

(B1) The Introduction should emphasise this more, noting theoretical and empirical dissociations between slope and intercept variation, the former linked to path integrative and spatial coding, the latter not so.

We thank this reviewer for making this suggestion and added the following paragraph to the Introduction of the revised manuscript.

“The slope of the LFP theta frequency vs. running speed relationship in particular has been linked to spatial cognition and locomotion in both hippocampus and MEC (Korotkova et al., 2018; Monaghan et al., 2017; Wells et al., 2013). Interestingly, environmental novelty selectively reduces the slope of the LFP theta vs. running speed relationship (Jeewajee et al., 2008; Wells et al., 2013) and also results in an increase in grid scale (Barry et al., 2012) as predicted by oscillatory interference models of grid cell firing (Burgess, 2008). In contrast, the y-intercept of the LFP theta vs. running speed relationship has been linked to emotion-sensitivity, in particular anxiolysis (Korotkova et al., 2018; Monaghan et al., 2017; Wells et al., 2013).“

(B2) The analysis centred on Figure 3 talks of investigating changes in “baseline rates in MEC neurons” (Results), which sounds like changes in y-intercept, but doesn't show that. Analysis of this point would be very helpful. A key point is that there wouldn't necessarily be any theoretical expectation of y-intercept rate control over spatial firing. Figure 4 implies that speed cells showed only intercept not slope changes, a point that should be emphasised more, brought back to the ideas of the augmented Introduction.

We agree with the reviewer that there wouldn’t be any theoretical expectation of y-intercept rate control over spatial firing or speed modulation of firing rates. However, we would like to point out that the key point of Figure 4 of the revised manuscript (previous Figure 3) was not the analysis of speed tuning curves, on which we present data later in Figure 5 of the revised manuscript. Instead, we investigated if a neuron’s mean firing rate changed between light and dark conditions. In case of a speed cell, a change in the mean firing rate of that cell could present itself as either a change in the y-intercept, a change in the slope, or changes in both the y-intercept and the slope. For better clarity, we now rephrased “baseline firing rates” with “mean firing rates”. The analysis underlying the data presented in Figure 4 of the revised manuscript includes all cell types and is not restricted to speed cells. Therefore, the analysis includes a heterogeneous population of cells including many cells whose firing rates are not modulated by running speed. The data show that visual inputs affect the mean firing rates of many neurons (in most neurons negatively).

However, the reviewer raises a very important point with respect to the analysis of changes in firing rates in speed cells. If the mean firing rate of a speed cell is changed, the speed tuning curve will be affected, even if there is no change in the gain of firing rate as a function of running speed (what we would define as speed modulation). Importantly, there is only one combination of a change in y-intercept and slope, which is consistent with a change in the mean firing rate that does not affect the speed modulation of firing rates, and it is exactly this combination that we observe in our data. We have clarified that point more in the revised version of this manuscript. In response to this and other reviewer comments, we added a new paragraph to the Results section as part of the introduction to results on speed modulation of firing rates shown in Figure 5 of the revised manuscript.

“Importantly, previous studies have compared speed tuning between conditions after normalization across conditions. […] In this study, we do not present evidence for the validity of either approach but argue in favor of the second approach of analyzing changes in speed modulation after normalization within conditions on the grounds that it is standard in the field to compare tuning of other parameters such as head directional tuning by head direction cells or spatial tuning by place cells, grid cells, or border cells after normalization within conditions.”

(B3) I would expect more figures showing rate-to-speed plots of single cells in dark and light. Figures 3 and 4 are very general, and should contain specific examples. Supplementary Figure 2 usefully shows examples broken down into cell types (paper re each cell type should be cited if not already done so), but there is no dark vs light comparison.

We thank the reviewer for this helpful suggestion. We have added a new supplementary figure (Figure 5—figure supplement 2) to the revised manuscript that shows specific examples of speed tuning for single cells in the light and dark condition and also compares the traditional speed tuning curves with tuning curves normalized across conditions and within conditions. The figure also shows the speed response curves derived from the LN-model for comparison with the traditional speed tuning curves.

C) Slope vs grid stability (Results).Results and Figure 6F. Intriguingly, the time course of changes in spatiotemporal correlation showed a remarkable similarity to the time course of changes in the slope of the LFP theta frequency vs. running speed relationship (Figure 6F). Taken together, these data demonstrate that the representation of space by grid cell firing is correlated to the speed signal by LFP theta frequency and that both of these signals undergo plastic changes at a behavioural time scale of tens of seconds.This finding, emphasised in the Abstract, is indeed interesting. Is there some way of providing an objective index of this “remarkable similarity” beyond visual inspection?

We thank the reviewer for this suggestion. We now included a new figure panel (Figure 8G in the revised manuscript) showing the strong correlation (R = 0.97, p = 1.8 x 10^–21^) between the slope of the LFP theta frequency vs. running speed relationship and the spatial stability of grid cells.<bold />*D “Complete darkness” and light related issues*

(D1) Are the authors really sure they want to claim the darkness was complete? What does “complete” darkness mean in practice? Note that it is not crucial for the main conclusions of the paper that this darkness is perfect. If they do, they need to offer some physics-type evidence regarding the LEDs, and set out their procedures in much more detail. For instance, "The open field was fully shielded from any remaining light (e.g. door slits) by a black curtain” (Materials and methods) sounds partial. As regards the LEDs, the authors used Axona systems: standard issue LEDs in Axona headstages bleed into the visible spectrum. Indeed, if you look at the LEDs without adaptation in normal lab lighting, it is easy to tell when they are off vs on, as they have a dim red glow when on. What were the part nos and suppliers for their LEDs? A less likely concern regards the night goggles. Did the experimenters ever use the beam function typically provided in night goggles? Again, these may bleed into the visible spectrum.

We agree with the reviewer that the wording of “complete” darkness needs additional clarification. We have expanded on our description of the experimental condition used for recordings during darkness in the Materials and methods section. We have added the more accurate and cautionary statement that the condition was completely dark from the perspective of the human eye after adaptation to the dark condition for at least five minutes.

The open field environment was located at the end of the recording room surrounded by three walls with no doors or windows and a thick laser-proof black curtain covering the front side. All electrical equipment except the camera for tracking and the Axona headstage and amplifier were located on the other side of the curtain. Room lights at both sides of the curtain were shut off during dark sessions. All LED lights of all electrical equipment were taped off with black duck-tape (at both sides of the curtain). The infrared LEDs of the Axona headstage were also covered with tape to minimize the brightness. This ensured that any light potentially bleeding out into the visible spectrum was blocked. The slits around the door to the recording room were covered with black shower curtain material or black cardboard materials to minimize light leaking into the space in front of the laser-proof curtain. The computer monitors outside of the curtain were also covered with black shower curtains. The experimenter frequently tested for any potential light leaks from the space in front of the laser-proof curtain into the open field environment space behind the laser-proof curtain by standing next to the open field for at least 5 minutes to adjust their eyes to the darkness and carefully screening for any visible dim spots of visible light. Recordings only began when no such leaks were discovered. The night-vision googles were only used for connecting the microdrive implant on the mouse to the tethered headstage of the recording system. The experimenter then left that side of the curtain. We are therefore confident that we recorded under conditions which can be described as completely dark for the human eye. We have mentioned these detailed precautions in the methods and added this phrase to the Materials and methods “completely dark without any visible features to the adapted human eye.”

(D2) Results: While being visible to the animal as an egocentrically stationary spot of light, it was too weak to illuminate the open field environment and thus did not contribute to the visibility of distal visual cues useful for navigational purposes (e.g. walls or the cue card of the open field arena).A) What is the evidential basis for this claim “too weak”? (B) If all those cues mentioned, i.e. the cue card, the walls and thus, most helpfully, corners, can be visually sensed proximally, that would be helpful for orientation and spatial stability, even if they were not perceivable at a distance. See points above re can path integration alone sustain stable grid cell signals.

We thank this reviewer for pointing us to this section of the manuscript which would benefit from additional clarification. As already outlined in the previous manuscript, these experiments were carried out under standard light conditions (room lights turned on during the whole recording session) and not during dark conditions. The egocentrically stationary spot of laser light was too weak to significantly add further to the visibility of cues in the already lighted environment.

<bold />We now added a sentence to the Results section of the revised manuscript for further clarification of the experimental setup.

“Importantly, this experiment was performed with the ceiling lights turned on during the whole recording session with the green laser light being turned on and off in alternating epochs during that session. While being visible to the animal as an egocentrically stationary spot of light, the laser light was too weak to illuminate the open field environment and thus did not change the visibility of distal visual cues useful for navigational purposes (e.g. walls or the cue card of the open field arena).”

E) DatasetsThe authors should better specify and offer more rationale re the methodological similarities and differences in the Results, and neuron overlap, of the different datasets in the paper. The following are just two examples. (1) What is the overlap between datasets in Figure 3 and Figure 4? (2) More descriptive details including resampling should be given of the dataset in the Results and Figure 6. The main text talks of n = 7 grids over many days, the Figure legend n = 27 cells.

We thank this reviewer for bringing this to our attention. We now added a paragraph with respect to the data set underlying this study to the subsection “Experimental design and statistical analysis”, which reads as follows. “The dataset used for analysis of differences in neural dynamics between light and dark conditions was obtained from recordings in a total of 15 mice. This dataset included LFP data from 14 out of those 15 mice and 1267 single units from all 15 mice. From the data set of 1267 single units, 495 single units with stable firing across the recording session and mean firing rates larger than 1 Hz were included in the LN-model analysis of firing rate modulation by running speed; 842 neurons, which showed spiking activity over at least three light-to-dark and three dark-to-light epoch pairs, were included in the analysis of firing rate modulation by visual inputs. Data with green laser light stimulation (related to Figure 4) were recorded in two additional mice.”

Importantly, Figure 4 of the revised manuscript (previous Figure 3) shows data on the modulation of mean firing rates of all cells that showed a significant change in the mean firing rate between light and dark conditions irrespective of the functional classification of those cells into speed, head direction, grid, theta rhythmic or border cells.

In contrast, data shown in Figure 5 of the revised manuscript (previous Figure 4) shows data on the subset of cells that showed significant tuning by running speed (speed cells defined by the LN-model). Out of the 164 cells with significant speed tuning, 153 cells were recorded over at least three light-to-dark and dark-to-light epoch pairs so that we could analyze modulation of firing rate by visual inputs in these cells. Out of those 153 speed-modulated cells, 34 (22.2 %) also showed significant modulation of mean firing rates by visual inputs. This proportion is very similar to the overall proportion of 171 out of 842 (20.3 %) neurons that show significant firing rate modulation by visual inputs. speed-modulated neurons do therefore not differ from other MEC neurons with respect to firing rate modulation by visual inputs. When interpreting these numbers, it is noteworthy that there is a methodological/statistical caveat. When selecting significantly modulated cells, we set an arbitrary threshold (e.g. α = 0.05). However, there may be cells which are not classified as being modulated by running speed at the chosen significance level due to lack in statistical power. The quantification of overlap thus only shows the tip of the iceberg and should be considered a conservative estimate with potentially many more neurons showing some modulation of mean firing rate by visual inputs. We added a new paragraph to the Results section describing those results:

“153 out of the 164 speed-modulated neurons could also be examined for potential co-modulation of firing rates by visual inputs. 34 neurons out of those 153 (22.2 %) speed cells showed significant modulation of mean firing rate by visual inputs. This proportion is very similar to the proportion of 171 out of 842 (20.3%) neurons with significant modulation of mean firing rates by visual inputs observed in the whole data set. Notably, this is a conservative estimate because the firing rate of many more neurons may be modulated by visual inputs without that modulation being detected as significant with the given statistical power and an α level of 0.05.”

In the Results of the previous manuscript, we refer to the sampling of the environment by the mouse. For analyzing spatially periodic firing of grid cells, mice have to visit a sufficient number of spatial bins. There is no clear mathematical/statistical definition of what is “sufficient”. However, the statistical power of detecting differences in spatial firing properties of neurons increases with higher sampling of the environment. We decided to include only those grid cells into the analysis of spatial accuracy as a function of time, from which we could record for least 2 hours in the light and 2 hours in the dark condition.

We thank the reviewer for detecting an error in the Figure legend to Figure 8 of the revised manuscript (Figure 6 of the previous manuscript). The Figure legend should read n = 7 cells. As mentioned previously, for the analysis of the time course of spatial stability of grid cells, we needed to record grid cells over multiple days to have enough statistical power to detect differences in spatial firing as a function of time.

Reviewer #2:[…]1) In general, removing visual inputs by placing animal in a completely darkness can have very complicated effects. As the author mentioned, to form stable grid cell activity, signals like head-direction and speed are both potentially important. Visual landmarks and boundaries may also guide the navigation and correct errors of grid activity (Hardcastle, 2015, Hoydal, 2019, Pollock, 2018, Kinkhabwala, 2020…). While all these components could be altered in complete darkness and contribute to the change of grid cell activity, none of them was systematically studied in the current work, except speed encoding.

We agree with the reviewer that head direction cells and boundary vector and/or border cells are also potentially important for stable grid cell firing. We followed this reviewer’s suggestion and examined the change in head directional tuning and border cells during darkness. The results of this analysis are now included in new Figure 7 of the revised manuscript and in Figure 7—figure supplement 1 and Figure 7—figure supplement 2 of the revised manuscript. These data show that the preferred head direction angle of head direction cells and the location of firing fields of border cells are not altered in darkness. However, similar to the reduction of spatial stability of grid cell firing in darkness, we observed a reduction in the accuracy of head directional tuning and border cell firing.

The correlated changes of speed coding by LFP theta frequency and spatial stability of grid cell firing cannot support the hypothesis that mice use an oscillatory speed signal to path integration, because there was no evidence showing any causal effect of the oscillatory speed signal. The authors may have to consider designing different experiments to show the potential causal link.

We thank this reviewer for raising this important issue of causality. We now made major revisions to the wording of the abstract and main conclusions of this study in the Results section and Discussion section to reduce the appearance that we are claiming causality. E.g., we removed the last sentence of the previous abstract which might have been misinterpreted as claiming causality. We also removed a whole paragraph in the Discussion section of the previous manuscript that claimed that we tested “the relative importance of both speed signals” and discussed “the concept of Granger causality”. Instead, we now added new paragraphs to the Discussion section which explore alternative explanations for the observed correlation between changes in spatial accuracy of grid cell firing and the slope of the LFP theta frequency. The new text reads as follows:

“In summary, one possible explanation for the correlation between changes in spatial accuracy of grid cell firing and the slope of the LFP theta frequency is that the latter serves as a speed signal in the path integration process generating the spatially periodic grid cell firing pattern as suggested by oscillatory interference models of grid cell firing (Burgess, 2008; Burgess et al., 2007; Hasselmo, 2008). […] In that scenario, both the slope of the LFP theta frequency and spatial periodicity of grid cell firing would depend on the history of a velocity signal, which would be consistent with the observed slow saturating exponential component of change.”

2) In Figure 1, the authors claimed that the change in the slope of the LFP theta frequency vs running speed in darkness is not an artifact of the change of running behavior. They excluded the potential effect of running acceleration, but how about the overall running speed as they did observe a significant reduction of running speed in darkness (Sup Figure 1A)? To show that the running speed and speed acceleration doesn't change the slop of LFP theta frequency, the author may consider using the light-only data (or dark-only, basically the data without visual differences), bin them according to different ranges of speed acceleration and speed, and compute the slop.

This reviewer is correct that a change in the mean running speed between light and dark conditions is an important factor. However, we already took this into consideration. Importantly, analyzing the slope of the LFP theta frequency vs. running speed relationship is *not* affected by changes in running speed because the analysis is an analysis as a function of running speed. With respect to changes in acceleration potentially affecting the slope in the theta frequency vs. running speed relationship, we already accounted for that by our analysis shown in Figure 1G of the revised manuscript. We believe we did exactly the kind of analysis this reviewer was suggesting in the comment, namely we binned all data at different acceleration bins, picked all data falling into one of those bins (acceleration bin from -5 cm/s^2^ to +5 cm/s^2^) and demonstrated that the resulting plot (Figure 1G) of theta frequency as a function of running speed shows the same reduction in slope during darkness as when all running acceleration bins were taken into account (Figure 1B). Since all data points in Figure 1G are from the same acceleration bin, acceleration can be ruled out as the cause for the reduction in slope. As mentioned above, differences in the mean running speed between light and dark conditions can also be ruled out because we are analyzing theta frequency as a function of running speed.

3) In Figure 2, the authors showed a different development of the magnitude and frequency of theta rhythmic firing in light and dark conditions. In addition to showing the data, could the author explain how these different changes related to the change of grid cell activity in darkness at different time points? Among the cells with theta rhythmic firing, is the same development pattern true for different functional cell types?

Figure 2 shows the development of frequency and magnitude of theta rhythmic firing as a function of running speed. Unfortunately, the available data do not allow conclusions about the relationship of changes in theta rhythmicity as a function of running speed with changes in grid cell firing as a function of time. Although the reviewer raises an interesting question, the appropriate analysis would run into a statistical power problem because we would have to analyze data on a population of neurons as a function of three parameters, namely light/dark, running speed, and time. A rough estimate reveals that this would require stable recordings of approximately 50 theta rhythmic neurons over approximately7 days, which is not feasible with tetrode recordings. (For each 10-s time bin, we need at least 100 interspike intervals. Assuming a firing rate of 2–3 Hz, we need 40-s of data per condition/time/speed bin. Given that mice spend only about ~5-10% of all time at high running speeds (2–25 cm/s), we need to record ~800 s per condition/time bin. Analyzing the first 2 min would therefore require 12 x ~800 s = ~9600 s = ~160 min of data per condition, i.e. in total ~320 min recording time. One daily session is ~46 min. We would therefore need ~7 recording sessions per cell.)

With respect to the analysis of theta rhythmic firing as a function of running speed or as a function of time in light and dark conditions, we now include a differential analysis of putative principal neurons and putative interneurons in the revised manuscript. These data are reported in the Results section of the revised manuscript and shown in new Supplementary Figures 2 and 3 related to Figure 2 of the revised manuscript. We also examined differences between putative principal neurons and interneurons regarding theta rhythmic firing as a function of running speed and visual inputs. The results of this analysis are reported in the Results section of the revised manuscript and statistics are reported in new Supplementary file 1 and Supplementary file 2 related to Figure 2.

“Population characteristics of running speed-dependent theta rhythmic firing and their changes with removal of visual inputs may differ between pyramidal neurons and interneurons. […] Differentiating between PYRs and INTs revealed a decrease in frequency of theta rhythmic firing as a function of time in both light and dark conditions in INTs but not PYRs (Figure 2—figure supplement 2F and Figure 2—figure supplement 3E).”

4) In Figure 3, the authors claimed that the visual inputs altered baseline firing rates. Is it possible that the difference in running speeds in light vs dark also contribute to this effect? When the green light was on, did it change the animal's running speed or running pattern? If so, how much did that contribute to the change of firing rate?

We thank this reviewer for giving us the opportunity for further clarification of the expected impact of changes in running speed between light and dark conditions on the firing rate of neurons. When considering that impact, a comparison of effect sizes is helpful. As stated in the Results section related to Figure 1D, the mean difference in running speed between light and dark conditions was 7.7 %. In the case of the data presented in Figure 4 of the revised manuscript presenting data on the green light presentation, running speed decreased by 13.1 % during laser stimulation. However, the observed changes in firing rate in the two cells shown in Figure 4E and 4F were 450 % and 137 %. Importantly, both cells were recorded simultaneously during the same recording session but were affected oppositely. The cell shown in Figure 4E increased its firing rate during laser light stimulation, whereas the cell shown in Figure 4F decreased its firing rate during laser stimulation. We can therefore conclude that the slight changes in mean running speed do not significantly contribute to the observed changes in firing rates. The same considerations apply for the effect of running speed on the changes in firing rates caused by transitions between light and dark epochs.

5) In the Results, the authors "corrected for effects on baseline firing rates by random spike deletion to match the number of spikes across light and dark conditions". By doing this, will the cells very likely have similar firing rates in general? One important aspect for the speed cell activity is its degree of firing-rate increasing with speed (the slope of the firing rate vs speed curve) and the baseline firing rate should not contribute to this parameter. What happens to the slope if the authors do not correct the baseline firing rate?

We thank this reviewer for raising this important question. A very similar question was raised by reviewer 1, point B2. First of all, we realized that the wording “baseline firing rate” was confusing and changed the wording to “mean firing rate”, which refers to the firing rate averaged over time. We also realized that we do not need to perform random spike deletion for this analysis. We therefore do not use random spike deletion any more in the revised version of the manuscript and changed the Materials and methods section accordingly. Note that the main results are very similar to the ones reported in the previous version and the conclusions did not change.

The reviewer raises the important question what happens to the slope of the speed tuning curve if the mean firing rate changes and we (a) do not correct for this change and (b) correct for this change? We now give a detailed explanation and description of these two scenarios in the Results section (see text below). We now Figure 5—figure supplement 1 which shows how changes in neurons’ mean firing rates affect their speed tuning curves and speed response curves (derived from the LN-model) if changes in the mean firing rate are not taken into account as a confounding factor. Importantly, if the mean firing rate of a speed cell is changed, the speed tuning curve will always be affected even if there is no change in the gain of firing rate as a function of running speed (what we defined as speed modulation). For further clarification and illustration, we now also include examples of speed tuning curves and speed response curves in Figure 5—figure supplement 2. Data shown in this Figure compare traditional speed tuning curves with speed tuning curves normalized either across light and dark conditions (not correcting for changes in mean firing rate) or within light or dark conditions (correcting for changes in mean firing rate). We now added a new paragraph to the Results section related to data shown in Figure 5.

“Importantly, previous studies have compared speed tuning between conditions after normalization across conditions. […] In this study, we do not present evidence for the validity of either approach but argue in favor of the second approach of analyzing changes in speed modulation after normalization within conditions on the grounds that it is standard in the field to compare tuning of other parameters such as head directional tuning by head direction cells or spatial tuning by place cells, grid cells, or border cells after normalization within conditions.”

6) In the Discussion: "However, if changes in the speed signal in question are uncorrelated to changes in grid cell firing, we can exclude a causal contribution of that potential speed signal to grid cell firing." This sentence is inappropriate. As mentioned in comment 1, when a speed signal did not correlate to changes to grid cell firing, it only means that this particular speed signal is not causal for the CHANGE of grid cell firing in the current experiment. It does not mean that it is not a potential speed signal to grid cells. Conversely, even though the change of a speed signal did correlate to the changes to grid cell firing, it still does not mean anything about its causal contribution.

We thank the reviewer for this comment and agree with the reviewer. We now made major revisions to our wording in the Results and Discussion segments. We also included an analysis of head direction cells and border cells in the revised manuscript in response to comment #1 raised by this reviewer.

We removed the paragraph including the statement that “we can exclude a causal contribution of a potential speed signal by firing rate”. Instead, we added a paragraph about an alternative explanation, namely that the reduction in spatial accuracy of grid cell firing may be caused by the observed reduction in head directional and boundary signals.

“However, our data also showed that the speed modulation of firing rates is not altered in darkness and that the reduction in spatial accuracy of grid cell firing is accompanied by a reduction of accuracy in head directional tuning and a reduction of spatial accuracy of border cell firing (Figure 7). An alternative explanation for the reduction of spatial accuracy in grid cell firing is therefore a reduction of accuracy in head directional (Winter et al., 2015) and/or boundary signals.”

7) In the Discussion: "these data suggest that vestibular signals and proprioceptive feedback as well as external geometrical cues are dominant factors maintaining the orientation, scale, and phase of grid cell firing fields, while visual cues are used for refining a self-motion-based cognitive representation of self-location." This is statement is inappropriate. In the darkness when visual information is unavailable, the animal may use a navigation strategy that is different from the visual-rich environment. The presence of grid cell activity in darkness only indicates that the animal can use other sensory information in darkness, but this does not necessarily mean that these types of information still remain dominant when visual information is rich.

We agree with the reviewer and have revised that statement. In particular, we have removed the sentence claiming a dominant role of vestibular signals, proprioceptive feedback and external geometrical cues. Instead, we now point out that vestibular signals and proprioceptive feedback as well as geometrical cues may be sufficient to maintain grid cell firing in darkness without making the generalization that this applies for the light condition in which visual cues may still be a dominant factor. The new paragraph now reads as follows.

“For maintaining a spatial representation in a completely dark environment, vestibular signals and proprioceptive feedback as well as external geometrical cues may be sufficient to maintain the orientation, scale, and phase of grid cell firing fields. During the light condition, visual cues may be used instead or for refining the self-motion-based cognitive representation of self-location.”<bold />*Reviewer #3:*

[…]1) The comparison with the study of Chen et al., (2016) and Perez-Escobar et al., is interesting, but the current manuscript does not provide the first recordings of grid cells from mice exploring a standard open field in darkness. For example, Allen et al., (2014) showed that the firing fields of grid cells can remain relatively stable in a square open-field with walls. This observation could be mentioned somewhere in the manuscript.

We agree with the reviewer and now cite Allen et al., (2014) together with Hafting et al., (2005) in the Introduction of the revised manuscript.<bold /><italic />

2) When analyzing the rhythmic firing of neurons separately in light and dark and for 5 speed ranges, one concern is that the number of inter-spike intervals to analyze is relatively low for neurons with a mean firing rate below 2-3 Hz. This could potentially affect the conclusions since the difference in theta frequency between the light and dark conditions is relatively small (0.5 Hz). It could be beneficial to show the spike time autocorrelograms separately for the different conditions in a supplementary figure in order to give the reader a sense of the data being processed.

We thank this reviewer for raising this important point and agree that low firing rates could potentially affect the conclusions. We reanalyzed the data shown in Figure 2 applying a more rigorous criterion for including data, namely a minimal number of 100 spike intervals per data bin. This is the minimal number of spike intervals suggested by (Climer et al., 2015). We have now included this criterion in the Methods section under the header “Theta rhythmic firing”: “The minimum number of interspike intervals which need to be present in the autocorrelogram to compute an accurate model is 100 (Climer et al., 2015), and no autocorrelogram model was computed if the number of interspike intervals was smaller than 100.” Notably, the model for estimating parameters of theta rhythmic firing by Climer et al., (2015) was developed for scenarios with low firing rates, in which conventional methods of estimating parameters from the spike time autocorrelogram often fail. Reanalyzing the data by applying the threshold criteria of at least 100 interspike intervals did not change the main results and conclusions as reported in the previous manuscript.

Furthermore, we now include a longitudinal analysis of changes in theta rhythmicity frequency and magnitude as a function of running speed, the results of which are shown in new Figure panels 2F and 2G. This longitudinal analysis only includes theta rhythmic neurons which show significant rhythmicity in each speed bin computed from at least 100 interspike intervals values per speed bin.

We agree with the reviewer that it would be beneficial to show the spike time autocorrelograms for different conditions to give the reader a sense of the data being processed. We now show the spike time autocorrelograms for the light and dark conditions for 24 randomly picked example neurons in Figure 2—figure supplement 4.

A related question is whether the rhythmic firing of high firing rate interneurons has a change more similar to that of the LFP?

We now analyzed changes in theta rhythmic firing separately for putative pyramidal neurons and putative interneurons and present data in Figure 2—figure supplement 2 and Figure 2—figure supplement 3. We provide statistics on the differences between putative pyramidal neurons and interneurons in new Supplementary file 1 and Supplementary file 2 related to Figure 2. Please see also our response to comment #3 of reviewer #2, where we cite the new paragraph in the Results section describing the similarities and differences between putative pyramidal neurons and interneurons with respect to theta rhythmic firing as a function of running speed or as a function of time in light and dark conditions.

3) I would welcome some clarifications regarding the interpretation of the results of two papers cited in the manuscript (Chen et al., 2016; Pérez-Escobar et al., 2016). In the introduction, the authors correctly state that any path integration mechanism ultimately suffers from the accumulation of error over distance traveled or elapsed time necessitating error correction by static environmental sensory inputs such as visual landmark cues or environmental boundaries. They then state that the work of Chen et al., and Perez-Escobar et al., questioned the role of grid cells for path integration-based navigation Results). Since Chen et al., and Perez-Escobar et al., removed most olfactory, auditory, tactile and visual cues from their experimental setup, is it not expected that the grid fields are not stable? Why would this question the role of grid cells in path integration. I would argue that the results of Chen et al., and Perez-Escobar et al., just suggest that errors in the path integration process accumulate faster than previously thought, at least in mice. Also, in both papers, there is evidence that distance coding is partially preserved in darkness.

We thank this reviewer for bringing these points to our attention and agree with the reviewer that grid cell firing is not expected to be stable after removal of most olfactory, auditory, tactile and visual cues from the experimental setup. In our response to an earlier comment (please see our response to comment A1 by reviewer #1), we now provide a more detailed description of the differences in the experimental setups between our study and the studies by Chen et al., and Pérez-Escobar et al.

In response to this comment, we removed the sentence that the work of Chen et al., and Pérez-Escobar et al., questions the role of grid cells for path integration-based navigation.

Furthermore, we now include an analysis of spike-triggered firing rate maps and distance coding in the revised manuscript showing that spatial accuracy of grid cell firing is reduced in spike-triggered firing rate maps, while the spacing of grid fields and distance coding is preserved in darkness. These data are shown in Figure 6—figure supplement 1 and Figure 6—figure supplement 2 of the revised manuscript.

4) Results. The magnitude of the effects shown in Figure 2A-F is relatively small, and no statistical tests are presented in the text. Ideally, the authors would find a way to show that these effects are statistically significant. For example, are the increase and reduction shown in Figure 2G statistically significant?

We thank this reviewer for this suggestion. We now include new Figure panels 2F and 2G showing the results of a longitudinal analysis of the changes in frequency and magnitude of theta rhythmic firing as a function of running speed. Only cells with significantly rhythmic activity in each of the running speed bins were included into this analysis. The statistics on these data are reported in the Results section related to Figure 2F and 2G and read as follows.

“The frequency of theta rhythmic firing increased with running speed during the light condition (p = 4.2 x 10^–8^, linear mixed effects model), but not during the dark condition (p = 0.8085, linear mixed effects model; significant difference in slopes between light and dark, p = 6.8 x 10^–6^) (Figure 2F). Similarly, the magnitude of theta rhythmic firing increased with running speed during the light condition (p = 0.0002, linear mixed effects model), but not during the dark condition (p = 0.0845, linear mixed effects model; difference in slopes between light and dark, p = 9.1 x 10^–6^) (Figure 2G).”

We further show data in Figure 2—figure supplement 1 demonstrating that the results do not depend on arbitrarily chosen significance thresholds for theta spiking rhythmicity.

In addition, we applied a linear regression model to test for differences in the number of neurons as a function of running speed between light and dark conditions. The results are now reported in the Results section related to Figure 2E (previous Figure 2G). The new text reads as follows:

“A linear regression model for running speeds in the range from 5 cm/s to 25 cm/s revealed a significant difference in the number of theta rhythmic firing neurons as function of running speed between light and dark conditions (p = 0.0003)”.

These results were again consistent across a range of different significance thresholds for classification of theta rhythmic firing (Figure 2—figure supplement 2).

We further analyzed changes in frequency and magnitude of theta rhythmic firing as a function of running speed separately for putative principal neurons and interneurons. The results of this analysis are now reported in the Results section related to Figure 2 and are shown in Figure 2—figure supplement 1 and Figure 2—figure supplement 2. The statistical results on differences in theta rhythmic firing between putative pyramidal neurons and interneurons are reported in new Supplementary file 1 and Supplementary file 2 related to Figure 2.

Last, we now report statistics on the changes in theta rhythmic firing frequency as a function of time and between light and dark conditions. We now report in the Results section: “The frequency of theta rhythmic firing was significantly reduced in the dark condition (effect size: –0.381 Hz, t-statistic: –8.598, df: 4316, p = 1x10^–17^, CI: [–0.468 –0.294]).”

Differentiating between pyramidal neurons and interneurons further revealed that the frequency of theta rhythmic firing decreased as a function of time in both light and dark conditions in interneurons but not in pyramidal neurons. These data are shown in (Figure 2—figure supplement 2F and Figure 2—figure supplement 3E), and statistics are reported in the Figure legends.

[Editors’ note: what follows is the authors’ response to the second round of review.]

In their resubmission, the authors now present a characterization of the visual input-evoked changes in neural dynamics in the MEC. New results were added to characterize the impact of visual landmarks on the theta phase modulation of neuron spikes, as well as on the activity of head-direction cells and border cells. All three reviewers are enthusiastic about this resubmission. The paper has been greatly improved and the reviewers agree that most of the concerns raised for the initial submission have now been addressed. In their reports, the reviewers suggest to clarify several points.

We thank the reviewers for their enthusiasm and overall positive comments on the revised manuscript as well as their helpful suggestions to further clarify and improve the presentation of results. We have addressed all comments of individual reviewers in the revised manuscript and provide a detailed point-to-point response to each of the comments in the review summary compiled by the reviewing editor.

1) The authors should better justify why firing rates are less affected by running speed. As pointed out by reviewer #3, the relationship between the two should be more clearly examined, in terms of effect size and distribution over the population as some neurons can show opposite effects. Reviewer #2 suggests that the authors discuss the discrepancies regarding the speed scores between their study and that of Perez-Escobar et al.

We agree with reviewer #3 that there is a possibility that the observed changes in the mean firing rates of neurons could be partially or even fully explained by changes in the animal’s running speed during darkness. To address this issue, we now compute the dark-to-light ratio of the areas under the speed tuning curves in the light and dark conditions, as a measure of running speed-adjusted change in mean firing rate for each neuron. If the change in a neuron’s mean firing rate is just a consequence of a change in the running speed of the animal during darkness, the area under the speed tuning curve is not expected to change in the dark condition because the speed tuning curve plots firing rate as a function of running speed. In that case, the running speed-adjusted dark-to-light ratios of all neurons would cluster around 1. If, however, changes in running speed do not contribute to the changes in firing rates, the speed-adjusted dark-to-light ratios would cluster around the same non-zero value as the non-adjusted dark-to-light ratios computed from the mean firing rates (<1 for negatively modulated neurons, > 1 for positively modulated neurons).

In the population of 139 negatively modulated neurons, we found that the speed-adjusted dark-to-light ratio was 1.4 percentage points lower than the non-adjusted dark-to-light ratio indicating that we previously slightly underestimated the decrease in firing rates during darkness. For the subpopulation of 32 positively modulated neurons, we found that the speed-adjusted dark-to-light ratio was 17 percentage points lower compared to the non-adjusted dark-to-light ratio indicating that changes in running speed resulted in an overestimation of the effect size. These data are now shown in Figure 4—figure supplement 2 of the revised manuscript.

Notably, even after the adjustment for running speed, the light versus dark changes in firing rates were very similar with respect to effect size in negatively and positively modulated neurons; on average, negatively and positively modulated neurons decreased and increased their firing rate by 39% and 36%. These data demonstrate that the changes in the mean firing rates of many neurons—including speed cells—during darkness are real and not an artifact of the interaction between speed modulation of firing rates and small changes in the animals’ running speed.

The added text in the Results section of the revised manuscript reads as follows: “Since we observed slight but significant changes in the animals’ mean running speeds during darkness and many neurons’ firing rates are modulated by running speed, we first tested whether the observed light versus dark changes in mean firing rates of neurons are merely a consequence of the interaction between speed modulation of firing rates and changes in the animal’s running speed. Towards that goal, we computed the dark-to-light ratios of changes in mean firing rates for negatively (0.62 ± 0.16) and positively (1.53 ± 0.28) modulated neurons, respectively, and compared these observed ratios with running speed-adjusted dark-to-light ratios computed from the areas under the speed tuning curves in light and dark conditions (Figure 4—figure supplement 2). After running speed adjustment, dark-to-light ratios were 0.61 ± 0.17 (mean adjustment = –0.014, CI = [–0.002 –0.026], t = –2.31, df = 138, p = 0.023, t-test) and 1.36 ± 0.23 (mean adjustment = –0.174 CI = [–0.112 –0.235], t = –5.78, df = 31, p = 2.3 x 10^–6^) for negatively and positively modulated neurons, respectively. Notably, even after adjustment for running speed, the dark-to-light effect sizes were very similar for negatively and positively modulated neurons, namely a 39% decrease and a 36% increase in firing rates, respectively. These data demonstrate that the observed dark versus light changes in mean firing rates are not an artifact of the interaction between speed modulation of firing rates and changes in the animals’ running speed.”

We thank reviewer #2 for the suggestion to discuss the seeming discrepancy between this study and the study of Pérez-Escobar et al., (2016) with respect to changes in speed scores between light and dark conditions. One likely explanation for this seeming discrepancy is the different way speed cells were identified in the two studies. The study by Pérez-Escobar et al., (2016) used a speed score threshold as the classification criterion for speed cells, which largely identifies linearly tuned speed cells because the speed score is defined as the linear correlation between firing rate and running speed. Notably, the speed score is sensitive to changes in mean firing rates because fewer spikes result in noisier firing rates and therefore a decrease in correlation with running speed as can be shown by random spike deletion. If the firing rates of many neurons decrease during darkness as observed in both this and the Pérez-Escobar et al. (2016) study, speed scores of linearly tuned speed cells are expected to decrease during darkness, as was indeed the case in the Pérez-Escobar et al., (2016) study.

However, our current study used an LN-model (Hardcastle et al., 2018) to identify speed cells and therefore includes many cells with non-linear speed tuning curves. The speed score is therefore a sub-optimal measure for speed tuning in our study. We added the following text to the Discussion section to discuss this point: “A previous study (Pérez-Escobar et al., 2016) has found that speed scores were changed between light and dark conditions. However, that study used a speed score threshold to identify speed cells which is biased for identification of linearly modulated neurons because the speed score is a measure of linear correlation. In contrast, the current study used the LN-model to identify speed cells. As a consequence, our data set includes many speed cells with non-linear speed response curves, where the speed score is a sub-optimal measure. Like the slope of the speed tuning curve, the speed score is also a function of firing rate and previously reported changes in the speed score of speed cells during darkness may have been driven by changes in mean firing rates as opposed to changes in the running speed-dependent gain in firing rate.”

To further clarify the finding that the changes in mean firing rates during darkness explain the observed changes in the slopes of speed tuning curves, we now distinguish between the slope of a speed tuning curve and the running-speed dependent gain in firing rate as two alternative measures of speed modulation.

2) The authors should clarify Figure 8 as many different conditions are presented. Correlation analysis should be done for intercept and the authors should specify which points are from light and dark conditions.

Following the suggestions by reviewer #1, we added red and black colors to Figure 8G to illustrate which data points are from the light condition (red color) and which are from the dark condition (black color). We now also provide a more detailed description of the data shown in Figure 8G in the revised Figure legend including the number of datapoints going into the correlation and the breakdown into light and dark conditions. The new text in the Figure legend to Figure 8G now reads: “Each data point corresponds to a 10-s time bin within the first 180-s period after the start of the light (red) or dark (black) condition. Data points show the mean values (same data as in F). Grey line shows the linear regression, n = 36 (18 timepoints x two conditions), R = 0.966; p = 1.83 x 10^–21^.”

In addition, we now also report in the Results section of the revised manuscript the correlation between grid cell stability and slope of theta frequency to running speed separately for light and dark conditions in addition to the correlation value computed across light and dark conditions. The added text in the Results section reads as follows: “The time courses of changes in grid cell stability and the slope of the LFP theta frequency vs. running speed relationship were also correlated within light and dark conditions (R = 0.825, p = 2.55 x 10^–5^ in the light condition; R = 0.490, p = 0.039) further confirming the link between grid cell stability and the LFP theta frequency to running speed relationship.” This additional information confirms that the correlation exists also within light and dark conditions further supporting the previous conclusion that grid cell stability is correlated to the slope of the theta frequency to running speed relationship (within and across conditions).

Following reviewer #1’s suggestion, we now analyzed the correlation between changes in grid cell stability and y-intercepts of the LFP theta frequency vs. running speed relationship in addition to our previous analysis of changes between grid cell stability and the slope of the LFP theta frequency. The results of this analysis show a negative correlation between grid cell stability and y-intercept of the LFP theta frequency vs. running speed relationship during the light condition and no significant correlation during the dark condition. These data are shown in new Figure 8—figure supplement 2 and described in the Results section as follows: “In contrast, changes in the y-intercept of the LFP theta frequency vs. running speed relationship as a function of time were negatively correlated with changes in grid cell stability in the light condition (R = –0.633, p = 0.0048, n = 18 time points) and uncorrelated to changes in grid cell stability in the dark condition (R = 0.056, p = 0.83, n = 18 time points) (Figure 8—figure supplement 2).”

3) The authors should clarify whether changes in theta rhythmic firing is different for different functional cell types, especially, whether there is anything specific to grid cells.

In response to this comment, we analyzed changes in theta rhythmic firing as a function of light and dark condition and as a function of running speed for grid cells (n = 20), head direction cells (n = 12), and border cells (n = 9). We found very similar changes in theta rhythmic firing across those three functional cell types indicating that the observed changes in theta rhythmic firing are ubiquitous across principal neurons in the medial entorhinal cortex. These data are shown in three new supplementary figures related to Figure 2 (Figure 2—figure supplement 4, Figure 2—figure supplement 5, and Figure 2—figure supplement 6) showing data for grid cells, head direction cells, and border cells, respectively. We also added these results to the Results section. The newly added text reads as follows: “We next addressed the question whether the observed changes in theta rhythmic spiking activity in principal neurons was different for different functional cell types such as grid cells, or alternatively was similar across principal cells. We found that changes in theta rhythmic spiking activity as a function of running speed during and between light and dark conditions were similar across grid cells, head direction cells, and border cells (Figure 2—figure supplement 4, Figure 2—figure supplement 5, and Figure 2—figure supplement 6) and indistinguishable from all PNs, indicating that changes in theta rhythmic activity is ubiquitous across different functional cell types in the medial entorhinal cortex.”

4) Some statements should be toned down, for example the slope in darkness is reduced but not lost (Discussion).

In response to this comment, we carefully read through the whole manuscript again to identify sentences that could be toned down. The sentence in the Discussion referenced by the reviewer now reads “However, an alternative speed signal by LFP theta frequency—the linear correlation between theta frequency and running speed—is a function of visual inputs and reduced in darkness.”

We further made a change to the Abstract by replacing the sentence “In contrast, visual inputs do not affect the speed modulation of firing rates” with the sentence: “In contrast, visual inputs do not alter the running speed-dependent gains in neuronal firing rates”.

Moreover, we changed the wording in the last summary sentence of the paragraph describing the analysis of changes in speed modulation during light and dark conditions from “In summary, these data demonstrate that the speed modulation of firing rates in the MEC remain unaltered in the absence of visual inputs, even if mean firing rates change substantially” to “In summary, these data demonstrate that the running speed-dependent gains in firing rates of MEC speed cells remain unaltered during the absence of visual inputs, even if mean firing rates and the slopes of speed tuning curves change substantially.”

5) Some discussion and interpretation of the relevance of increased slope to spatial stability of grids, yet not to gridscale, would be helpful for the reader.

The reviewers raise an interesting question, namely why is the slope of the LFP theta frequency vs. running speed relationship correlated to the stability of grid cell firing but not the scale of grid cell firing. We believe that more research is needed to gain a better mechanistic understanding of the interesting link between theta frequency and grid cell firing. We can only speculate about the mechanistic details of that link. One potential mechanism is that the reduced slope results in a lower resolution in a potential readout mechanism due to reduction in the dynamic range of an oscillatory speed signal resulting in less accurate readout due to increased variability. Importantly, this would require a rescaling of the oscillatory speed signal which may be facilitated by knowledge of the distances between the borders of a familiar environment. Previous studies have shown that environmental novelty reduces the slope of the LFP theta frequency to running speed relationship to a similar degree as darkness (Wells et al., 2013) but resulted in an increase in grid scale (Barry et al., 2012) as opposed to an increase in spatial stability. These data may indicate that a reduced slope in the LFP theta frequency to running speed relationship may also be the consequence of underestimating the distances between borders in a novel environment resulting in a larger grid scale.

We added the following discussion and possible interpretation of the relevance of increased slope to spatial stability of grid cells yet not to grid scale, to the Discussion section:

“We demonstrate in this study that the slope of the LFP theta frequency to running speed relationship and grid cell stability change as a function of visual inputs and as a function of time in a familiar environment. […]Taken together, these data may indicate that in a novel environment, a reduced slope in the LFP theta frequency to running speed relationship may be the consequence of underestimating the distances between borders resulting in a larger grid scale but that in a familiar environment, the lack of error correction in a path integration mechanism may result in a reduction of the dynamic range in an oscillatory speed signal by LFP theta frequency resulting in less accurate speed coding.”

6) The 0.68 threshold for border score (Materials and methods) should be justified.

Previous papers have defined border cells as cells with a border score > 0.5, which is typically in the range between the ninetieth and ninety fifth percentile of border scores from all neurons (Solstad et al., 2008; Bjerkness et al., 2014). In our data set, we set the threshold to the ninetieth percentile (0.68) of the distribution of border score values computed from whole recording sessions (light and dark epochs combined). We updated the text in the Methods sections which now reads as follows: “Cells with a border score larger than 0.68 in either the light or dark condition were defined as border cells. The threshold of 0.68 was the 90^th^ percentile of the distribution of border scores from all neurons computed from whole recording sessions.”

7) The presentation of the result can still be improved, paragraphs should be shorter and have clear headings.

We shortened most paragraphs and added headings to structure the Results section and increase readability.<bold />8 The reviewers have suggested several changes in the wording, for example to use “Principal” neurons should be used instead of “pyramidal” as some of the neurons reported in the manuscript may be stellate cells.

We changed “pyramidal” to “principal” throughout the text and in the Figure legends and Table legends.<bold />Please note that reviewer #1 has pointed out during the discussion that examining the first-half vs second-half of trial grid cell spatial stability is not absolutely necessary, although it would be an interesting addition to the study.

We agree with reviewer #1 that examining potential changes in grid cell stability between the first and second halves of trials would be interesting. However, we decided to include this analysis in a future manuscript. <bold />Finally, the authors are encouraged to make their electrophysiological and behavioral data available online.

We are working on organizing and annotating the electrophysiological and behavioral data to make them publicly available in the near future.

Unrelated to a specific reviewer comment, we decided to rearrange panels in main Figure 5 and Figure 5—figure supplement 1 so that the main Figure 5 now shows data on speed response curves that are normalized across light conditions (Figure 5A and C) next to speed response curves that are normalized within conditions (Figure 5B and D). Showing these figure panels next to each other in the main figure further clarifies the distinction between the slope of a speed response curve and the running-speed dependent gain in firing rate as two alternative measures of speed modulation.